# Nitrogen signaling factor triggers a respiration-like gene expression program in fission yeast

Shin Ohsawa[1], Michaela Schwaiger[1,2], Vytautas Iesmantavicius[1], Rio Hashimoto[3,4], Hiromitsu Moriyama [ID][4], Hiroaki Matoba [ID][5], Go Hirai [ID][5,6], Mikiko Sodeoka [ID][6], Atsushi Hashimoto[3], Akihisa Matsuyama [ID][3,7], Minoru Yoshida [ID][3,8,9], Yoko Yashiroda [ID][3,7 ✉] & Marc Bühler [ID][1,10 ✉]

## Abstract

**Microbes have evolved intricate communication systems that enable individual cells of a population to send and receive signals in response to changes in their immediate environment. In the fission yeast *Schizosaccharomyces pombe*, the oxylipin nitrogen signaling factor (NSF) is part of such communication system, which functions to regulate the usage of different nitrogen sources. Yet, the pathways and mechanisms by which NSF acts are poorly understood. Here, we show that NSF physically interacts with the mitochondrial sulfide:quinone oxidoreductase Hmt2 and that it prompts a change from a fermentation- to a respiration-like gene expression program without any change in the carbon source. Our results suggest that NSF activity is not restricted to nitrogen metabolism alone and that it could function as a rheostat to prepare a population of *S. pombe* cells for an imminent shortage of their preferred nutrients.**

**Keywords** Nitrogen Signaling Factor; Fission Yeast; Nitrogen Catabolite Repression; Cell-to-Cell Communication; Mitochondrial Respiration
**Subject Categories** Metabolism; Microbiology, Virology & Host Pathogen Interaction

## Introduction

Unicellular organisms must be able to sense and react to changes in their immediate environments to maximize growth and survival. Classic examples are acute responses to starvation or other stressful conditions such as osmotic, oxidative, or temperature stress, heavy metals, or DNA damage. Besides the shortage of nutrients, they must also accommodate changes in the abundance of nutrients that they favor. For this, catabolite repression (CR) strategies are utilized by various species of bacteria and fungi, which enables them to preferentially utilize high-quality nutrients (Nair and Sarma, 2021). By modulation of CR, they can accommodate nutritional changes in their environment.

Carbon and nitrogen are the main energy sources for sustaining biosynthetic processes and must be taken up in large quantities from the environment. Different species have preferences for certain carbon and nitrogen sources that they rapidly metabolize to generate energy for growth and niche colonization. In the presence of these favored energy sources, utilization of less preferred carbon and nitrogen sources is repressed, phenomena referred to as carbon and nitrogen catabolite repression (CCR and NCR, respectively). CCR and NCR can impact the virulence of human pathogens, which must adapt to their different niche environments. Therefore, a good mechanistic understanding of carbon and nitrogen metabolism in microbes is of great biomedical importance (Ries et al, 2018). It is also relevant for industrial applications that rely on microorganisms for the generation of valuable bio-products (Nair and Sarma, 2021). For instance, CCR is essential in the brewing industry for efficiently synthesizing ethanol through the fermentation pathway using carbon sources. On the other hand, NCR is necessary for effectively utilizing nitrogen sources and increasing biomass. However, excessive NCR has been found to lead to the accumulation of substances such as urea and proline, contributing to the deterioration of quality in products like wine (Zhao et al, 2016).

The selective usage of one energy source over another is generally considered to be transient, acute, and cell-autonomous. However, recent studies revealed the existence of chemical communication between cells that transforms metabolic responses (Jarosz et al, 2014; Takahashi et al, 2012). This can occur between cells of the same or across species, or even kingdoms, as exemplified by the bacterial metabolite lactic acid, which serves as a diffusible signal that enables neighboring *Saccharomyces cerevisiae* cells to bypass CCR (Jarosz et al, 2014; Garcia et al, 2016). An intraspecies chemical communication system that regulates NCR was discovered in the fission yeast *Schizosaccharomyces pombe* (Sun et al, 2016; Takahashi et al, 2012). Here, uptake of the branched-chain amino acids (BCAA) leucine (Leu), isoleucine (Ile), and valine (Val) is suppressed in the presence of

[1]Friedrich Miescher Institute for Biomedical Research, Fabrikstrasse 24, 4056 Basel, Switzerland. [2]Swiss Institute of Bioinformatics, 4056 Basel, Switzerland. [3]Chemical Genomics Research Group, RIKEN Center for Sustainable Resource Science, Wako, 351-0198 Saitama, Japan. [4]Graduate School of Agriculture, Tokyo University of Agriculture and Technology, Fuchu, 183-8538 Tokyo, Japan. [5]Graduate School of Pharmaceutical Sciences, Kyushu University, Maidashi Higashi-ku, 812-8582 Fukuoka, Japan. [6]Catalysis and Integrated Research Group, RIKEN Center for Sustainable Resource Science, Wako, 351-0198 Saitama, Japan. [7]Molecular Ligand Target Research Team, RIKEN Center for Sustainable Resource Science, Wako, 351-0198 Saitama, Japan. [8]Office of University Professors, The University of Tokyo, Bunkyo-ku, 113-8657 Tokyo, Japan. [9]Collaborative Research Institute for Innovative Microbiology, The University of Tokyo, Bunkyo-ku, 113-8657 Tokyo, Japan. [10]University of Basel, Petersplatz 10, 4003 Basel, Switzerland. ✉E-mail: ytyy@riken.jp; marc.buehler@fmi.ch

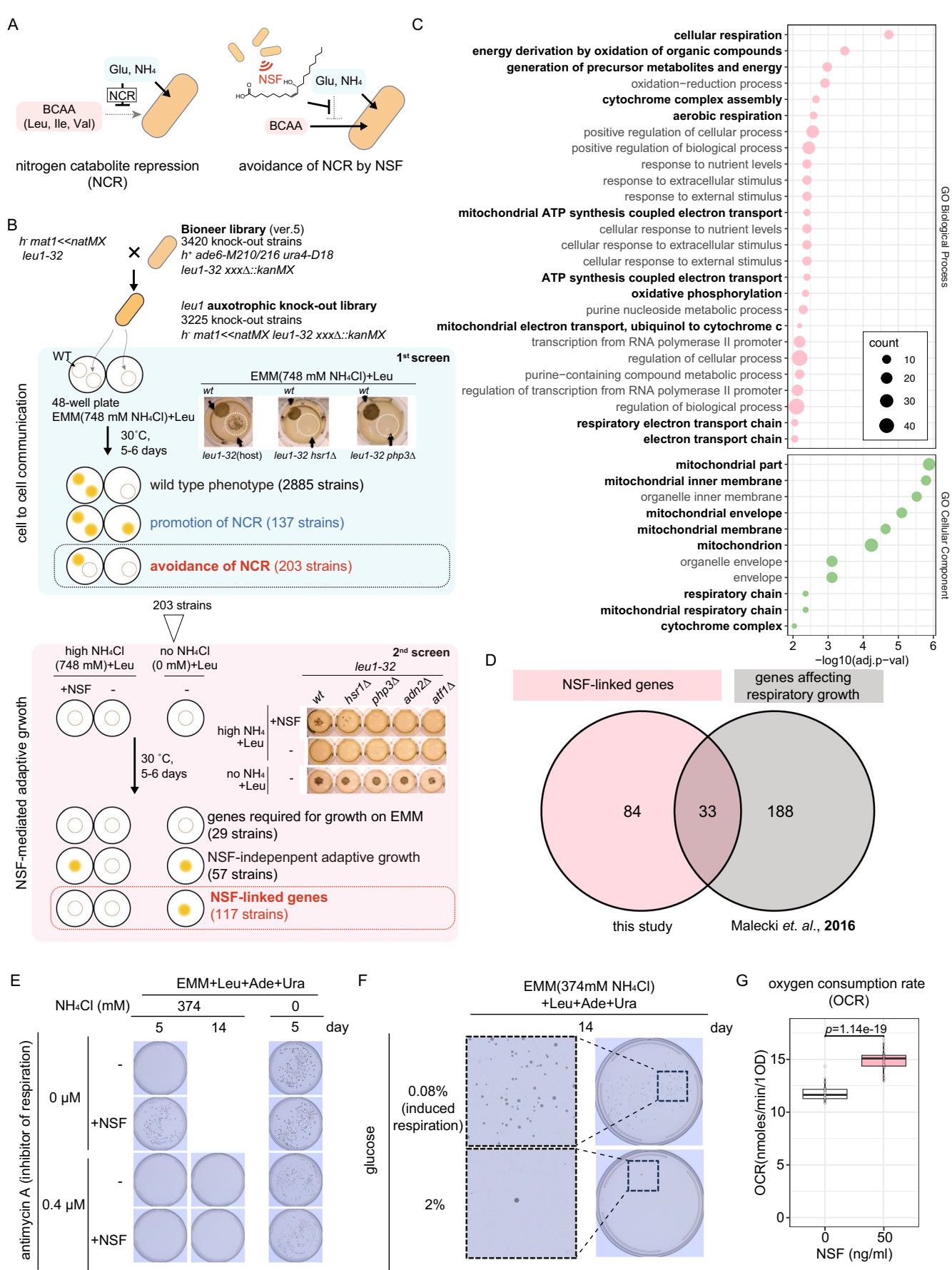

◄

**Figure 1. NSF-mediated adaptive growth is linked to respiration.**

(A) Scheme illustrating NCR. Cellular uptake of branched-chain amino acids (BCAA) is suppressed in the presence of glutamate (Glu) or ammonium (NH₄) (left). This is bypassed upon exposure to NSF (right). (B) Outline of the genetic screens performed in this study. (C) GO enrichment analyses of NSF-linked genes by AnGeLi (Bitton et al, 2015). AnGeLi applies the two-tailed Fisher's exact test and two-sided Wilcoxon rank-sum test as multiple statistical tests. GO terms linked to mitochondria are written in bold letters. (D) Venn diagram showing the overlap of NSF-linked genes revealed by this study and genes that were previously shown to affect respiratory growth (Malecki and Bähler, 2016). (E) NSF-mediated adaptive growth in *leu1-32 ade6-M216 ura4-D18* (ED668) under high ammonium conditions when respiration is inhibited. (F) Cellular growth in ED668 under high ammonium conditions when respiration is induced by low glucose concentrations (0.08%). (G) Oxygen consumption rate of cells (ED668) that were exposed to synthetic NSF. Cells were exposed to 50 ng/mL NSF for 8 h at low cell density (OD 0.01). *p* value was calculated using a two-sided *t*-test. The center line inside the box represents the median of the data, bounds of the box correspond to the interquartile range with the bottom and top of the box indicating the first and third quartiles, and bounds of whiskers extend to the minimum and maximum values within the range of non-outlier data. Each condition was done in technical triplicates. Source data are available online for this figure.

high-quality nitrogen sources such as ammonium or glutamate, likely because the expression of transporters or permeases that are needed for the uptake of poorer nitrogen sources are downregulated, as it was shown in *S. cerevisiae* (Zhang et al, 2018). Thus, *S. pombe* cells may rely predominantly on their own BCAA synthesis for at least the initial stages of growth under NCR conditions. This would explain why auxotrophic mutants, such as Leu-auxotrophic *leu1-32* or BCAA-auxotrophic *eca39Δ* cells, are unable to grow on a minimal medium containing a high-quality nitrogen source, even when supplemented with BCAA (Takahashi et al, 2012; Sun et al, 2016). Notably, though, the growth of *leu1-32* and *eca39Δ* cells can be restored when prototrophic cells are plated adjacent to the mutant cells. This adaptive growth is not observed near *S. cerevisiae* cells (Takahashi et al, 2012; Sun et al, 2016). This intriguing observation suggested that the prototrophic cells secrete a diffusible molecule that revokes NCR in the auxotrophic cells. Indeed, the fatty acids 10(*R*)-acetoxy-8(*Z*)-octadecenoic acid and 10(*R*)-hydroxy-8(*Z*)-octadecenoic acid were found to be secreted by the prototrophic cells, now referred to as nitrogen signaling factors (NSFs). Importantly, synthetic NSFs are sufficient to bypass NCR, i.e., *leu1-32* and *eca39Δ* cells can grow on a minimal medium containing a high-quality nitrogen source when supplemented with BCAA and synthetic NSF (Sun et al, 2016). Thus, NSFs are part of a species-specific communication system that enables *S. pombe* cells adapting to changing nutritional conditions by evading NCR. However, the pathways in which NSFs function in such adaptation are not known.

Here, we employed forward genetics, genomics, and chemical biology approaches to gain insights into NSF-mediated adaptive growth of *S. pombe* cells. We show that mitochondrial respiration counteracts NCR, that NSF exposure triggers a change from a fermentation- to a respiration-like gene expression program, and that the mitochondrial sulfide:quinone oxidoreductase Hmt2 is a direct target of NSF. Thus, NSF is a transmissible signal that turns on mitochondrial respiration to maximize growth in response to changes in the availability of nutrients.

## Results

### Identification of genes that are required for NSF-mediated adaptive growth

Because cellular uptake of BCAA is suppressed in the presence of high ammonium concentrations, *leu1-32* auxotrophs are unable to grow unless they are exposed to a diffusible signal such as NSF that

is secreted by neighboring prototrophs (Fig. 1A). To identify genes that are required for NSF-mediated adaptive growth, we decided to perform genetic screens using the BIONEER haploid *S. pombe* Genome-wide deletion mutant library that covers 3420 non-essential gene knock-out strains (Kim et al, 2010) (Dataset EV1). Notably, this library harbors auxotrophic mutations not only in *leu1*, but also in the *ura4* and *ade6* genes (*leu1-32, ura4-D18*, and *ade6-M210* or *M216*). To exclude potential confounding effects that might stem from the *ura4* and *ade6* auxotrophic alleles (Takahashi et al, 2012), we first crossed the *h*⁺ BIONEER library with an *h*⁻ strain that had an auxotrophic mutation in the *leu1* gene only and a nourseothricin resistance marker downstream of the *mat1-Mc* gene (*leu1-32 mat1≪natMX*, Fig. 1B). As a result, we could isolate nourseothricin-resistant strains that were of the *h*⁻ mating-type. Hence, we could rule out slight differences in growth that could be conferred by different mating types (*h*⁺ or *h*⁻). In total we recovered 3225 individual *h*⁻ knock-out strains that are auxotrophic for leucine specifically (Leu-auxotrophic knock-out library; Fig. 1B; Dataset EV1).

We performed a first screen to identify those genes that are required when auxotrophs must communicate with prototrophs to sustain growth under high ammonium conditions (cell-to-cell communication). For this, the 3225 Leu-auxotrophic knock-out strains were spotted in 48-well high ammonium agar plates (EMM(748 mM NH₄Cl)+Leu; severe NCR condition) either alone, or in the vicinity of a wild-type (*wt*) prototroph. When grown for an extended period (>1 week), instances of cell-to-cell communication-independent growth can be observed in *S. pombe*. Therefore, we assessed growth after 5 or 6 days of cultivation in our screen. This revealed three different growth phenotypes (Fig. 1B, middle): (i) 2885 mutants that grew only in the presence of *wt* prototrophs (wild-type phenotype; Dataset EV1), (ii) 137 mutants that grew irrespective of neighboring *wt* prototrophs (genes required for NCR; Dataset EV1), and (iii) 203 mutants that failed to grow either alone or in the presence of neighboring *wt* prototrophs (genes required for the avoidance of NCR; Dataset EV1). Thus, we have identified 203 genes that are necessary for *leu1-32* auxotrophs to respond to signals, sent out by *wt* prototrophs, that enable them to take up leucine from the environment (avoidance of NCR).

Although NSF has been identified as a diffusible signaling molecule that is sufficient to sustain the growth of leucine auxotrophs under high ammonium conditions (Sun et al, 2016), it remains unknown whether other signals are secreted that could function redundantly to NSF. To test how many of the 203 genes

required for the avoidance of NCR would be specifically required for NSF-mediated adaptive growth, we performed a second screen (Fig. 1B, bottom). For this, the 203 mutant strains were spotted on EMM(748 mM $NH_4Cl$)+Leu plates coated either with synthetic NSF (10(*R*)-hydroxy-8(*Z*)-octadecenoic acid) (Sun et al, 2016) or methanol as a mock control. The strains were also spotted on plates lacking ammonium (EMM(0 mM $NH_4Cl$) + Leu) to check the abilities of growth on EMM, and leucine uptake and utilization. As expected, none of the strains grew on high ammonium plates (EMM(748 mM $NH_4Cl$) + Leu). Of those, 29 mutants grew under neither condition tested, indicating that these genes are related to EMM growth or leucine uptake ("genes required for growth on EMM" in Fig. 1B). Growth on high ammonium plates was rescued for 57 mutants in the presence of NSF (NSF-independent adaptive growth in Fig. 1B). That is, these genes are not needed to respond to NSF. Yet, because they are needed for adaptive growth (1st screen), they are likely required to respond to a transmissible signal that is different from NSF (Sun et al, 2016; Chiu et al, 2022), or else they could alter the sensitivity of the strain to NSF. We speculate that these mutants are less sensitive to NSF for avoidance of NCR, and the amount of NSF provided by neighboring *wt* cells in the first screen was below the threshold, while the amount provided in the second screen surpassed this threshold, leading to the manifestation of adaptive growth. Although these factors may be partially involved in the NSF response, we did not include them in any further analyses in this study to keep the focus on the strongest hits. Finally, we identified 117 mutants that grew normally without ammonium, but not at high ammonium concentrations despite the addition of NSF. Thus, we have identified 117 genes that are required for NSF-mediated adaptive growth (referred to as "NSF-linked genes" listed in Datasets EV1 and EV3).

## Mitochondrial respiration counteracts NCR

Analyzing the list of 117 NSF-linked genes with AnGeLi (Bitton et al, 2015), we noticed the enrichment of genes encoding proteins that are related to mitochondrial respiration (Fig. 1C; Dataset EV2). Interestingly, 33 NSF-linked genes have been previously implicated in respiratory growth when glycerol serves as a carbon source (Malecki et al, 2016) (Fig. 1D; Dataset EV3). This suggests that NSF-mediated adaptive growth depends on mitochondrial respiration. To test this directly, we assessed NSF-mediated adaptive growth under standard NCR conditions (EMM(374 mM $NH_4Cl$) + Leu + Ade + Ura) alongside the addition of antimycin A, which is a potent mitochondrial electron transport inhibitor and can thus be used to block respiration (Heslot et al, 1970; Malecki et al, 2016). As shown in Fig. 1E, antimycin A blocked NSF-mediated adaptive growth in the presence of high ammonium concentrations. In the absence of ammonium, 0.4 μM antimycin A had no effect on cell growth. Vice versa, when we induced respiration by lowering the glucose concentration (0.08%) (Takeda et al, 2015), NSF even became dispensable for growth under high ammonium conditions (Fig. 1F). To check whether NSF treatment stimulates respiration, we measured oxygen consumption rates (OCR) after 8 h of NSF treatment, when no difference in growth (with or without NSF treatment) had yet been observed. Indeed, we observed an increased OCR upon NSF treatment (Fig. 1G). From these results, we conclude that mitochondrial respiration counteracts NCR, and that this is regulated by NSF signaling.

## Changes in gene expression upon exposure to NSF

To investigate global changes in gene expression in response to NSF exposure, we performed total RNA sequencing with cells that were grown in the presence of high ammonium concentrations and exposed to synthetic NSF for 2, 4, or 6 h. Already after 2 h exposure to NSF, we observed 99 and 75 genes that were significantly up- or downregulated, respectively (Fig. 2A; Dataset EV2). Of the 117 NSF-linked genes described above, only *hsr1* was differentially expressed upon NSF exposure. The expression of respiration-related genes that were revealed by the screen remained unchanged. This remained largely the same when cells were exposed to NSF for longer (Fig. EV1A,B). Although there is a possibility that differential expression of most NSF-linked genes may only become significant at later time points, or that changes might occur at the protein level rather than at the transcriptional level, we conclude that most of the NSF-linked genes identified in our screen appear not to be differentially expressed upon NSF treatment.

Notably, we observed a partial overlap between NSF-regulated genes and genes that were previously shown to be expressed differentially in cells that had been shifted to respiration by glycerol feeding (Malecki et al, 2016) (Fig. 2B). On a genome-wide level, differential gene expression patterns between NSF-treated cells (this study) and cells that had been fed with glycerol tended to correlate positively (Fig. EV1C). Interestingly, genes previously shown to be upregulated in respiratory conditions tended to be upregulated in response to NSF exposure as well. Likewise, genes downregulated in respiratory conditions appeared to be less expressed also in cells exposed to NSF (Figs. 2C and EV1D) (Malecki et al, 2016). These results are consistent with the above findings and reveal that NSF triggers a change from a fermentation- to a respiration-like gene expression program.

Gene Ontology (GO) enrichment analysis (Dataset EV2) of the 75 downregulated genes revealed an overrepresentation of GO terms related to flocculation or adhesion (Fig. 2D). The 99 upregulated genes were enriched for GO terms related to trehalose synthesis (Paulo et al, 2014). Coherent with the induction of a respiration-like gene expression program, the trehalose synthesis pathway is also upregulated under respiratory conditions (Malecki et al, 2016). Notably, it has been reported that trehalose functions as an antioxidant in the probiotic yeast *Saccharomyces boulardii* (Moon et al, 2020). This could indicate a potential role in counteracting reactive oxygen species (ROS) production associated with NSF-induced respiration. Enrichment of this GO term thus suggests that NSF could contribute to the induction of expression of antioxidants for ROS that is generated by the respiration pathway in mitochondria. We find it interesting that flocculation is enriched in down-regulated genes and speculate that this could serve to avoid cell-cell adhesion. This would be particularly pronounced at high cell density, when NSF would be most concentrated. Because cell-to-cell communication is often linked to morphological changes such as biofilm formation in bacteria (Miller and Bassler, 2001; Hammer and Bassler, 2003; Mukherjee and Bassler, 2019), mating, or filamentation in fungi (Chen et al, 2004; Hornby et al, 2001; Merlini et al, 2013; Ramage et al, 2002), it is possible that NSF-mediated cell-to-cell communication could be linked to morphological changes. Although the actual impact of NSF on flocculation has not been confirmed, it might be worth investigating this further.

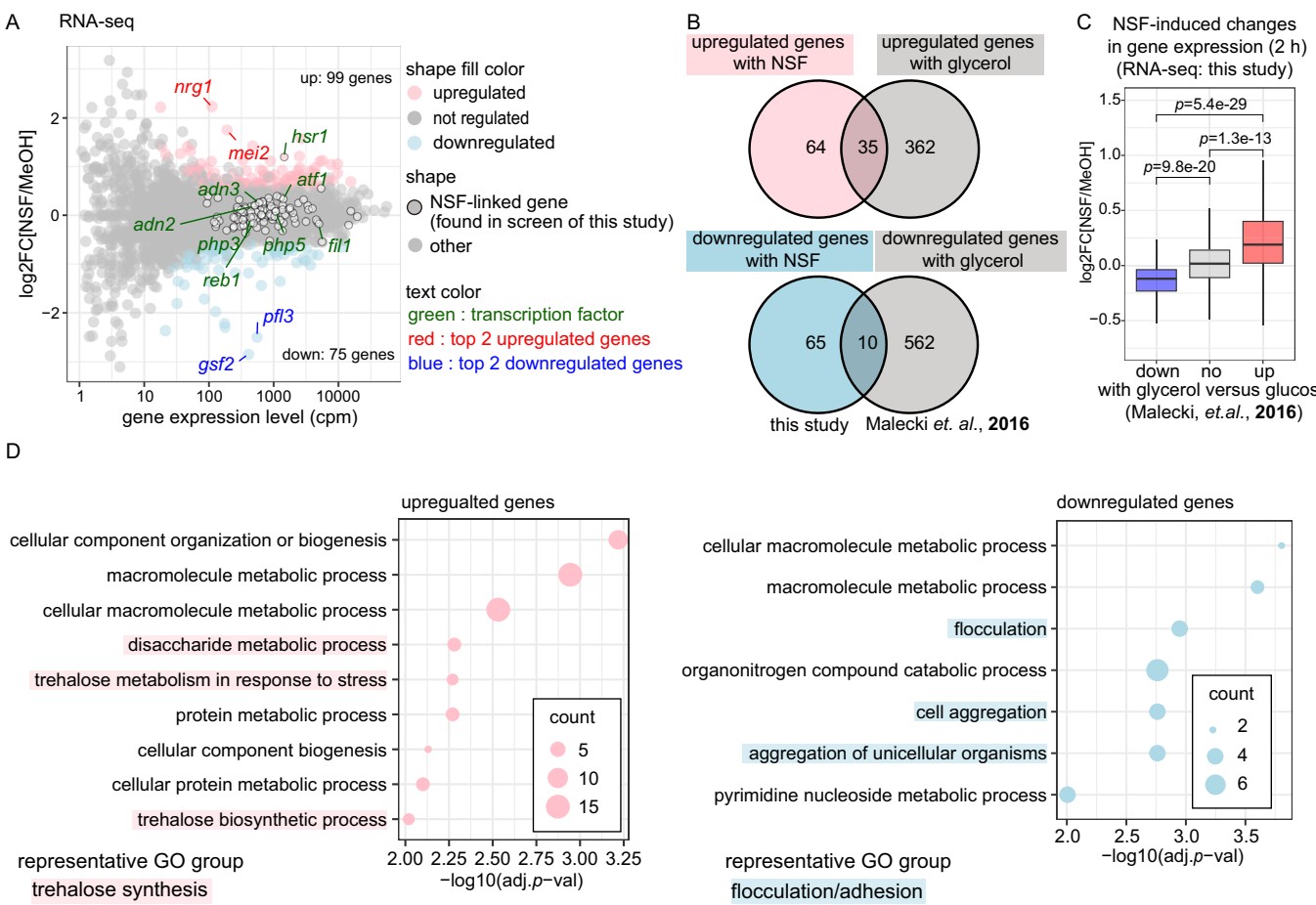

**Figure 2. NSF-induced changes in gene expression.**

(A) MA plot showing differential gene expression (log2FC) in cells treated with MeOH or NSF for 2 h at low cell density (OD 0.01) (y-axis). x-axes denote total transcript abundance in counts per million (cpm) in both conditions. p values were calculated using the Wald test and adjusted with the Benjamini and Hochberg (BH) method. Up- or down-regulated genes (FC >1.5 and adjusted p value <0.05) are highlighted in pink or blue, respectively. NSF-linked genes revealed by the genetic screen are marked with a black outline. The names of the top two upregulated and downregulated genes are written in red and blue, respectively. The green color was used to label the genes encoding transcription factors. (B) Venn diagram showing the overlap of NSF-regulated genes revealed by this study and glycerol-regulated genes that were previously shown (Malecki and Bähler, 2016). (C) Box plot showing logFC distribution of NSF-induced gene expression changes, grouped by gene expression changes induced by respiration (Malecki et al, 2016). p values were calculated using a two-sided t-test and were adjusted using the holm method. The center line inside the box represents the median of the data, bounds of the box correspond to the interquartile range with the bottom and top of the box indicating the first and third quartiles, and bounds of whiskers extend to the minimum and maximum values within the range of non-outlier data. Each condition was done in biological triplicates. (D) GO enrichment analysis of genes that are differentially expressed upon NSF exposure by AnGeLi (Bitton et al, 2015). AnGeLi applies the two-tailed Fisher's exact test and two-sided Wilcoxon rank-sum test as multiple statistical tests. GO term enrichment is shown on the x-axis (as −log10 adjusted p value) for all GO terms enriched (adj.p val <0.01) among upregulated (left) or downregulated (right) genes. Representative GO groups are highlighted with indicated color.

## The NSF-responsive gene 1

Of the genes with a gene expression level higher than 100 cpm (expression levels less prone to qRT-PCR measurement errors), *SPBPB2B2.01* and *mei2* were the two most highly upregulated genes, whereas *gsf2* and *pfl3* were the two most strongly downregulated genes (Fig. 2A). Further analyses revealed that expression of all four genes changed in an NSF dose-dependent manner (Fig. 3A). Notably, *SPBPB2B2.01* is an uncharacterized gene whose protein product has an inferred biological role in amino acid transmembrane transport (https://www.pombase.org/gene/SPBPB2B2.01). This is interesting in light of BCAA uptake regulation being a hallmark of the NCR pathway. Because *SPBPB2B2.01* is strongly induced by NSF, we suggest naming it

"NSF-responsive gene 1" (*nrg1*). To confirm the specificity of *nrg1* induction by NSF, we used oleic acid as a negative control for NSF. Oleic acid is a fatty acid with an identical 18-carbon chain and a chemical structure like NSF, but it does not induce adaptive growth (Sun et al, 2016). Even with three times the amount of NSF (150 ng/ml), oleic acid did not induce an increase or decrease in the expression of NSF-responsive genes (Fig. 3B).

We noted that the stimulatory effect of synthetic NSF on *nrg1* expression also occurred in prototrophs (Fig. 3C), was highest at low cell density (OD 0.01), and ceased at higher concentrations (OD 0.05, 0.1). Yet, expression of *nrg1* increased with increasing cell density (Fig. 3D). This might indicate that *S. pombe* cells produce and secrete NSF, which accumulates with increasing cell density, thereby negating any stimulatory effect of synthetic NSF.

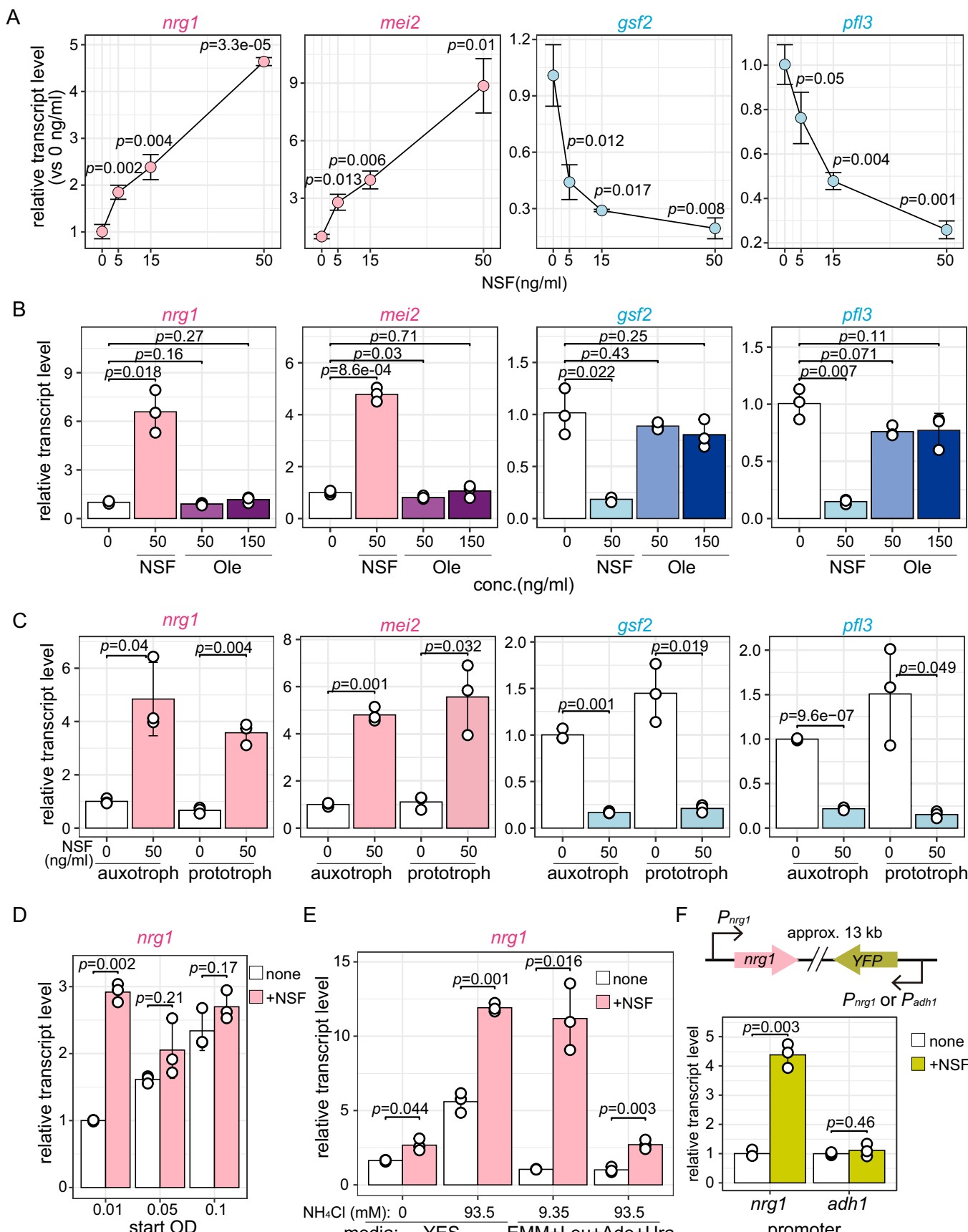

◀

**Figure 3.** Relative gene expression levels of the two most up- or downregulated genes.

(A) NSF dose-dependent changes in gene expression. Total mRNA was prepared from cells cultured in EMM(187 mM $NH_4Cl$) + Leu + Ade + Ura with increasing concentrations of NSF (0, 5, 15, 50 ng/mL) for 6 h at low cell density (OD 0.01). (B) Gene expression changes in response to NSF or oleic acid (Ole) treatment. Total mRNA was prepared from cells cultured in EMM(187 mM $NH_4Cl$) + Leu + Ade + Ura with NSF (50 ng/mL) or Ole (50, 100 ng/mL) for 6 h at low cell density (OD 0.01). (C) Gene expression changes in response to NSF in auxotrophs (ED668: *leu1-32 ade6-M216 ura4-D18*) and prototrophs (SPB5163). Total mRNA was prepared from cells cultured in EMM(187 mM $NH_4Cl$) + Leu + Ade + Ura with NSF (50 ng/mL) for 6 h at low cell density (OD 0.01). (D) *nrg1* expression in cultures with increasing cell density. Total mRNA was prepared from cells cultured in EMM(187 mM $NH_4Cl$) + Leu + Ade + Ura and harvested at different cell densities ($OD_{600nm}$ = 0.01, 0.05, 0.1). (E) *nrg1* expression in YES or EMM medium with various $NH_4Cl$ concentrations. Total mRNA was prepared from cells cultured in YES, YES + 93.5 mM $NH_4Cl$, EMM(9.35 mM $NH_4Cl$) + Leu + Ade + Ura, and EMM(187 mM $NH_4Cl$) + Leu + Ade + Ura at low cell density (OD 0.01). (F) Yellow fluorescent protein (YFP) gene expression under the control of *nrg1* or *adh1* promoters. The YFP reporter gene was integrated 13 kb from the endogenous *nrg1* locus. Total mRNA was prepared from cells cultured in EMM(187 mM $NH_4Cl$) + Leu + Ade + Ura for 6 h at low cell density (OD 0.01). The mean and standard deviation from three independent experiments is shown. *p* values were calculated using a two-sided *t*-test. Source data are available online for this figure.

Indeed, when grown at low cell density, *nrg1* expression was induced by NSF irrespective of the nitrogen concentration, or whether cells were grown in rich medium (YES) or synthetic defined minimal medium (EMM) (Fig. 3E). Hence, NSF-mediated *nrg1* induction is not limited to specific conditions. Finally, we constructed a strain harboring a YFP gene controlled by either the *nrg1* or *adh1* promoter (Fig. 3F). Exposure to synthetic NSF-induced YFP expression if driven by the *nrg1* but not the *adh1* promoter, which is known to be constitutively active in *S. pombe* (Russell and Hall, 1983). In conclusion, *nrg1* exemplifies an NSF-responsive gene that is controlled at the level of transcription.

## NSF treatment can affect transcription factor occupancy in NSF-responsive genes

Coherent with the above finding that NSF triggers changes in gene expression at the level of transcription, our genetic screen revealed eight transcription factor (TF) genes that are required for NSF-mediated adaptive growth. Specifically, these genes encode the TFs Atf1, Adn2, Adn3, Fil1, Hsr1, Php3, Php5, and Reb1 (Dataset EV3). Consistent with the potential role of Atf1 in activating transcription of NSF-responsive genes, we found the GO term "Atf1 activated" enriched among the group of genes that we found to be upregulated in response to NSF treatment (Dataset EV2). Vice versa, Adn2 and Adn3 have previously been linked to flocculation and adhesion (Kwon et al, 2012), two other GO terms that we found enriched among the genes that are downregulated upon exposure to NSF (Dataset EV2). As mentioned above, *hsr1* was the only gene required for NSF-mediated adaptive growth that was also expressed more upon NSF treatment (Fig. 2A). That is, cellular abundance of the TFs Atf1, Adn2, Adn3, Fil1, Php3, Php5, and Reb1 remains unchanged, raising the question how they contribute to changing the transcriptional program. One possibility is that they remain bound to their target genes but become activated or deactivated by NSF directly, or a posttranslational modification, such as phosphorylation, which is known to regulate the activity of Atf1 in response to oxidative or osmotic stress (Lawrence et al, 2007, 2009a). Alternatively, NSF exposure might strengthen or weaken the binding to their target genes or redirect binding to other genes. To explore this, we tagged Adn2, Atf1, Fil1, Hsr1, Php3, and Reb1 with a 3xFLAG tag at their C-termini and performed chromatin immunoprecipitation coupled to next-generation sequencing (ChIP-seq) with and without NSF treatment for 2 h under NCR conditions. Previously determined target genes of these TFs were significantly enriched over input samples in our dataset, demonstrating that the experiment has worked (Fig. EV2).

Globally, TF binding patterns were not grossly altered by NSF treatment. Yet, at a few specific gene promoters, we observed a modest increase or decrease in TF enrichment (Fig. 4A). For Hsr1, and to some extent for Adn2, Php3, and Atf1, these differences in TF occupancy were positively correlated with target gene expression changes. That is, individual genes that were upregulated by NSF tended to be more strongly bound by the TFs, whereas downregulated genes were less occupied by the respective TFs (Fig. 4B). We chose the *nrg1*, *mei2*, *gsf2*, and *pfl3* genes to exemplify this: increased TF occupancy of Hsr1 at the *mei2* promoter correlated with increased *mei2* mRNA levels, whereas decreased TF occupancy of Hsr1, Adn2, and Php3 at the *gsf2* and *pfl3* promoters correlated with reduced *gsf2* and *pfl3* mRNA levels (Fig. 4C). Consistent with the latter TFs stimulating transcription of *gsf2* and *pfl3*, mRNA levels were strongly reduced in the respective TF knock-out strains (Fig. 4D). At the *nrg1* promoter, Hsr1 and Php3 occupancy remained unchanged (Fig. 4C). Yet, *nrg1* expression was reduced in *hsr1Δ* and *php3Δ* cells, and it was hardly induced by NSF, underscoring the regulatory role of these TFs (Fig. 4D).

Globally, TF binding correlating with gene expression was particularly pronounced for Hsr1 (Fig. 4B), which indicates that NSF treatment does not only stimulate *hsr1* expression but also increases as well as reduces Hsr1 binding at specific genes, while Hsr1 occupancy on other genes, like *nrg1*, remains unchanged. For Adn2 and Atf1, NSF mainly caused reduced binding at their target genes, which correlated with reduced target gene expression. As mentioned earlier, Atf1's positive effect on transcription could be activated by phosphorylation, potentially explaining why NSF mainly causes reduced Atf1 binding at downregulated genes but does not affect Atf1 binding at upregulated genes. For Php3, NSF mainly caused increased binding at its target genes, which correlated with upregulated target gene expression. Occupancy of the other TFs at NSF-responsive genes remained largely the same (Figs. 4A,B and EV2).

## Ayr1 may metabolize NSF

The foregoing results establish NSF-mediated adaptive growth as a paradigm to investigate how external factors, such as chemicals or metabolites in a cell's environment can lead to transcriptional network changes. To get first insights into the mode of action of NSF, we functionalized NSF with an alkyne tag to generate a probe that can be used for click chemistry, fluorescence microscopy, and affinity purification of putative proteins that might physically interact with NSF (AlkNSF, Fig. 5A, Matoba et al, 2024). Alkyne probes are commonly used in chemical biology because structures

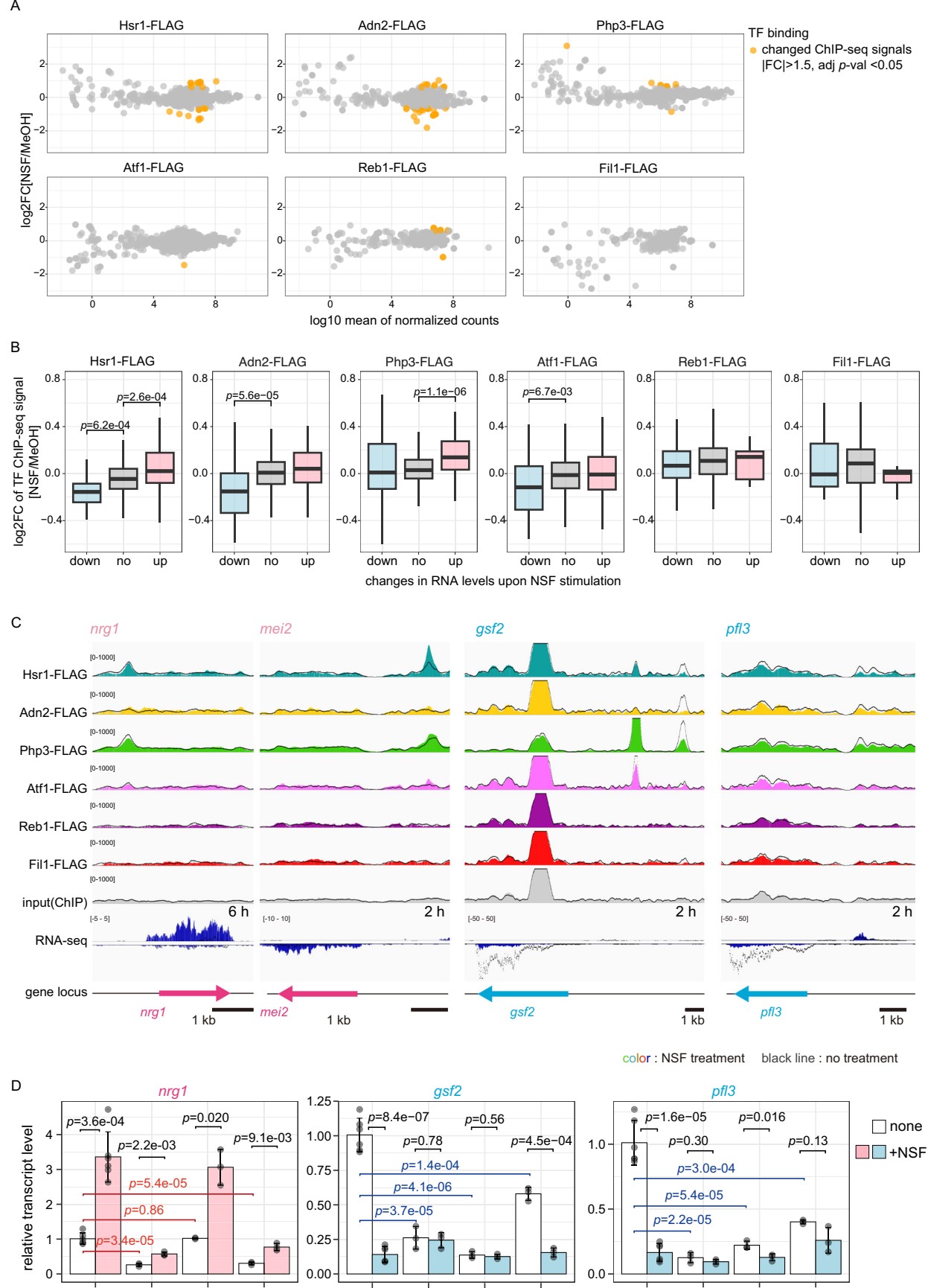

**Figure 4. NSF can affect occupancy of NSF-linked TFs on NSF-responsive genes.**

(A) Changes in the ChIP-seq signal of the respective transcription factors in cells treated with MeOH or NSF for 2 h at low cell density (OD 0.01) are represented as MA plots. Log2 fold changes are shown on the y-axis. $p$ values were calculated using the wald test and adjusted with the BH method. The x-axis shows the average normalized read counts (log10) of samples with and without NSF treatment. Significantly changing ChIP-seq signals (FC >|1.5| and adjusted $p$ value <0.05) are highlighted in orange. (B) Box plots showing log2FC distributions of the respective TF ChIP-seq signals, grouped by NSF-induced gene expression changes, shown in Fig. 2A (downregulated, no change, upregulation). $p$ values were calculated using a two-sided $t$-test and adjusted using the Holm method. The center line inside the box represents the median of the data, bounds of the box correspond to the interquartile range with the bottom and top of the box indicating the first and third quartiles, and bounds of whiskers extend to the minimum and maximum values within the range of non-outlier data. Each condition was done in technical triplicates. (C) ChIP enrichments of NSF-linked TFs were revealed by our screen on the promoters of three representative genes that are either up- or down-regulated by NSF treatment (RNA-seq). Black lines denote ChIP enrichment in untreated cells. Colored areas represent ChIP enrichments upon NSF treatment. The RNA-seq tracks were derived from RNA-seq data of cells under NCR conditions with or without NSF for 2 h for $mei2$, $gsf2$, and $pfl3$, and 6 h for $nrg1$ (for which transcript levels accumulate most prominently after 2 h of treatment; see Figs. 2A and EV1). (D) Transcript levels of NSF-responsive genes in $hsr1\Delta$, $adn2\Delta$, or $php3\Delta$ cells. Total mRNA was prepared from cells cultured in EMM(187 mM $NH_4Cl$) + Leu + Ade + Ura. Cells were exposed to 50 ng/mL NSF for 6 h at low cell density (OD 0.01). The mean and standard deviation from three independent experiments are shown. $p$ values were calculated using a two-sided $t$-test. Source data are available online for this figure.

and physicochemical properties of small molecules are minimally affected (Wright and Sieber, 2016). Compared to the minimum effective concentration (MEC) that was previously determined for NSF (12 ng/ml) (Sun et al, 2016), we had to use 30-fold higher concentrations of AlkNSF to sustain growth (Fig. EV3A). Yet, although the MEC of AlkNSF was higher than that of NSF, AlkNSF promoted adaptive growth on EMM(374 mM $NH_4Cl$) + Leu plates (Fig. 5B) and induced $nrg1$ expression in a dose-dependent manner (Fig. 5C). Furthermore, we observed cellular uptake of azide fluor 488-labeled AlkNSF by fluorescence microscopy (Fig. EV3B). These results indicated that AlkNSF could indeed be successfully employed as a probe in a chemical biology experiment. Therefore, we coupled AlkNSF to azide beads (FG beads®) and incubated these with *S. pombe* whole-cell lysates to affinity purify putative NSF interacting proteins. As a control, we preincubated a fraction of the lysate with non-alkylated NSF, which is expected to compete off AlkNSF. Following separation and washing of the AlkNSF-beads, we subjected the immobilized samples to mass spectrometry. This revealed Ayr1 as the sole protein that was copurifying with AlkNSF, and that was competed away with NSF significantly (Fig. 5D). Interestingly, $ayr1$ was not among the genes that our screen has identified to be required for adaptive growth (Dataset EV1), which we confirmed in a newly generated $ayr1\Delta$ strain (Fig. EV3C). Consistent with this phenotype, NSF-mediated gene expression changes were not abrogated in $ayr1\Delta$ cells. In contrast, the stimulatory effect of NSF on $nrg1$ or $mei2$ expression was enhanced in the absence of Ayr1 (Fig. 5E), suggesting that Ayr1 might function as a negative regulator of NSF-linked changes in gene expression. Because Ayr1 is annotated as a 1-acyldihydroxyacetone phosphate reductase that has been connected to lipid metabolism in yeast (Athenstaedt and Daum, 2000; Ploier et al, 2013), it is tempting to speculate that Ayr1 dampens adaptive responses by metabolizing NSF. Consistent with this, we observed that over-expression of Ayr1 (Fig. EV3D) affected NSF-mediated adaptive growth (Fig. 5F) and dampened NSF-mediated induction of $nrg1$ and $mei2$ expression (Fig. 5G).

## NSF physically and functionally interacts with Hmt2

Because Ayr1 could avert the interaction of AlkNSF with other true interaction partners, we repeated the experiment described above with whole-cell lysates generated from an $ayr1\Delta$ strain. Indeed, this experiment revealed several additional proteins that were significantly enriched compared to the control sample that contained competitive

NSF (Fig. 5H). Yet, a potential caveat of this result is the hydrophobicity of AlkNSF, which could result in false positive NSF-protein interactions upon prolonged incubation with cell lysates. Therefore, we included an oleic acid alkyne probe (AlkOle), which is similar to AlkNSF but has no biological activity (Fig. 5B), and repeated the experiment with *wt* cell lysates to assess which of the above proteins would specifically associate with the AlkNSF but not the AlkOle probe. Reassuringly, Ayr1 was again specifically interacting with AlkNSF. The other proteins that were significantly enriched in the AlkNSF sample were Hmt2 and Gst3 (Figs. 5I and EV3E). $gst3$ was not required for NSF-mediated adaptive growth, neither in our screen nor in growth assays performed with newly generated $gst3\Delta$ cells (Fig. EV3F). $hmt2$ was identified by our genetic screen to be required for growth on EMM media (Dataset EV1; Fig. EV3F), which can be explained by the cysteine auxotrophy of $hmt2\Delta$ cells (Pluskal et al, 2016). Because our screen was conducted in EMM, $hmt2$ was thus not revealed as a gene necessary for NSF-mediated adaptive growth. Therefore, we reassessed the NSF-mediated adaptive growth of $hmt2\Delta$ and $gst3\Delta$ cells in YES medium supplemented with 93.5 mM $NH_4Cl$. Whereas wild-type and $gst3\Delta$ cells grew in the presence of NSF but not in its absence, $hmt2\Delta$ cells did not show NSF-mediated adaptive growth (Fig. 5J). Likewise, $hmt2\Delta$ cells did not show NSF-mediated adaptive growth on EMM(374 mM $NH_4Cl$)+Leu+Ade+Ura supplemented with cysteine (Fig. EV3G). Notably, these effects were not observed with $hmt2\Delta$ prototrophs. Because leucine auxotrophy induced the phenotype, the growth defect in $hmt2\Delta$ cells under NCR condition is likely caused by defective leucine uptake (Fig. EV3H,I). Consistent with this phenotype, neither $nrg1$ nor $mei2$ expression was induced by NSF in the absence of Hmt2 (Fig. 5K).

These results identify Hmt2 as a direct target of NSF that is necessary for the activation of NSF-responsive genes. Hmt2 is localized at the inner membrane of mitochondria where it functions as a sulfide:quinone oxidoreductase (SQR) by which hydrogen sulfide acts as an electron donor to the electron transfer chain (ETC) via reduction of quinone to a hydroquinone (Zhang et al, 2021; Weghe and Ow, 1999). Notably, Hmt2 is required for respiratory growth (Malecki et al, 2016), which is coherent with our finding that mitochondrial respiration enables evasion of NCR. Thus, the identification of Hmt2 as a direct functionally relevant target of NSF provides a first insight into how the NSF signal is received by the recipient cells. How this affects mitochondrial respiration and how this leads to changes in the cell's transcriptional program are exciting questions that should become the subject of future investigations.

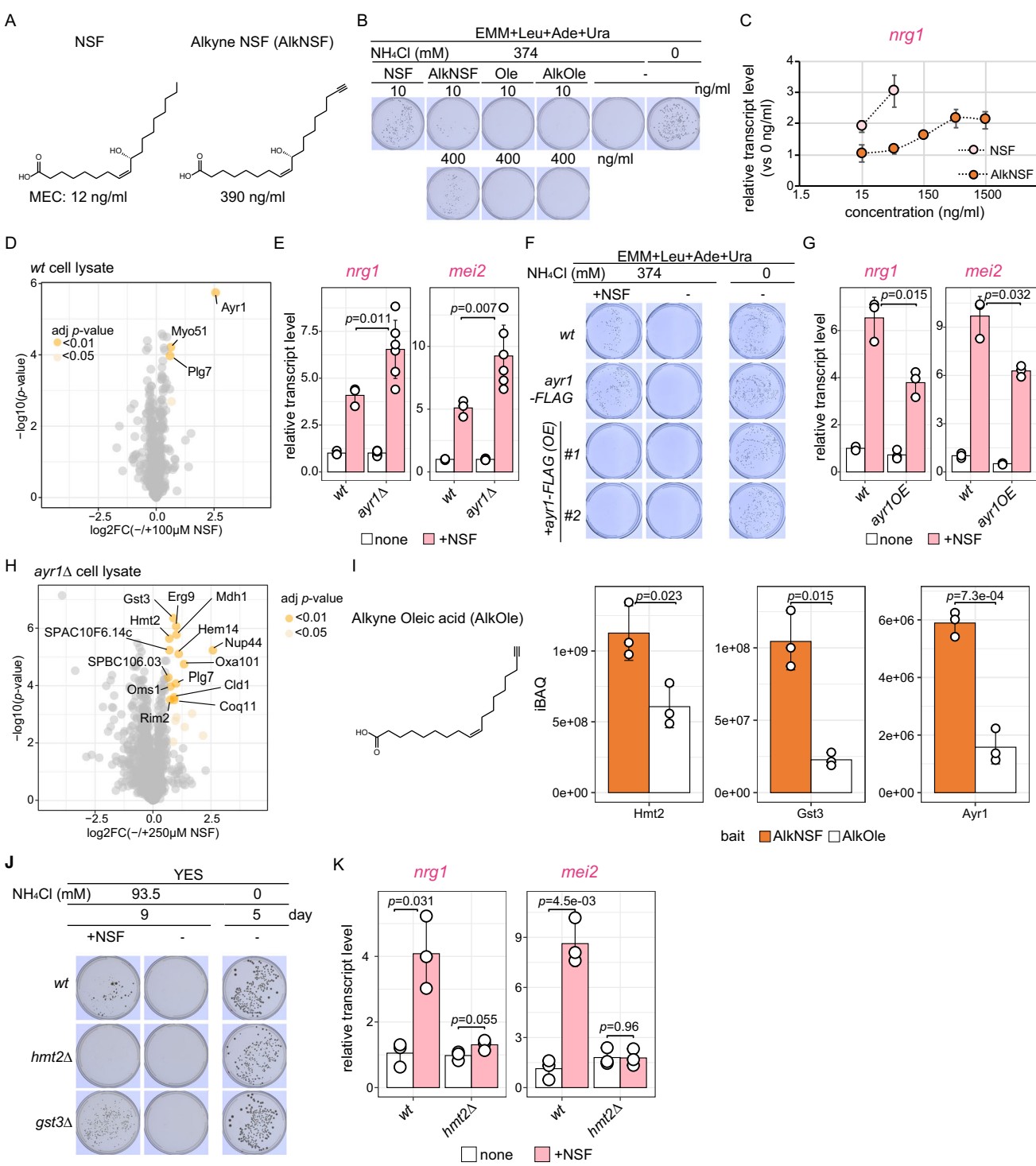

## Discussion

Cell-to-cell communication is a widely observed phenomenon among microbes. It is used to react to changes in the environment and to coordinate such responses at the population level to keep the organism thriving. In this study, we have focused on cell-to-cell communication mediated by NSF, a diffusible oxylipin that is produced and secreted by *S. pombe* cells to their extracellular milieu. A characteristic effect of NSF is that it enables the uptake of BCAA, which is essential for cells that cannot produce their own BCAA. Because excess ammonium supplementation in the media leads to the inhibition of BCAA uptake, a phenomenon known as nitrogen catabolite repression or NCR (Mitsuzawa, 2006; Magasanik and Kaiser, 2002; Ljungdahl and Daignan-Fornier, 2012), it is

◄ **Figure 5. A chemical biology approach identifies potential NSF target proteins.**

(A) Chemical structures of NSF and alkyne-adducted NSF. 10(*R*)-hydroxy-8(*Z*)-octadecenoic acid (NSF) on the left and alkyne-adducted NSF (AlkNSF) on the right. The minimum effective concentration (MEC) was determined by cellular growth assays (see Fig. EV3A). (B) Bioactivity assessments for AlkNSF, Ole, and AlkOle. Leu-auxotrophic cells (ED668) were spotted onto EMM(374 mM or 0 mM NH$_4$Cl) + Leu + Ade + Ura containing either NSF (10 ng/mL), alkyne-adducted NSF (AlkNSF) (10 or 400 ng/mL), Ole (10 or 400 ng/mL), or alkyne-adducted oleic acid (AlkOle) (10 or 400 ng/mL). Plates were incubated at 30 °C. (C) Stimulation of *nrg1* expression by increasing concentrations of NSF (15, 50 ng/mL, light pink) or AlkNSF (15, 50, 150, 500, 1500 ng/mL, orange) was measured from cells that were grown in EMM(187 mM NH$_4$Cl) + Leu + Ade + Ura for 6 h at low cell density (OD 0.01). The mean and standard deviation from three independent experiments is shown. (D) Whole-cell lysates were incubated with AlkNSF in the presence or absence of NSF that we used as a competitor. Proteins copurifying with the AlkNSF probe were identified by mass spectrometry. The data were displayed with enrichment values on the x-axis (AlkNSF versus AlkNSF + NSF) and the *p* values (moderated *t*-statistic) on the y-axis. Significantly enriched proteins are labeled by their name. The significance level is indicated by orange (FC >1.5 and adjusted *p* value <0.01) or light orange color (FC >1.5 and adjusted *p* value <0.05). The statistical test was based on the t-statistic returned from limma. (E) Transcript levels of NSF-responsive genes in *wt* (ED668) and *ayr1Δ* cells. Total mRNA was prepared from cells cultured in EMM(187 mM NH$_4$Cl) + Leu + Ade + Ura. Cells were exposed to 50 ng/mL NSF for 6 h at low cell density (OD 0.01). (F) NSF-mediated adaptive growth in *wt* (ED668), *ayr1-FLAG* (SPB5122), and *ayr1-FLAG* overexpression (OE) (SPB5227 and SPB5228) cells in EMM media. (G) Transcript levels of NSF-responsive genes in *wt* (ED668) and *ayr1-FLAG* overexpression (*ayr1OE*) cells. Total mRNA was prepared from cells cultured in EMM(187 mM NH$_4$Cl) + Leu + Ade + Ura. Cells were exposed to 50 ng/mL NSF for 6 h at low cell density (OD 0.01). (H) *ayr1Δ* whole-cell lysates were incubated with AlkNSF. Proteins copurifying with the AlkNSF probe were identified by mass spectrometry. The data were displayed with enrichment values on the x-axis (AlkNSF versus AlkNSF + NSF) and the BH-adjusted *p* values (moderated t-statistic) on the y-axis. Significantly enriched proteins are labeled by their name, or the systematic gene ID. The significance level is indicated by orange (FC >1.5 and adjusted *p* value <0.01) or light orange color (FC >1.5 and adjusted *p* value <0.05). The statistical test was based on the t-statistic returned from limma. (I) iBAQ values of proteins copurifying with AlkNSF or AlkOle. Whole-cell lysates were prepared from wild-type cells. The mean and standard deviation from technical triplicates is shown. *p* values were calculated using a two-sided *t*-test. (J) NSF-mediated adaptive growth of *wt* (ED668), *hmt2Δ* (SPB5150), or *gst3Δ* (SPB5158) cells in YES medium. (K) Transcript levels of NSF-responsive genes in *hmt2Δ* cells. Total mRNA was prepared from cells cultured in EMM(187 mM NH$_4$Cl) + Leu + Ade + Ura. Cells were exposed to 50 ng/mL NSF for 6 h at low cell density (OD 0.01). The mean and standard deviation from three independent experiments is shown. *p* values were calculated using a two-sided *t*-test. Source data are available online for this figure.

generally assumed that NSF functions to control the usage of different nitrogen sources (Takahashi et al, 2012). However, the pathways and mechanisms by which NSF functions and whether these are restricted to nitrogen metabolism have remained elusive. We have performed a genetic screen with a non-essential gene deletion library to identify factors that are required for NSF-mediated adaptive growth at high ammonium concentrations. Essentially, the screen was designed such that every mutation that disables the uptake of a BCAA would be identified. This has revealed 117 mutants that grew normally without ammonium, but not at high ammonium concentrations despite the addition of synthetic NSF and the presence of exogenous BCAA. Interestingly, many of the genes identified in this screen are required for mitochondrial respiration (Fig. 1C), suggesting that NSF-mediated adaptation to high ammonium concentrations depends on the respiratory capacity of the cell. This hypothesis is further supported by our observation that inhibition of mitochondrial electron transport by antimycin A stops cells from growing in the presence of high ammonium concentrations. Conversely, indirect induction of respiration by glucose limitation makes NSF dispensable for growth. Moreover, we found that NSF triggers a respiration-like gene expression pattern, and that NSF physically interacts with the mitochondrial sulfide:quinone oxidoreductase Hmt2. Therefore, we propose a model in which NSF activates mitochondrial respiration, which eventually leads to global changes in gene expression, the evasion of NCR, and the uptake of BCAA (Fig. EV4). The latter is reminiscent of previous work conducted in budding yeast, which revealed that the induction of respiration by glucose de-repression demands and increases the uptake of leucine for intermediates of the TCA cycle (Hothersall and Ahmed, 2013). In one of our previous studies, we found that the addition of ferrichrome results in the recovery of growth under high ammonium conditions, and we speculated that this could be linked to increased mitochondrial activity (Chiu et al, 2022). From the current study, a model could be proposed in which ferrichrome enhances BCAA uptake under high ammonium conditions through increased mitochondrial activity.

We find it interesting that NSF triggers a change from a fermentation- to a respiration-like gene expression program without any change in the carbon source. This raises the question of how *S. pombe* could benefit from respiration in the presence of glucose, which is well known to promote rapid proliferation by fermentation (Pfeiffer and Morley, 2014; Hagman and Piškur, 2015). Importantly, the concentration of NSF in the milieu is positively correlated with cell density. That is, the fewer cells, the less NSF will be secreted. Therefore, the switch from fermentation to respiration would be expected to occur only when cells become denser and the concentration of NSF increases. Indeed, *nrg1* expression increased with higher cell density without the addition of extra NSF (Fig. 3D). Moreover, the stimulatory effect of synthetic NSF was highest at low cell density and ceased when the population grew denser. This raises the possibility that NSF may function as a rheostat to prepare a population of *S. pombe* cells for a foreseeable shortage of glucose when they grow exponentially. Thereby, reaching a certain NSF concentration would cause a switch to respiration, also known as diauxic shift (Brauer et al, 2005; Bartolomeo et al, 2020), preparing the cells to utilize other carbon sources such as ethanol, because glucose will eventually become limiting. Whereas diauxic shifts are well-established, to our knowledge, this phenomenon has not been linked to cell-to-cell communication. Because NSF is an intraspecies-specific signal (Yashiroda and Yoshida, 2019; Takahashi et al, 2012; Sun et al, 2016), the NSF-mediated shift from fermentation to respiration would exemplify a "social interaction" between cells of the same species, preparing them for an imminent shortage of their preferred carbon source. This would increase the population's fitness and thus confer *S. pombe* a competitive advantage in such an environment. In that sense, NSF could be considered a mediator not only for revoking NCR but also for inducing respiration in carbon metabolism, preceding the release of CCR.

An important question that will need to be addressed in the future is how NSF exposure triggers the observed change in *S. pombe*'s gene expression program. Because our chemical biology approach has not revealed transcription factors or chromatin

regulators as direct targets of NSF, it is unlikely signaling directly to the nucleus. Rather, the identification of Hmt2 as a physical interaction partner of NSF suggests that the transcriptional changes observed in the nucleus are a consequence of the altered metabolic activity of mitochondria. Hmt2 is a sulfide:quinone oxidoreductase (SQR), which couples sulfide oxidation to coenzyme $Q_{10}$ reduction in the ETC. This reaction generates highly reactive sulfur species (RSS), such as glutathione persulfide, which can be further oxidized or can modify cysteine residues in proteins by persulfidation (Filipovic et al, 2018; Cuevasanta et al, 2017; Mishanina et al, 2015; Mustafa et al, 2009a, 2009b). Interestingly, there are several reports that persulfidation of transcription factors could regulate their activities (Yao et al, 2023; Tian et al, 2021; Shimizu et al, 2023). Thus, persulfide signaling is an attractive hypothesis for how NSF signaling could be relayed to the cell nucleus that will be worthwhile testing.

# Methods

## Yeast strains and growth media

*S. pombe* strains were generated following a PCR-based protocol (Bähler et al, 1998) and strains were validated by colony PCR. For a list of strains generated in this study, see Dataset EV4. Cells were grown in rich yeast extract (YE) medium with 3% glucose, YE with 2 mM each adenine and uracil (YES), or in minimal (EMM) medium with 2% glucose (EMM containing 2% glucose from FORMEDIUM™) and 0.5% (93.5 mM) ammonium chloride ($NH_4Cl$). Appropriate leucine, adenine, uracil, and glutamate (2 mM) were added to the EMM medium. For nitrogen catabolite repression (NCR) condition, YES and EMM were supplemented with extra $NH_4Cl$ (for YES; final concentration 93.5 mM. for EMM; final concentration 187 mM for liquid culture, 374 mM for normal NCR condition on solid culture, and 748 mM for severe NCR condition on solid culture) and for non-NCR condition, EMM without ammonium sulfate (0 mM $NH_4Cl$) with 2 mM Leu as a sole nitrogen source. For supplementation of NSF, 100 μL of 10 μg/mL synthesized NSF (10(*R*)-hydroxy-8(*Z*)-octadecenoic acid) (Sun et al, 2016) dissolved in 50% methanol was added onto 20 mL agar media (final concentration 50 ng/mL). Where indicated, media were supplemented with antimycin A (0.4 μM) for inhibition of respiration or with lower glucose (0.08%) for induction of respiration. The lower glucose EMM media was prepared using glucose-free EMM (MP Biomedical™). For cell mating, a sporulation agar (SPA) medium was used.

## Plasmids

Plasmids were cloned by standard molecular biology techniques. For a list of plasmids generated in this study, see Dataset EV5.

## Genetic screening for adaptive growth

To enable selection for the $h^-$ mating type in the Leu-auxotrophic knock-out library, an $h^-$ strain having the drug-resistant markers at the *mat1* locus was generated. The kanamycin resistance (*kanMX*) marker was inserted downstream of the *mat1-Mc* gene, which was subsequently replaced with the nourseothricin resistance marker (*natMX*) by the marker switch technique. First, two homology arms

were amplified from genomic DNA by PCR using two primer sets: insert-kanR-F1 and insert-kanR-F2, and insert-kanR-R1 and insert-kanR-R2. The purified PCR products were then used for assembling the disruption fragment by amplifying the *kanMX* cassette by PCR. The resultant PCR products were introduced into the JY265 strain. The strain in which the *kanMX* marker was replaced by the *natMX* marker were generated as previously described (Sato et al, 2005). We prepared the PCR product using MS-TEP and MS-TET as primers and pCR2.1-nat as a template. The Leu-auxotrophic knock-out library (3225 mutants) was made from the BIONEER haploid deletion mutant library v5.0 (3420 mutants) by mating with *leu1-32* auxotroph on SPA + Ade + Ura + Leu (2 mM each) medium. For selection after mating, mutants were grown on SD + Leu media, twice on YE + G418, and nourseothricin, twice. For the first screening, mutants are spotted onto 48-well plate solid EMM (748 mM $NH_4Cl$) + Leu (2 mM) media with or without a prototroph spotted next to the mutant spot. Plates were incubated at 30 °C, and images were acquired after 5 or 6 days. For the second screening, mutants were spotted onto 48-well plate solid EMM (0 mM or 748 mM $NH_4Cl$) + Leu media with NSF or methanol. Plates were incubated at 30 °C, and images were acquired after 5 or 6 days. GO category enrichments in each mutant cluster were calculated using the AnGeLi web tool (Bitton et al, 2015).

## Oxygen consumption rate measurement

The respiratory capacity was assessed using the Seahorse XF HS mini analyzer (Agilent Technologies). Seahorse XFp cell culture miniplate was coated with 50 μg/mL poly-lysine (50 μL each well) for 30 min at room temperature and then aspirated, followed by air drying at 4 °C overnight. The Seahorse XFp extracellular flux cartridge was hydrated with sterile water and incubated overnight at 30 °C and Seahorse XF calibrant solution (Agilent Technologies). After this, the water was removed, then added the calibrant solution to the sensor cartridge and incubated at 30 °C for 1 h before measurement.

Yeast cultures were grown in YES media at 30 °C for 16 h, followed by subculturing in EMM(561 mM $NH_4Cl$) + Leu + Ade + Ura media with or without NSF (50 ng/mL) for 30 °C for 8 h, start $OD_{600nm}$ at 0.01. For measurement, all cultures were diluted in EMM(561 mM $NH_4Cl$) + Leu + Ade + Ura to seed an $OD_{600nm}$ at 0.01 into a poly-lysine coated Seahorse XFp cell culture miniplate in 50 μL of Seahorse XF assay media (Agilent Technologies). For background measurement, two wells containing only assay media were included. The loaded plate was centrifuged at 300 × *g* for 3 min at room temperature. After centrifugation, the volume of the media was made up to 150 μL, and the loaded plate was incubated for 30 min at 30 °C to facilitate the transition of the plate into the Seahorse machine's temperature. Measurements by Seahorse were performed for 10 cycles.

## RNA isolation and cDNA synthesis

For RNA-seq, *leu1-32* auxotrophic (SPB0230) cells were grown in YES media at 30 °C for 16 h, followed by subculturing in EMM(187 mM $NH_4Cl$) + Leu + Ade + Ura media with or without NSF (50 ng/mL) for 30 °C for 2, 4, and 6 h, start $OD_{600nm}$ at 0.01. Each condition was done in biological triplicates. Total RNA was isolated using the MasterPure Yeast RNA Purification Kit

(Epicentre). cDNA was synthesized using the PrimeScript RT Master Mix (Takara).

## Total RNA sequencing and analysis

Total RNA libraries were prepared with TruSeq Stranded Total RNA kit (Illumina) according to the manufacturer's instructions and sequenced with an Illumina HiSeq2500 (50 bp single-end). RNA-seq reads were aligned to the *S. pombe* genome (ASM294 version 2.24) using STAR (Dobin et al, 2013) (version 2.7.3a) (STAR—runMode alignReads—outFilterType BySJout—outFilter-MultimapNmax 100—outFilterMismatchNoverLmax 0.05—out-SAMmultNmax 1—outMultimapperOrder Random—outSAMtype BAM SortedByCoordinate—outSAMattributes NH HI NM MD AS nM—outSAMunmapped Within). The reads per gene were counted with featureCounts (Liao et al, 2014) of uniquely mapping reads only (useMetaFeatures = TRUE, allowMultiOverlap = FALSE, minOverlap = 5, countMultiMappingReads = FALSE, fraction = FALSE, minMQS = 255, strandSpecific = 2, nthreads = 20, verbose = FALSE, isPairedEnd = FALSE). An external feature annotation file for *S. pombe* was used based on a GFF3 file from PomBase (Lock et al, 2018) that was converted to a GTF file with rtracklayer (Lawrence et al, 2009b). Differential gene expression analysis was performed using DESeq2 (Love et al, 2014). Each MA plot depicts log2 fold changes of cells with NSF treatment against cells without NSF treatment. For GO term enrichment analysis, we downloaded GO term annotation for *S. pombe* genes from PomBase and used the GO.db R package to build a GO annotation map for *S. pombe*. Then, we used the enricher function from the ClusterProfiler package to test for enriched GO terms in up- or down-regulated genes compared to all tested genes. Total RNA sequencing data have been deposited at the NCBI Gene Expression Omnibus (GEO) database and are accessible through GEO series number GSE250095.

## RT–PCR

PCR on cDNA was performed using the fast-cycling PCR kit (Qiagen). Primer sequences are listed in Dataset EV6.

## Chromatin immunoprecipitation (ChIP) and ChIP-sequencing

Cells were grown in YES media at 30 °C for 16 h, followed by subculturing in EMM(187 mM $NH_4Cl$) + Leu + Ade + Ura media with or without NSF (50 ng/mL) for 30 °C for 2 h, start $OD_{600nm}$ at 0.01. Each condition was done in technical triplicates. ChIP experiments were performed as described previously (Kuzdere et al, 2023) with 2.5 µg anti-FLAG M2 antibodies (Sigma). For input samples, sonicated but not immunoprecipitated lysates were used. ChIP-sequencing libraries were generated using the NEBNext Ultra II DNA Library Prep Kit (New England Biolabs) and sequenced with an Illumina HiSeq2500 (50 bp single-end). Raw data were demultiplexed, converted to fastq format using bcl2fastq2 (v1.17), and mapped using STAR (Genome_build: Spombe.ASM294v2.24). For bigwig track generation by bedtools (v2.26.0) and bed-GraphToBigWig (from UCSC binary utilities), nonaligned reads were discarded and read coverage was normalized to 1 million genome mapping reads (RPM). Peak finding was done using MACS2 (Gaspar, 2018) and peaks with a score above 100 were used. Differential binding was calculated using Limma–Voom and the total number of mapped reads as library size. To find the corresponding gene promoters under TF binding peaks, we used the R package GenomicFeatures and the "nearest" function of GenomicRanges. *S. pombe* genomic range file was downloaded from PomBase (Lock et al, 2018), and non-coding RNA genes were removed from the list. And then, genomic ranges of protein coding gene were changed to promoter range (200 bp upstream from transcription start site).

## Microscopic observation of AlkNSF with click chemistry

Cells were inoculated into EMM (187 mM $NH_4Cl$) + Leu + Ade + Ura, with 5 µM AlkNSF (Matoba et al, 2024) adjusted to $OD_{600nm}$ at 0.01. The culture was incubated at 30 °C for 4 h. Subsequently, cells were harvested and fixed in 70% ethanol. After fixation, cells were washed twice with TBS buffer, followed by incubation in a solution containing 20 µM Azide fluor (Sigma), 2 mM $CuSO_4$ (Sigma), and 10 mM ascorbate in TBS buffer in the dark for 30 min. Following three washes with TBS, cells were observed under a fluorescence microscope.

## Affinity purification of AlkNSF

To make AlkNSF pre-coupled beads, 2.5 mg (125 µL) azide beads (Tamagawa Seiki) with 125 µM AlkNSF or Alkyne oleic acid (AlkOle) (Cayman), 62.5 µM tris[(1-benzyl-1H-1,2,3-triazol-4-yl) methyl]amine (TBTA) (Sigma), 1.25 mM $CuSO_4$, and 1.25 mM (+)-sodium L-ascorbate (Sigma) were incubated for 16 h at room temperature. The pre-coupled beads were three times washed with t-BuOH(Supelco)/DMSO/water (4:1:5) and three-time washed with 50% (v/v) methanol. The washed beads were washed and resuspended with Lysis buffer (150 mM NaCl, 20 mM HEPES pH 7.5, 5 mM $MgCl_2$, 1 mM EDTA pH 8.0, 10% glycerol, 0.25% (v/v) Triton X-100, 0.5 mM DTT, and 1x HALT protease inhibitor cocktail (Thermo Fisher Scientific)).

Cells grew in YES media for 16 h and were harvested at 2500 rpm. Harvested cells were washed with TBS (50 mM Tris-HCl pH 7.5, 150 mM NaCl) twice. Cells were resuspended in Lysis buffer and were disrupted with silica beads by a bead-beating machine (MP Biomedicals™ FastPrep-24™ 5 G Instrument, 3x 20 s at 6.5 m/s, 3 min breaks on ice in between rounds). Cell lysates were collected from a tube having punched a hole in the bottom with a needle, followed by centrifugation. Crude lysates were centrifuged at 13,000 rpm for 10 min at 4 °C. Protein concentration of clear lysates was measured by Bradford assay. As competition assay, cell lysates were incubated with NSF (100 µM for parent cell lysate or 250 µM for *ayr1Δ* cell lysate) for 2 h at 4 °C. Pre-coupled beads of AlkNSF or AlkOle were added and were incubated for 2 h at room temperature. Each condition was done in technical triplicates. The beads were washed twice, each with lysis buffer and wash buffer (100 mM NaCl, 20 mM HEPES pH 7.5, 5 mM $MgCl_2$, 1 mM EDTA pH 8.0, 10% glycerol, 0.25% (v/v) Triton X-100). Washed beads were resuspended with 6 µL digest buffer (3 M guanidine HCl, 20 mM HEPES pH 8.5, 10 mM CAA, 5 mM TCEP) and supplemented with 0.2 µg LysC. After incubation for 4 h at room temperature, 17 µL of 50 mM HEPES pH 8.5 and 20 ng trypsin were added to further digest the proteins. Trypsin digestion was conducted overnight at 37 °C.

## Mass spectrometry

The generated peptides were acidified with TFA to a final concentration of 0.8% and analyzed by LC–MS/MS on an Orbitrap FUSION LUMOS or ECLIPSE tribrid mass spectrometer (Thermo Fisher Scientific) connected to a Vanquish Neo UHPLC (Thermo Fisher Scientific) with a two-column set-up. The peptides were applied onto a C18 trapping column in 0.1% formic acid and 2% acetonitrile in $H_2O$. Using a flow rate of 200 nL/min, peptides were separated at RT with a linear gradient of 2–6% buffer B in buffer A in 1 min followed by a linear increase from 6 to 20% in 42 min, 20–35% in 22 min, 35–45% in 2 min, 40–100% in 1 min, and the column was finally washed for 10 min at 100% buffer B in buffer A (buffer A: 0.1% formic acid; buffer B: 0.1% formic acid in 80% acetonitrile) on a 15 cm EASY-Spray column (Thermo Fisher Scientific) mounted on an EASY-Spray™ source (Thermo Fisher Scientific). The survey scan was performed using a 120,000 resolution in the Orbitrap, followed by an HCD fragmentation of the most abundant precursors. The fragment's mass spectra were recorded in the ion trap according to the recommendation of the manufacturer (Thermo Fisher Scientific). Protein identification and relative quantification of the proteins was performed with MaxQuant v.2.2.0.0 using Andromeda as search engine (Cox et al, 2011), and label-free quantification (LFQ) (Cox et al, 2014). The fission yeast subset of the UniProt v.2021_05 combined with the contaminant database from MaxQuant was searched, and the protein and peptide FDR were set to 0.01. The LFQ intensities estimated by MaxQuant were analyzed with the einprot R package (https://github.com/fmicompbio/einprot) v0.7.6 as previously described (Welte et al, 2023).

## Western blot

Whole-cell lysates (WCL) for western blotting were extracted as described previously (Matsuo et al, 2006). WCL were loaded on Bolt 4–12% Bis-Tris gel (Thermo Fisher Scientific). Gels were transferred using the Trans-Blot Turbo Transfer System (Bio-Rad). The blots were incubated overnight with anti-FLAG M2 antibody (Sigma), or TAT1 anti-tubulin tissues culture supernatant (anti-tubulin, 00020911) at 1:1000 dilution.

## Statistical analysis

For GO analysis, AnGeLi (Bitton et al, 2015) applies the two-tailed Fisher's exact test and the two-sided Wilcoxon rank-sum test as multiple statistical tests. For bar plots of OCR, qRT-PCR, and iBAQ values, data were presented as mean ± SEM. Comparisons between the two groups were made using the student's *t*-test with R software version 4.4.0 in combination with Bioconductor version 3.19 R packages.

## Data availability

All custom codes used to analyze data and generate figures are available upon reasonable request. RNA-Seq and ChIP-seq data sets have been deposited to the Gene Expression Omnibus with the dataset identifier GSE250095. The mass spectrometry proteomics data have been deposited to the ProteomeXchange Consortium via the PRIDE (Perez-Riverol et al, 2021) partner repository with the dataset identifier PXD047795.

The source data of this paper are collected in the following database record: biostudies:S-SCDT-10_1038-S44318-024-00224-z.

## Peer review information

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

## Acknowledgements

We thank the members of the Bühler lab for their constant support and discussions. Special thanks go to Fabio Mohn for data deposition, and Yukiko Shimada and Nathalie Laschet for technical support. We are also grateful to the FMI Functional Genomics facility for library construction and next-generation sequencing and to Laurent Gelman for assistance with fluorescence microscopy. This work was supported by the Novartis Research Foundation (Friedrich Miescher Institute for Biomedical Research, MB), the Japan Society for the Promotion of Science (JSPS; KAKENHI Grant Numbers 18H02131 and 22K05397 to YY and GH, 19K15755 to SO, 23H05473 to MY, and 23H04882 to MY and YY) and the Ohsumi Frontier Science Foundation (to YY).

## Author contributions

**Shin Ohsawa**: Conceptualization; Formal analysis; Funding acquisition; Validation; Investigation; Visualization; Methodology; Writing—original draft; Writing—review and editing. **Michaela Schwaiger**: Formal analysis; Investigation. **Vytautas Iesmantavicius**: Formal analysis; Investigation. **Rio Hashimoto**: Formal analysis; Investigation. **Hiromitsu Moriyama**: Conceptualization; Supervision; Writing—review and editing. **Hiroaki Matoba**: Conceptualization; Methodology; Writing—review and editing. **Go Hirai**: Conceptualization; Funding acquisition; Methodology; Writing—review and editing. **Mikiko Sodeoka**: Conceptualization; Supervision; Writing—review and editing. **Atsushi Hashimoto**: Formal analysis; Investigation. **Akihisa Matsuyama**: Formal analysis; Investigation. **Minoru Yoshida**: Conceptualization; Resources; Supervision; Funding acquisition; Project administration; Writing—review and editing. **Yoko Yashiroda**: Conceptualization; Resources; Supervision; Funding acquisition; Project administration; Writing—review and editing. **Marc Bühler**: Conceptualization; Resources; Supervision; Funding acquisition; Visualization; Writing—original draft; Project administration; Writing—review and editing.

Source data underlying figure panels in this paper may have individual authorship assigned. Where available, figure panel/source data authorship is listed in the following database record: biostudies:S-SCDT-10_1038-S44318-024-00224-z.

## Disclosure and competing interests statement

The Friedrich Miescher Institute for Biomedical Research (FMI) receives significant financial contributions from the Novartis Research Foundation. Published research reagents from the FMI are shared with the academic community under a Material Transfer Agreement (MTA) having terms and conditions corresponding to those of the UBMTA (Uniform Biological Material Transfer Agreement).

# Expanded View Figures

**Figure EV1.   Differential gene expression analysis of cells treated with NSF for 4 or 6 h.**

(A) Pairwise correlation plots comparing log2FC gene expression changes at different time points of NSF treatment. Correlation coefficients were calculated with Pearson's r. (B) MA plot showing differential gene expression (log2FC) in cells treated with MeOH or NSF for 4 or 6 h (y-axis). The 2-h treatment is shown in Fig. 2A. The x-axis denotes total transcript abundance in counts per million (cpm) in both conditions. *p* values were calculated using the Wald test and adjusted with the Benjamini and Hochberg method. Up- or down-regulated genes (FC >1.5 and adjusted *p* value <0.05) are highlighted in pink or blue, respectively. NSF-linked genes revealed by the genetic screen are marked with a black outline. The names of the top two upregulated and downregulated genes are written in red and blue, respectively. The green color was used to label the genes encoding transcription factors. (C) Heatmap of Pearson correlation coefficients between log2FC gene expression changes induced by NSF and glycerol feeding. Pearson's r values were calculated among log2FC gene expression changes of genes with a gene expression level higher than 100 cpm after a 2-h NSF treatment under NCR conditions ([NSF/MeOH]) from two repetitive experiments (Exp.1 and Exp.2) and glycerol feeding ([Gly/Glu]) (Malecki et al, 2016). (D) Box plots showing logFC distribution of NSF-induced gene expression changes upon 4 or 6 h NSF treatment, grouped by gene expression changes induced by respiration (Malecki et al, 2016). *p* values were calculated using a two-sided *t*-test and were adjusted using the Holm method. The center line inside the box represents the median of the data, bounds of the box correspond to the interquartile range with the bottom and top of the box indicating the first and third quartiles, and bounds of whiskers extend to the minimum and maximum values within the range of non-outlier data. Each condition was done in biological triplicates.

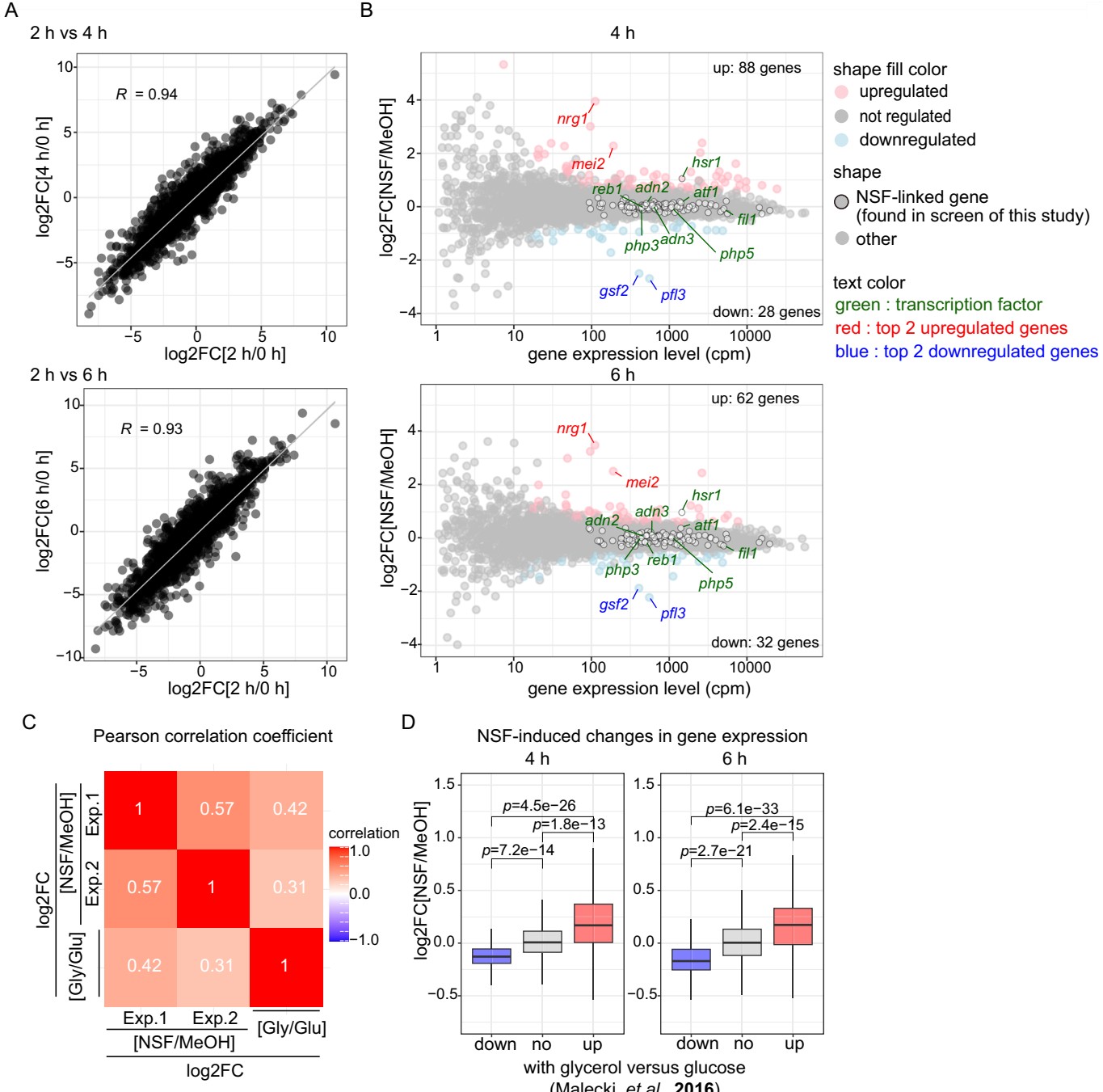

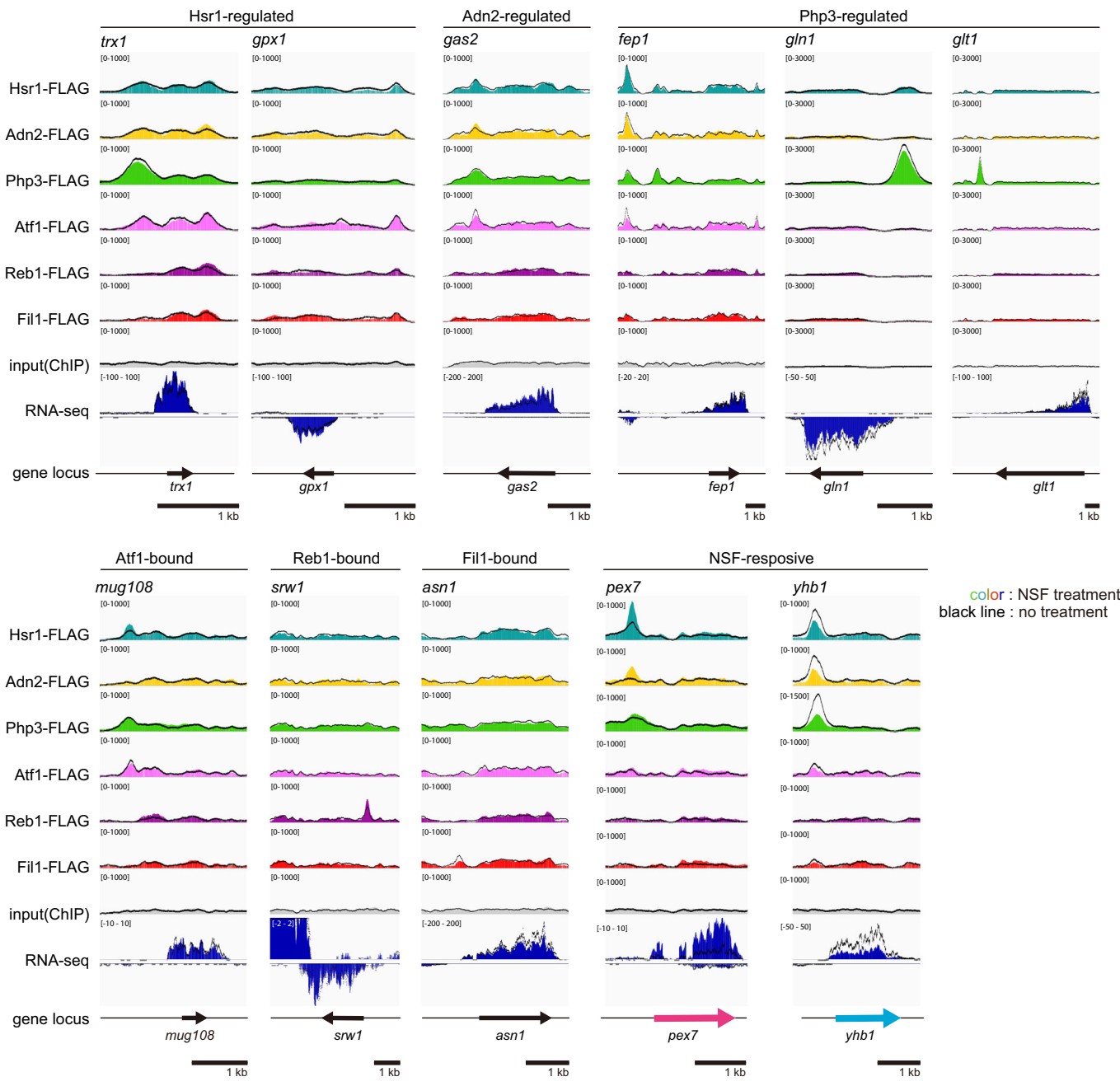

**Figure EV2. ChIP-seq analysis of TFs required for NSF-mediated adaptive growth.**

ChIP enrichments of Hsr1-FLAG, Adn2-FLAG, Php3-FLAG, Atf1-FLAG, Reb1-FLAG, and Fil1-FLAG on NSF-linked genes, showing occupancies of indicated TF on target or putative target genes (Hsr1: *trx1*, *gpx1* (Chen et al, 2008), Adn2: *gas2* (Kwon et al, 2012), Php3: *fep1*, *gln1*, *glt1* (Mercier et al, 2008, 2006), Atf1: *mug108* (Takemata et al, 2016), Reb1: *srw1* (Rodríguez-Sánchez et al, 2010), Fil1: *asn1* (Duncan et al, 2018)). Black lines denote ChIP enrichment in untreated cells. Colored areas represent ChIP enrichments upon NSF treatment. The RNA-seq tracks were derived from RNA-seq data of cells under NCR conditions with or without NSF for 2 h (see Fig. 2A).

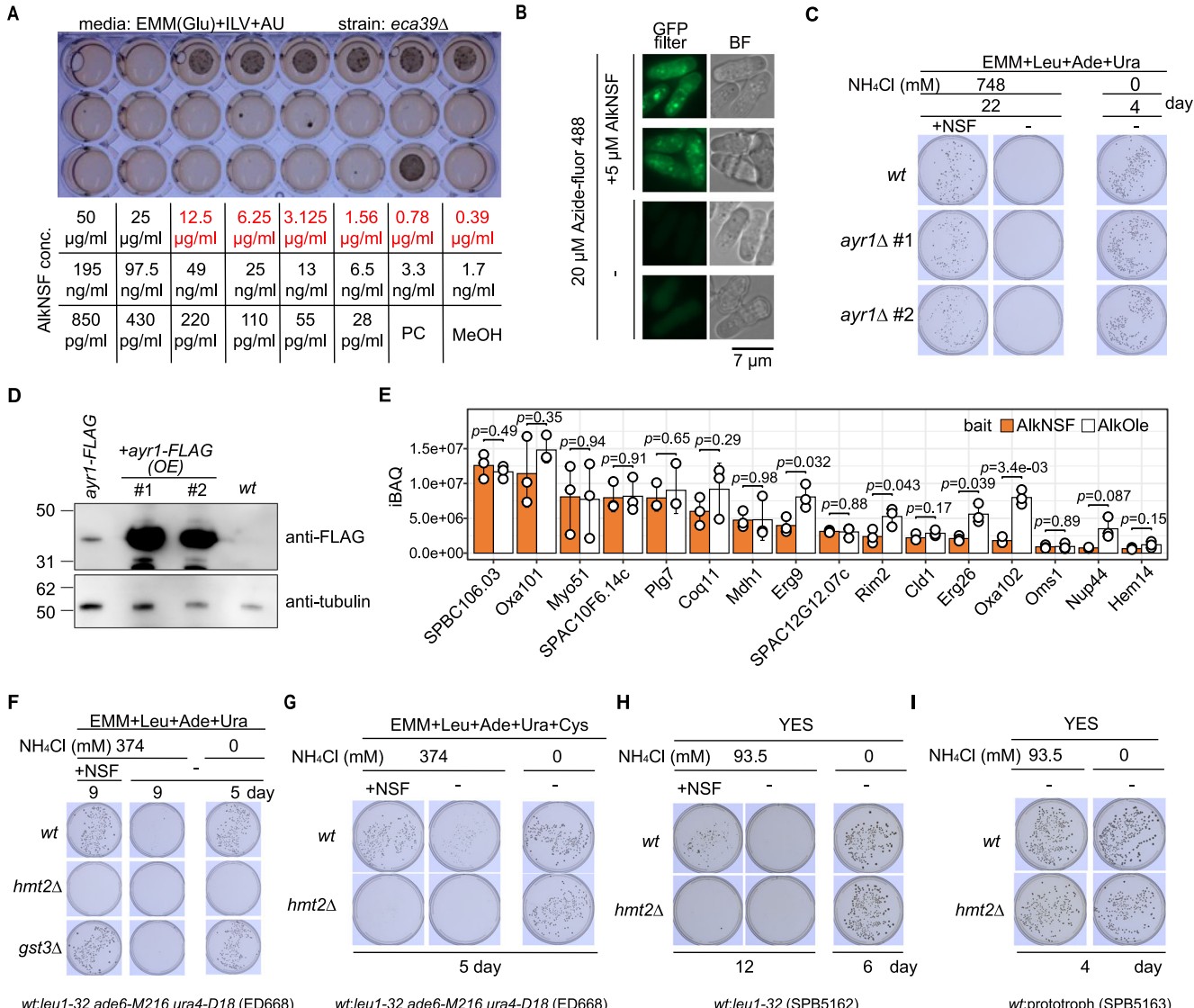

**Figure EV3. Identification of NSF interacting proteins with a chemical biology probe.**

(A) Determination of the minimum effective concentration of AlkNSF. eca39Δ cells were spotted onto EMM(Glu) + ILV + Ade + Ura with decreasing AlkNSF concentrations. As a positive control for this assay (PC), cells were exposed to secreted signaling factors that were purified from the supernatant of an S. pombe culture with the ethyl acetate method (Sun et al, 2016). 50% methanol (MeOH) served as a negative control. The plate was incubated for 6 days at 30 °C. (B) Visualization of cellular AlkNSF uptake by fluorescence microscopy. Cells were incubated with 5 μM AlkNSF in EMM(187 mM NH₄Cl) + Leu + Ade + Ura for 4 h. Subsequently, AlkNSF was conjugated with azide-flour 488 by click chemistry. (C) NSF-mediated adaptive growth of wt (ED668) or ayr1Δ (SPB5108 and SPB5109) cells in EMM medium. (D) Western blot analysis of Ayr1 protein levels in wt (ED668), ayr1-FLAG (SPB5122), and ayr1-FLAG overexpression (+ayr1-FLAG (OE), SPB5227 and SPB5228) strains. Proteins were detected with anti-FLAG and anti-tubulin antibodies. (E) iBAQ values of proteins that co-purify with AlkNSF and AlkOle probes when incubated with wild-type cell lysates. The mean and standard deviation from technical triplicates is shown. p values were calculated using a two-sided t-test. (F) NSF-mediated adaptive growth of wt (ED668), hmt2Δ (SPB5150), or gst3Δ (SPB5158) cells in EMM medium. (G) NSF-mediated adaptive growth of wt (ED668) or hmt2Δ (SPB5150) cells in EMM medium supplemented with 2 mM cysteine. (H) NSF-mediated adaptive growth of wt (carrying only leucine auxotrophy: SPB5162), or hmt2Δ (SPB5234) cells in YES medium. (I) NSF-mediated adaptive growth of wt prototroph (SPB5163) or hmt2Δ (SPB5233) cells in YES medium.

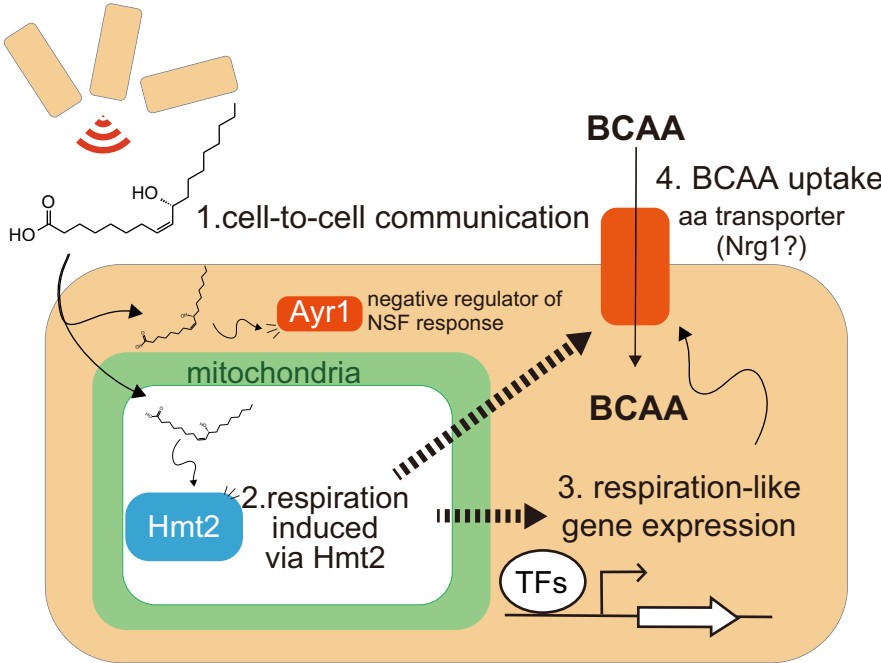

**Figure EV4.  Working model.**

Schematic summary of the key findings of this study. NSF activates mitochondrial respiration by binding to Hmt2, eventually triggering changes in gene expression, the evasion of NCR, and the uptake of BCAA. Ayr1 regulates the response negatively, possibly by metabolizing NSF.

