## [Peer Review File · The EMBO Journal]

Nitrogen signaling factor triggers a respiration-like gene expression program in fission yeast

Shin Ohsawa, Michaela Schwaiger, Vytautas Iesmantavicius, Rio Hashimoto, Hiromitsu Moriyama, Hiroaki Matoba, Go Hirai, Mikiko Sodeoka, Atsushi Hashimoto, Akihisa Matsuyama, Minoru Yoshida, Yoko Yashiroda, and Marc Bühler

Corresponding authors: Marc Bühler (marc.buehler@fmi.ch) , Yoko Yashiroda (ytyy@riken.jp)

Review Timeline:

Transferred from Review Commons:	1st Mar 24
Editorial Decision:	12th Mar 24
Revision Received:	14th May 24
Editorial Decision:	21st Jun 24
Revision Received:	28th Jun 24
Editorial Decision:	1st Aug 24
Revision Received:	8th Aug 24
Accepted:	16th Aug 24

Editor: Daniel Klimmeck

Transaction Report:

This manuscript was transferred to The EMBO Journal following peer review at Review Commons.

Review #1

1. Evidence, reproducibility and clarity:

Evidence, reproducibility and clarity (Required)

Summary

We have now reviewed the manuscript by the groups of Dr. Bühler and Dr Yashiroda entitled: "Nitrogen signaling factor triggers a respiration-like gene expression program". We have enjoyed the topic, the experiments and the science behind it all. The authors study here one part of the fission yeast 'quorum sensing'-like mechanism of counteracting NCR, mediated by the small molecule NSF: they identify pathways required to respond to NSF, and more specifically determine the mechanism by which NSF counteracts NCR: triggering respiration. This is a very interesting manuscript, with nicely executed experiments, and the topic is of great interest. Regarding the major comments, they are specific to the current data. The minor comments are questions raised in light of the present set of data, which should be appropriate for future research and future manuscripts.

Major comments

1. Consistency with the numbers of mutants/genes should be improved. Line 119: 203 genes, 206 in 4th datasheet of Table S1; line 139: 117 mutants but 119 analyzed for GO analysis (1st datasheet in Table S2); lines 179 and 181: where are these numbers (92 down and 156 up) coming from? (compare with 74 and 98 in line 167) (maybe they come from the merge of 2, 4 and 6 h, but it is not indicated).
2. Lists of genes up- and down-regulated from the RNA seq data should be provided. GO terms are not useful. Add supplementary table, please.
3. Comparing the transcriptomic response to that of Malecki et al 2020 in response to Antimycin A (EMBO Rep. 21:e50845) would be useful.
4. The optical densities and whether NCR has been induced has to be clearly specified in each experiment. For instance, RNA seq data. Line 165: for the transcriptome experiment, NSF is added or not to low density cells (not indicated in results, figure legend nor materials and methods). Should addition of NSF to wild-type strain en MM trigger the same transcriptomic changes?
5. Fig. 1G: Addition of NSF can enhance oxygen consumption at any cell density? And in prototrophs? And without NCR? Add in figure legend that this has been done at OD600 of 0.01.
6. Fig. 1 E and F: why 14 d after growth there is not growth at 2% glucose in panel F, but it is 10 d after in panel E? What is EMM Biomedical?
7. Fig. 2BC: Venn diagrams should be more useful to demonstrate overlap with the Malecki data.
8. Fig. 4A: not very useful
9. Is Hsr1 required for some of the RNA seq changes upon NSF addition? Same with other TFs
10. Line 287: '...it is tempting to speculate that Ayr1 dampens adaptive responses by metabolizing NSF'. Calculating MEC for NSF in delta ayr1 and in cells over-expressing Ayr1

would be required to confirm this speculation. According to Pombase, cells lacking Ayr1 have their respiratory functions compromised (no growth in galactose, glycerol...), why is so? The opposite should be expected, if NSF-mediated respiration is enhanced in this background.

11. Regarding the two pull-down experiments, one to identify Ayr1 and the second Hmt2, why different negative controls are used? Is addition of NSF to WCE prior to pull-down also used in the second experiment (with delta ayr1 and AlkOle)?

12. The data regarding Hmt2 is very interesting. As for delta ayr1, delta hmt2 cells cannot grow in glycerol nor galactose according to Pombase. Is the result shown in Fig. 5J (lack of NSF-dependent activation of nrt1 and mei2 in delta hmt2) a consequence of the absence of the NSF receptor, or is it due to the lack of respiration of this background? Is delta hmt2 really auxotrophic for Cys? Why? In this background, H₂S should be enhanced, and Cys and Met biosynthesis improved. In fact, in one manuscript these cells grow fine in SG minimal media (Mol Microbiol 01 42:29), while another report indicates they are auxotrophic for Cys (Genes to Cells 2016, 21:530).

13. M&M: regarding RNA isolation and sequencing: add info about OD of cultures, genotype (leu1-32?), growth media; also, number of replicates and filtering (fold-change used, Q value...)

14. M&M, ChIP seq: same as above. Also, MACS2 can be used for the unbiased identification of bona fide TF targets, by using a quantification tool reporting percentage of occupancy upstream the TSS (callpeak function).

****Minor comments****

1. Who triggers NCR? Analysis of 137 genes in Figure 1b.

2. Synthesis of NSF: how is it regulated, where does it come from?

3. NCR impairs import of BCAA. How are the aa importers such as Cat1 or Aqp3 eliminated from the plasma membrane? Transporter internalization, degradation, transcriptional repression... And how does NSF block the NCR regarding aa uptake? Or aa usage?

4. How can enhanced respiration by NSF counteract all of the above? How can now leu1-32 cells grow?

5. Addition of NSF to any cell type would do the same, enhance respiratory rates? With or without previous NCR? Should this signaling molecule also drive different respiratory rates in a cell density-independent manner regarding glucose catabolite repression?

2. Significance:

Significance (Required)

Within this manuscript, the authors study a cell-to-cell communication process, by which nitrogen catabolite repression can be counteracted by a small molecule called NSF. Specifically, the authors demonstrate here that NSF up-regulates respiratory metabolism as a mechanism to overcome the repression of amino acid internalization, which was blocked by excess nitrogen. This is a wonderful manuscript, with splendid data, on a very interesting topic.

3. How much time do you estimate the authors will need to complete the suggested revisions:

Estimated time to Complete Revisions (Required)

(Decision Recommendation)

Between 1 and 3 months

No

Review #2

1. Evidence, reproducibility and clarity:

Evidence, reproducibility and clarity (Required)

This paper uses the model system *Schizosaccharomyces pombe* to investigate how the oxylipin nitrogen signaling factor functions to send signals and adapt the metabolism upon a change in nutrients in the environment. Combining genome-wide screens, RNA-sequencing and chemical biology, the authors find that the nitrogen signaling factor triggers a change from fermentation to a respiratory metabolism, through a direct interaction with a mitochondrial oxidoreductase Hmt2.

****Major comments:****

Overall, the manuscript lacks readability and coherence. Quite a lot of genes/TFs and proteins are mentioned, it is difficult to find a coherent story and clear overview and connection between these subparts. The manuscript would benefit from a general proposed scheme/working mechanism in the discussion and streamlining the results and data into a single biological storyline.

Several statements or results are not sufficiently clear, elaborated or nuanced. The paper would benefit from more explanation and discussion.

In the discussion section, the authors are not consistently referencing figures.

179-181: 'GO enrichment analysis of the 92 downregulated genes' but on line 167, it is '74 downregulated genes' that are mentioned. It is unclear where this difference in number of downregulated genes comes from. Similarly, for the upregulated genes. '156 genes' are mentioned on line 181 but only '98' on line 167.

189: The statement that the downregulation of flocculation could serve to avoid mating, though sounding logical, is undermined by the finding that mating-related processes are upregulated in the experiment. I find this statement rather speculative

247-249: The statement is too broad, the effects are visible for maybe 3 TFs, the others don't seem to make a difference in occupancy. Also, why are these two highlighted genes of importance (*pex7+*, *yhb1+*), this is the first and last time they are mentioned?

254-256: The statement that these TFs are indispensable may be too strong. Right before, the authors showed that most of these TFs don't change occupancy (and especially *Fill* and *Reb1* do not show a correlation with up- and down-regulated genes, nor does *Fill* in FigS2B show a changed ChIP-seq signal).

365: 'independently of the carbon source'. As far as we can see, all experiments were performed using glucose as the carbon source, so this statement seems too strong as there is no clear proof for this. This could be an easy extra experiment to perform these tests on media with other carbon sources than glucose?

Fig1E: It is not clear if the experiment was performed with the Wild-Type or a deletion strain. In the case of the WT, colonies grew in the media not containing NSF but in Fig5E and Fig5I, the WT did grow in the media not containing NSF. It could be more relevant to plate out 1 colony like in the second screen. Thus, unless different strains were used for both experiments, the results seem inconsistent with each other, which is not mentioned in the manuscript.

Fig1E, Fig5B, Fig5E, and Fig5I: For these experiments, different nitrogen concentrations were used depending on the media, but this has not been addressed/mentioned in the manuscript.

****Minor comments:****

53-57: I would like more elaboration on why CCR and NCR are important for virulence of human pathogens or relevant for industrial applications, and link back to this in the discussion. Otherwise, it is superfluous to include this in the introduction.

108: Does having a h- library have any impact on the outcome compared to the original h+ library?

170: Why would only one of the 117 NSF-linked genes change expression in the RNA-sequencing experiment? Any explanation as to why the expression remains unchanged for the 116 other NSF-linked genes?

212: Please elaborate the discussion of these results. I understand the point that at low cell densities, the cells do not produce NSF as much, and thus adding NSF induces *nrt1*⁺. However, the added value of testing this in different media is unclear, especially when the results of strength of increase in *nrt1*⁺ show the opposite trend for the two different media between low and high nitrogen content.

216: Why was the *ADH1*⁺ promoter chosen as a 'negative control'?

284-288: Fig5E: To test whether *AYR1* is indeed metabolizing NSF (and thus supporting this statement), an overexpression strain of *AYR1* could be made to see if it grows on the EMM + *NH4Cl* without NSF added.

385: 'NSF would not strictly revoke NCR only, but also CCR': the authors should try to provide experimental evidence, citation(s), or clearly state it to be a hypothesis. This comment links back to the major comment on line 365.

405: typo: strains were validation, should be 'validated'

644: typo: '+' sign not in superscript

Fig1G: Could the differences in OCR be due to differences in growth rate or remaining glucose? It could for instance be that the culture in the control condition grew less fast, thus still having glucose and therefore still in fermentative metabolism. Showing or mentioning growth rates, nutrient concentrations could help to strengthen this finding.

Fig2A: The top two 'most' upregulated genes (*nrt1* and *mei2*) were taken along for additional experiments. However, one gene with a significant upregulation labeled in red on the left seemingly shows stronger induction than the second gene (*mei2*). Why was this gene not taken along?

Fig2C: the x-axis label is not immediately clear to the reader.

Fig4B: typo: 'non treatment'

2. Significance:

Significance (Required)

This manuscript advances our understanding of nitrogen signaling pathways and nitrogen catabolite repression in the model organism *S. pombe*. Specifically, it shows how a nitrogen signaling factor functions to send signals and adapt the metabolism upon a change in nutrients and reveals that this nitrogen signaling factor triggers a change from fermentation to a respiratory metabolism. These findings are relevant for the broad fields of applied microbiology, signal transduction and metabolic regulation. Relevant literature is appropriately cited, although the links with Crabtree repression in *S. cerevisiae* are perhaps not fully supported.

This manuscript was reviewed by experts with expertise in *S. cerevisiae*, Crabtree effect, respiration-fermentation balance, adaptation to changing environments.

3. How much time do you estimate the authors will need to complete the suggested revisions:

Estimated time to Complete Revisions (Required)

(Decision Recommendation)

Less than 1 month

Yes

Review #3

1. Evidence, reproducibility and clarity:

Evidence, reproducibility and clarity (Required)

****Summary:****

In this manuscript the authors are interested in understanding how fission yeast respond to a Nitrogen Signaling Factor (NSF) that has previously been shown to allow Leucine auxotrophs to grow in the presence of Leucine when Nitrogen Catabolite Repression (NCR) is triggered by the presence of a high quality Nitrogen source such as Ammonium Chloride (NH₄Cl).

The authors begin with a screen to identify genes that affect the ability of wild type cells grown near cells with leucine auxotrophy to enhance or abolish NCR phenotype. They screened the

non-essential gene deletion library which they manipulate so that it only contains a leucine auxotrophy (unlike the original gene deletion library which contains additional auxotrophies). They identify 137 genes whose deletion allows growth of Leu auxotrophs in the presence of Leucine and Ammonia without the presence of WT cells. These genes are required for NCR. They further identify 203 genes which do not bypass NCR even in the presence of wild type cells, and are thus important for bypassing NCR in the presence of WT cells.

They then conduct a second screen to identify which of these genes are important for bypassing NCR in response to the Synthetic NSF, 10(R)-hydroxy-8(Z)-octadecenoic acid, by looking for genes which grow in the presence of leucine when ammonia is not present, but do not grow in the presence of leucine when ammonia is present, even when NSF is added. This second screen identifies 117 strains carrying deletions in a gene set enriched for genes related to cellular respiration and mitochondria. They then show that the NSF bypass of NCR is linked to respiration by showing that it is abolished in the presence of the respiration inhibitor Antimycin A, that growth in low levels of glucose can bypass NCR in the absence of NSF and that cells supplemented with NSF have a higher oxygen consumption rate.

To gain insight into how the cell responds to NSF, the authors then gather RNA expression data from cells grown in high ammonium concentrations following treatment with NSF relative to a negative control treated only with Methanol (the vehicle into which NSF is dissolved). They argue that the gene expression pattern resembles gene expression data from cells undergoing respiration in glycerol relative to cells undergoing fermentation in glucose. They show that the upregulated genes relate to trehalose synthesis, detoxification of Reactive Oxygen Species, and cellular fusion and the downregulated genes are related to cellular adhesion and flocculation.

They validate their RNA-seq measurements by showing that the two most highly induced and two most highly repressed genes respond to NSF addition in a dose dependent manner and do not respond oleic acid which is chemically similar to NSF. The most highly responsive gene they identify is an uncharacterized gene, SPBPB2B2.01, which they suggest naming "NSF-responsive amino acid transporter 1" (*nrt1*). They also show that the *nrt1* response is dependent on the culture density, and that the response is present (though the magnitude varies) in YES and in EMM under varying nitrogen concentrations, and that *yfp* driven by the *nrt1* promoter is induced by NSF.

The authors then investigate the 8 transcription factors that were present in their list of genes required for NSF-mediated adapted growth. They note that *Hsr1* was the only one of these transcription factors, indeed the only gene, that was a hit in their screen for NSF-mediated adapted growth and whose expression was induced upon NSF treatment. To see if the activity of the other transcription factors changed in response to NSF treatment, the authors then gathered ChIP-seq data using 6 of these transcription factors as targets for IP. They saw that for *Hsr1* and *Php3*, targets that had increased RNA-seq expression showed an increase in promoter occupancy while for *Hsr1*, *Php3*, *Adn2*, and *Atf1*, genes that had decreased RNA-seq expression showed a decrease in promoter activity.

Finally the authors attempt to identify the mode of action of NSF by generating a functionalized NSF with an alkyne tag (AlkNSF) which they then use as a probe to identify NSF binding

partners. They first show that AlkNSF does allow bypass of NCR, although at 30-fold higher concentration. Also AlkNSF induces *nrt1* expression in a dose dependent manner, although the expression saturates at a lower level and requires a much higher concentration for induction. They then look for proteins that co-purify with AlkNSF compared to a control that was pre-incubated with NSF which was expected to compete off AlkNSF. The only significant protein they saw was *Ayr1*, which was not identified in their screen and which did not abrogate NSF bypass of NCR when deleted independently. They saw that *Ayr1* deletion actually increases the response of *nrt1* and *mei2* targets to NSF, and speculate that *Ayr1* metabolises NSF and reduces the cell's ability to respond to NSF to bypass NCR.

They then repeat the affinity purification / mass spec protocol in an *Ayr1* delete cells to identify other interaction partners, this time incubating with a higher concentration of NSF, and also comparing to an experiment using Alkyne Oleic Acid as a control for non-specific binding. The top two specific hits from this assay are *Hmt2* and *Gst3*. NSF was still able to rescue NCR in *gst3* deletes, indicating that it was not relevant for the phenotype. Cells lacking *hmt2* did not grow in EMM, but did grow in YES when not supplemented with ammonium and when supplemented with ammonium did not grow, and addition of NSF did not rescue growth. They also see that *nrt1* and *mei2* gene induction in response to NSF is abolished when *hmt2* is deleted. They then argue that *hmt2*, a sulfide:quinone oxidoreductase localized in the inner membrane of mitochondria is a direct target of NSF that triggers a switch to respiratory metabolism and allows bypass of NCR.

Below are comments that I think ought to be addressed prior to publication (Major comments)

1. In line 70, the authors state that "S. pombe cells rely on their own BCAA synthesis to sustain growth" when grown alongside Leucine when ammonium is supplied in the media. If prototrophs can inhibit NCR via NSFs in neighboring auxotrophic cells on the same plate, couldn't they also inhibit NCR within their own colony? How do we know that prototrophic cells grown in high quality nitrogen sources along with, say leucine, are not taking up leucine? The fact that leucine auxotrophs cannot grow in high quality nitrogen sources when leucine is present does not imply that wild type cells must use be synthesizing BCAAs rather than importing them. In a recent paper (Kamrad et al Nat. Microbiol. 2023, <https://www.nature.com/articles/s41564-022-01304-8>), it was shown that *S. cerevisiae* cells grown in lysine and in high concentrations of ammonium uptake lysine rather than synthesize it as lysine concentrations in the media are increased. I am aware via unpublished results that this is the case for Leucine as well. I would be surprised if the same isn't true in *S. pombe*. The authors should caveat or remove this assertion.
2. It is important for the authors to put their observation linking respiration to rescue from NCR in context with findings from a closely related study (Chiu et al 2022) which included some authors from this manuscript and which the authors cite. In that paper, it was shown that the siderophore ferrichrome can also rescue NCR in fission yeast. That paper stated "It is likely that ferrichrome increased mitochondrial activity, which enabled efficient utilization of glucose downstream of the glycolytic pathway" based on experiments in different concentrations of glucose. This evidence seems to support the link between respiration and rescue from NCR proposed by the authors of this manuscript. The authors should acknowledge this closely related and earlier work as it strengthen's the case they are trying to make. They could even test if

ferrichrome addition makes cells sensitive to antimycin A (as in fig 1E), but that extra experiment would be optional in my opinion.

3. In figure 1B for the second screen I do not understand what the photos represent. For the photos, two rows are meant to have no NH₄ and also no NSF and the label on that image makes no mention of Leucine supplementation. In the diagram there are two rows that have NH₄ and leucine and one row that has no NH₄ but does have leucine. I assume the diagram is correct and the labels on the images are incorrect.

4. It would be important for the authors to put their observation linking respiration to rescue from NCR in context with findings from Chiu et al 2022 which the authors cite. In that paper, it was shown that the siderophore Ferrichrome can also rescue NCR in fission yeast which the authors cite which found that a siderophore rescues NCR. Also the authors of that paper stated "It is likely that ferrichrome increased mitochondrial activity, which enabled efficient utilization of glucose downstream of the glycolytic pathway." based on experiments in different concentrations of glucose. This evidence seems to support the link between respiration and rescue from NCR proposed by the authors of this manuscript.

5. In line 133. The authors state that the 29 mutants that didn't grow under Leucine supplementation either without NH₄Cl or with NH₄Cl whether or not NSF was present were "related to EMM Growth, leucine uptake, or utilization of ammonium as the sole nitrogen source." The first two make sense, but I can't see why a strain with deletion of a gene related to utilization of ammonium as a sole nitrogen source wouldn't grow when supplemented with leucine. In fact for all the leucine auxotrophs in the screen, if one was to try to grow them with ammonium as the sole nitrogen source they would not grow, so it isn't clear that this screen can identify genes responsible for utilization of ammonium as a sole nitrogen source. The authors should clarify or remove this point.

6. 203 strains are important for avoidance of NCR (because in the presence of Ammonium and Leucine, as well as a WT strain, they cannot grow). Of these 57 strains can't grow in the presence of a WT strain but they can grow in the presence of NSF. The authors conclude in line 138 that these strains are "likely to respond to a transmissible signal that is different from NSF". This is confusing because deletion of these genes still does allow cells to respond to NSF, however when these cells are growing in the presence of wild type cells (which in their model are releasing NSF), the cells don't grow. I am confused about the nature of the transmissible signal that the authors suggest. It would appear that when these genes are deleted and grown next to a wild type cell which sends the alternative signal and the NSF, the other transmissible signal would inhibit the ability of NSF to release NCR (as NSF can still rescue the gene). It is not clear how the other transmissible signal would work when the gene is present as it is clearly not necessary to rescue growth.

A simpler explanation might be that there was contamination in the second screen, or that there was a threshold effect - perhaps in the first screen the strains grew just below a threshold and in the second screen it grew just above that level.

The authors should clarify their interpretation for these strains, and acknowledge any alternative technical explanations.

7. The authors' efforts to remove confounding effects that might stem from additional auxotrophic alleles made the screen more convincing. However, Fig 1E, 1F, 5B, and 5E were done with EMM+Leu+Ade+Ura, while the initial strain was just done in the presence of

additional Leucine. It is unclear why this was done from the text and captions, but I assume it was because they used a strain that was ade- and ura- in addition to being leu-. Given that they had strains without these additional mutations, this seems like a strange choice. The authors should acknowledge that there are possible confounding effects of adding adenine and uracil to the media, and, if they did have additional metabolic deletions, acknowledge that that could possibly be confounding.

8. Fig 1E, it appears that cells can grow without NSF in the presence of ammonium and additional amino acids after 10 days (although NSF is required for growth at 5 days). This is not a problem for the screen as that was taken at 5-6 days, but it appears as though NSF does not rescue growth so much as speed it up. The authors should acknowledge this when describing the phenotype. It also argues for a quantitative time course growth experiment to compare growth over the course of 10 days with and without NSF, although this would not be necessary to the paper's main argument.

9. In line 191 and 192, the authors suggest that the "downregulation of flocculation/adhesion related genes by NSF could serve to avoid undesirable mating during growth". If this is the case, I don't understand why mating genes and cellular fusion genes would be upregulated. What do the authors mean by undesirable mating? Wouldn't flocculation increase desirable mating as well? If all mating is undesirable, wouldn't upregulation of mating and cellular fusion genes be detrimental?

10. The authors mention that trehalose is an antioxidant, for which they reference Malecki 2019, however that paper shows no direct evidence of trehalose functioning as an antioxidant under respiratory conditions. It only shows that some trehalose synthesis genes are upregulated when cells are grown under glucose. The authors should identify primary literature to back this statement up, or soften the wording. Also trehalose is known to be a storage metabolite (which is mentioned in Malicki et al 2019, but not in this manuscript). In fact work in budding yeast has shown that trehalose can be a shared metabolite that can be produced by respiring cells and used as a fermentable carbon source in communities of budding yeast cells that consist of fermenting and non-fermenting cells (Varahan et al, eLife 2019 <https://doi.org/10.7554/eLife.46735>). It seems that this role should be considered as an alternative explanation for the induction of trehalose in respiratory cells.

11. Line 208: The stimulatory effect of NSF on NRT1 decreased with cell density, thus cell density is likely to be an important factor in terms of gene expression. The methods section, text and figure captions do not mention the density at which cells were inoculated/harvested for RNA-seq and other experiments. If that density was more than OD 0.1, then this would be inconsistent with the measurements from Fig 3. Also in fig 3D, The culture density is not mentioned in the figure or the caption, even though the text suggests that for that experiment cells were grown at low density (Lines 212-213). The authors should provide information on density for their experiments in order for them to be reproducible, as they show it is a key factor.

12. In suggesting a name for NRT1 (NSF-responsive amino acid transporter 1), the authors assume that the gene has a role in amino acid transmembrane transport, but they have no experiments showing this phenotype. They mention that it is Inferred from homology with other amino acid transporters. I presume this name has already been approved by Pombase and is not provisional, but it seems that including phenotypes inferred from homology, rather than from experiments is unwise. Do the authors have any other direct evidence that this is a bona fide Amino Acid Transporter? Perhaps a name like "NSF-responsive gene" would be more appropriate.

Related to this, it appears that the expression level of Nrt1 may be very low (see Fig S2B in which the scale of the RNA-seq track is very small [-1,1] and the amount of expression is very small even when NSF is added). Looking at Fig 2A, the total transcript abundance did not appear to be very low in terms of counts per million (over 100) is this a discrepancy in fig S2B? Perhaps the large fold change is the result of counts very close to zero in the control condition? Also in Fig 3 the nrt1 expression levels did not appear to be especially low and they appeared repeatable. Is the RNA-seq data shown in fig S2B for nrt1 a fluke or am I misinterpreting it?

13. To show that their Chip-seq worked, the authors showed specific examples of Chip-seq reads for target genes Line 240, "Previously determined target genes of these TFs were significantly enriched in our data set, demonstrating that the experiment has worked (Figure S2A)." Is the significance here, the threshold from fig S2B? If so that threshold should be clearly stated here in the text. If it is the fact that *asn1* shows up as "Fill bound" is strange as there are no genes that had significant changes in ChIP-seq signals for fig S2B. If there is another threshold the authors should describe it. While some of the examples they showed were convincing (e.g. *php3*-flag for the *php3* regulated gene *gln1* and the increased reads for *srw1* for the *reb1* target *srw1*), there were some targets that didn't seem to be especially enriched for their designated transcription factor. For example, the gene *trx1* which was identified as an *Hsr1* binding target had some binding from *Hsr1*, but more from *Php3* and equivalent amounts for many of the other transcription factors. A clear description of how genes are chosen to be significant in the text, alongside references/selection criteria the authors used to select the specific genes shown should be provided to improve reproducibility.

14. In lines 244-246 the authors state that "These differences in TF occupancy were positively correlated with target gene expression changes. That is, individual genes that were upregulated by NSF tended to be more strongly bound by the TFs, whereas downregulated genes were less occupied by the respective TFs (Figure 4A)." This is far from a general trend. The trend is not there for *reb1* and *fill1*. In fact *fill1* looks to the eye like it shows a decrease in occupancy for genes with increased expression, and I worry that the authors did a one sided test for significance that would have missed this, although the variability of the genes that don't change in this case is very high, so there could be no significant effect. The authors elaborate on some of the detail in following statements, but they should soften or remove this statement.

Related to this, in line 254, the authors state: "These results imply that NSF exposure rewires the recipient cell's transcriptional program, for which the TFs *Atf1*, *Adn2*, *Adn3*, *Fill1*, *Hsr1*, *Php3*, *Php5*, and

Reb1 are indispensable (Table S3)." While I am convinced from the RNA-seq evidence and some of the chip-seq evidence that NSF exposure rewires cell's transcriptional program, I am not convinced that the 8 transcription factors they mention are indispensable for rewiring the transcriptional program. While they may be indispensable for the phenotype itself, *Reb1*, and *Fill1* show no significant enrichment in occupancy of upregulated or downregulated targets (Fig 4A) and, along with *Atf1*, *Reb1*, and *Fill1*, have very few genes in which occupancy is changed significantly (Fig S2B), while no chip-seq experiments were shown for *Php5* and *Adn3*.

The more specific summary of the data (Lines 250-253) from Fig S2B describing how *hsr1* and *adn2* have the strongest effects of the transcription factors required for NSF-mediated NCR bypass is a much stronger message for this section.

15. In line 335, the authors state that "in contrast to other communication systems, NSF does not induce noticeable changes in *S. pombe*'s morphology", referring to changes in mating, filamentation, and bacterial biofilm formation. However they do show very clearly that NSF does cause a large decrease in expression in flocculation/adhesion genes. The fact that they do not see a change in morphology is likely due to the fact that the lab strain in the conditions used for this assay do not flocculate. We have recently identified conditions and strains which do exhibit flocculation in this preprint [<https://www.biorxiv.org/content/10.1101/2023.12.15.571870v2>]. It is likely that if they had a strain and conditions that did flocculate addition of NSF would break up flocculation and thus change the morphology based on their evidence. The authors should remove or caveat this point.

16. Line 270 Fig 5B: The concentration of NH₄Cl listed in the text (374mM) does not match the concentration shown on the figure (748mM). I assume this is a typo but it should be corrected prior to publication.

Also I have several minor comments to help improve the manuscript.

m1: Lines 66-70- state that "uptake of the branched-chain amino acids (BCAA) isoleucine (Ile), leucine (Leu), and valine (Val) is suppressed in the presence of high-quality nitrogen sources such as ammonium or glutamate, because the expression of transporters or permeases that are needed for the uptake of poorer nitrogen sources are down regulated (Zhang et al, 2018)." This reference is for *S. cerevisiae* and is a review. The authors should cite original results in *S. pombe* if possible, and if that is not available, alert the reader that this result is from a different species.

m2: It is unclear from the methods section how the images taken for the screens were analyzed. Were they analyzed and scored by hand, or using custom image analysis software. Either way, when publishing the authors should publish the scores for each deletion mutant in their screen. If there was custom image analysis, the authors should mention in their methods the cutoffs which they used to score growth, and consider plotting the data as a supplement so readers can get a sense of how sensitive the screen was.

m3: The authors identify 137 mutants that did not require NSF signaling to bypass NCR and claimed these genes were required for NCR. It would be helpful and give more confidence in this screen to demonstrate the extent to which the genes identified in this study overlap with any previous genes required for NCR, and whether there was any GO-term enrichment in this set.

m4: It would be interesting if the authors could speculate a bit in their discussion on why mitochondrial respiration counteracts NCR. Is there something about cells undergoing respiration that would make it easier for them to use BCAAs than to produce them, or conversely something about fermenting cells that makes it easier for them to produce BCAAs rather than importing them?

m5: It is unclear why Figure 1F has 'MP biomedical TM' listed in the figure. It doesn't seem to be listed in the caption or the methods. Is this different media than in other experiments? If so, the authors should add that information to the methods or the caption.

m6: In Line 160, positively influenced is strange wording, do the authors mean "induced"?

m7: In the section on gene expression change upon exposure to NSF, the authors use a + after each gene name. My understanding is that that notation is meant to refer to strains with the wild type genotype of that gene, and not the gene itself. Shouldn't the gene be italicised in lower case to represent the gene? See: Lera-Ramirez et al 2023 <https://doi.org/10.1093/genetics/iyad143>.

m8: In Fig 2A, genes are displayed on a plot that depicts level vs log₂FC, but a comparison between the fold change and p-value would be more useful, and I believe DESeq2 should provide an adjusted p-value for these genes. A related issue is that it appears as though there were no biological replicates, though there was data gathered at different time points. In these genome wide experiments, replicates can give confidence to data and help distinguish true change from intrinsic variability of expression in specific genes. Though the authors did qPCR to validate specific results, it would have improved the quality of their systems-level data to have replicates for these and other key experiments (Chip-seq, affinity purification and even the screen).

m9: Supp Fig S1: To show that similar gene expression profiles exist for other time points, it would be more convincing to show Log fold change 2h vs 4h and 2h vs 6h and show correlation, or else to make a heat map with all genes to see that genes that go up in one condition go up in the other conditions. It is not clear if the red and blue colors are defined for the 2h dataset and then mapped onto the 4 and 6h dataset, or if they are independently assigned for each plot.

m10: Mbx2 is a key transcription factor related to flocculation and adhesion genes, and its expression is correlated with expression of its targets. If this transcription factor's expression levels decreased in response to NSF, that might strengthen and help explain the decrease in expression the authors observe in flocculation/adhesion genes when cells encounter NSF. If it does not change, it might also be interesting for readers interested in these phenotypes.

m11: In Fig 3D, The notation for the Ammonium concentrations for EMM and YES are inconsistent (+ vs parentheses), also the units (mM from the caption) are not on the figure, but the abbreviation "N" is which is confusing and inconsistent with the other plots in which NH₄CL is not abbreviated. Additionally, the caption lists additional nutrients in the media for the EMM conditions (Leu, Ade, Ura) which ought to also be listed.

m12: In lines 233-235, the authors say "One possibility is that they remain bound to their target genes but become activated or deactivated by NSF directly, or posttranslational modification, such as phosphorylation in the case of Atf1". I don't think the authors intend this, but this sentence could be taken to mean that Atf1 has been shown to be phosphorylated by NSF in the reference they cite. I think the authors should clarify, i.e. by saying "...such as phosphorylation which is known to regulate activity of Atf1 in response to oxidative and osmotic stress [Lawrence et al 2009]".

m13: In Fig 4B and Fig S2A, there are grey and colored tracks for the chip-seq (- and + NSF), but they are very difficult to see. If grey is in front it is hard to tell how close the colored peak when the colored peak is lower. For example, grey is in front for pex7 while color is in front for

yhb1. Could the authors add some transparency so that the data for both conditions could be seen at once?

Also there is little information on the control. My assumption for the input(ChIP) sample was that it was cross-linked and sonicated but not immunoprecipitated, but it is not clear what conditions it was in. I would assume it was done without NSF treatment in WT cells, but those details should be added in the caption or methods. In particular, in the input there is a large spike for Gsf2. Do the authors have any explanation for this and does it have anything to do with that gene's NSF responsiveness?

m14: The authors might consider putting something like Fig S2B (or even a corresponding volcano plot) as a main figure for Fig 4 in addition to the other two panels, as the individual examples from fig 4B are nice to see, but do not give a broad overview of the data.

m15: In line 348, the wording "Would score" might be better replaced by "would be identified."

2. Significance:

Significance (Required)

Assessment:

In general I find the authors arguments compelling and their experiments convincing. The initial and follow on screens were well designed and the authors linked respiration and the action of NSF in a convincing way. The analysis of RNA-seq data was also convincing, especially regarding the decreased expression of flocculation and adhesion genes, and the follow up of specific targets gives confidence in the data (though see Major point 12 below regarding the naming and expression levels of nsf1). The identification of hmt2 as a functional target of NSF was compelling and rigorous, and the authors offer an interesting hypothesis to connect this to respiration that could form the basis of future studies.

At times I thought that some of the interpretation of the results was hard to follow, poorly worded, or off the mark (see comments below). The presentation of the CHiP seq data also felt incomplete, though the influence of Hsr1 and Adn2 on expression of NSF1 targets was convincing. The genome wide assays (RNA-seq, CHiP seq, screen and pull-down/mass spec) could have done with replicates which would have improved statistics and reliability of the results presented for those experiments, although for key messages, the authors followed up with convincing targeted experiments.

The study represents an advance on recent work in NCR in fission yeast in linking this with the broad metabolic switch between fermentation and respiration, and in that sense makes this of interest to a broader swathe of the microbiology community, outside those interested in metabolic regulation in microbes. In addition to being of interest to applied researchers interested in producing metabolites with yeast and other microbes, the link to cell signaling and, via flocculation and adhesion genes, to microbial multicellular-like phenotypes would make this work of interest to those interested in microbial communities.

3. How much time do you estimate the authors will need to complete the suggested revisions:

Estimated time to Complete Revisions (Required)

(Decision Recommendation)

Between 1 and 3 months

Yes

Revision Plan

Manuscript number: RC-2023-02313

Corresponding author(s): Marc Bühler

1. General Statements [optional]

We thank the three reviewers for taking the time to read our manuscript carefully and providing us with very useful feedback to improve clarity. We are also grateful for the few suggested additional experiments which we are currently performing. Finally, we are pleased they all acknowledge the importance of our work – thank you!

Below we provide a detailed response to the comments that we have received. Many of the revisions we have incorporated in the provisional revision of the manuscript that we are transferring. There is a few experiments that are currently performing and will include in the final revision if the outcome will be positive/conclusive.

2. Description of the planned revisions

REVIEWER 1

MAJOR COMMENTS

5) Fig. 1G: Addition of NSF can enhance oxygen consumption at any cell density? And in prototrophs? And without NCR? Add in figure legend that this has been done at OD600 of 0.01. -> We performed OCR measurement at higher cell-density. We observe no difference between with and without NSF treatment, which means that the effect of NSF is most prominent at low cell density. This consistent with what we already discussed in the manuscript (see Figure 3D). For the final revision of the paper, we will do this experiment also with prototrophs.

10) Line 287: ‘...it is tempting to speculate that Ayr1 dampens adaptive responses by metabolizing NSF’. Calculating MEC for NSF in delta ayr1 and in cells over-expressing Ayr1 would be required to confirm this speculation. According to Pombase, cells lacking Ayr1 have their respiratory functions compromised (no growth in galactose, glycerol...), why is so? The opposite should be expected, if NSF-mediated respiration is enhanced in this background. -> We are purposely speculating here, because much more work will be needed to firmly show that Ayr1 metabolizes NSF (including in vitro biochemical experiments). Because this would go beyond the scope of this study, we would like to leave this speculative. Nevertheless, we checked MEC through a growth assay using the ayr1Δ mutant, but no decrease in MEC was observed with the ayr1Δ mutation. It is possible that the result could be attributed to the impact of respiratory deficiency, as indicated by the growth delay in glycerol or galactose conditions on Pombase. For the final revision of the paper, we will conduct the suggested experiment of overexpressing Ayr1. Should that affect growth as expected, we will be happy to include this in the final revision.

Revision Plan

REVIEWER 2

MAJOR COMMENTS

Major comments:

Overall, the manuscript lacks readability and coherence. Quite a lot of genes/TFs and proteins are mentioned, it is difficult to find a coherent story and clear overview and connection between these subparts. The manuscript would benefit from a general proposed scheme/working mechanism in the discussion and streamlining the results and data into a single biological storyline.

-> We will be happy to provide a working model in the final revision of the paper.

284-288: Fig5E: To test whether AYR1 is indeed metabolizing NSF (and thus supporting this statement), an overexpression strain of AYR1 could be made to see if it grows on the EMM + NH₄Cl without NSF added.

-> As mentioned above, we will be doing this experiment. If successful, we will be happy to include this in the final revision.

REVIEWER 3

MINOR COMMENTS

m13: In Fig 4B and Fig S2A, there are grey and colored tracks for the chip-seq (- and + NSF), but they are very difficult to see. If grey is in front it is hard to tell how close the colored peak when the colored peak is lower. For example, grey is in front for pex7 while color is in front for yhb1. Could the authors add some transparency so that the data for both conditions could be seen at once?

Also there is little information on the control. My assumption for the input(ChIP) sample was that it was cross-linked and sonicated but not immunoprecipitated, but it is not clear what conditions it was in. I would assume it was done without NSF treatment in WT cells, but those details should be added in the caption or methods. In particular, in the input there is a large spike for Gsf2. Do the authors have any explanation for this and does it have anything to do with that gene's NSF responsiveness?

-> The cultivation conditions and information about the input samples used for ChIP-seq are described in page 26. Additionally, the input track shown in Figure S2A represents the input sample from Adn2-FLAG under NSF treatment for reference. We have confirmed that there are no significant changes in the tracks between different inputs at the specific gene loci of interest. Moreover, each peak was called using the respective conditions and the input from the corresponding strain.

For the final revision, we will try to make a new figure panel with transparent tracks as suggested.

m14: The authors might consider putting something like Fig S2B (or even a corresponding volcano plot) as a main figure for Fig 4 in addition to the other two panels, as the individual

Revision Plan

examples from fig 4B are nice to see, but do not give a broad overview of the data.

-> We will consider this when assembling the final revision.

3. Description of the revisions that have already been incorporated in the transferred manuscript

REVIEWER 1

MAJOR COMMENTS

1) Consistency with the numbers of mutants/genes should be improved. Line 119: 203 genes, 206 in 4th datasheet of Table S1; line 139: 117 mutants but 119 analyzed for GO analysis (1st datasheet in Table S2); lines 179 and 181: where are these numbers (92 down and 156 up) coming from? (compare with 74 and 98 in line 167) (maybe they come from the merge of 2, 4 and 6 h, but it is not indicated).

-> We have made corrections to Table S1, changing it to 203 mutants. We have repeated the GO analysis for the 117 genes, and the results are now updated in Figure 1C.

The initial GO analysis of the RNA-seq data was done for all time points combined. In response to this comment, we have now focused the RNA-seq-based GO analysis on the 2h RNA-seq time point. Analysis was performed for 72 protein-coding genes among the 98 up-regulated genes (which includes non-coding genes) and 38 protein-coding genes among the 74 down-regulated genes (which includes non-coding genes). The new data has been incorporated into Table S2 and Figure 2.

2) Lists of genes up- and down-regulated from the RNA seq data should be provided. GO terms are not useful. Add supplementary table, please.

-> We have attached up- and down-regulated gene lists to Table S2.

3) Comparing the transcriptomic response to that of Malecki et al 2020 in response to Antimycin A (EMBO Rep. 21:e50845) would be useful.

-> We are uncertain why this would be useful. Moreover, in the transcriptome analysis by Malecki et al. 2020, there is no direct comparison between antiA-treated and untreated samples from the same time points. Nevertheless, we compared the up- or down-regulated genes identified in Malecki et al. 2016 under antiA treatment with those under NSF treatment. This did not reveal any significant differences.

4) The optical densities and whether NCR has been induced has to be clearly specified in each experiment. For instance, RNA seq data. Line 165: for the transcriptome experiment, NSF is added or not to low density cells (not indicated in results, figure legend nor materials and methods). Should addition of NSF to wild-type strain en MM trigger the same transcriptomic changes?

-> Thank you for pointing this out. We have added 'at low cell density (OD 0.01)' to the legend

Revision Plan

for RNA-seq at page 32. Additionally, we have added the qRT-PCR results for prototrophs in new Figure 3C and mentioned in page 11.

6) Fig. 1 E and F: why 14 d after growth there is not growth at 2% glucose in panel F, but it is 10 d after in panel E? What is EMM Biomedical?

-> see response to reviewer 3, minor comment 5.

Note that sometimes we observe NSF-independent adaptive growth when cells are cultured for a very long time. To avoid confusion, we have chosen not to discuss this NSF-independent adaptive growth no longer show the longer time points. We have modified Figure 1E accordingly.

7) Fig. 2BC: Venn diagrams should be more useful to demonstrate overlap with the Malecki data.

-> We have created the venn diagrams. We have put them into supplementary figure S1B and mentioned on page 9.

8) Fig. 4A: not very useful

-> We find it useful, as it displays general trends in TF occupancy.

9) Is Hsr1 required for some of the RNA seq changes upon NSF addition? Same with other TFs

-> We put results of NSF-responsive gene expression in *hsr1Δ*, *adn2Δ*, *php3Δ*, *php3Δhsr1Δ* mutants in Figure 3C and mentioned in page 14.

11) Regarding the two pull-down experiments, one to identify Ayr1 and the second Hmt2, why different negative controls are used? Is addition of NSF to WCE prior to pull-down also used in the second experiment (with delta ayr1 and AlkOle)?

-> We do not understand the critique. Both pull-down experiments were done with NSF as competitor.

The confusion may have occurred when comparing associations of AlkNSF with Hmt2, Gst3, and Ayr1 (Figure 5H). Note that we did not conduct a competition; instead, we compared the iBAQ values from pulldown experiments using alkNSF and alkOle.

12) The data regarding Hmt2 is very interesting. As for delta ayr1, delta hmt2 cells cannot grow in glycerol nor galactose according to Pombase. Is the result shown in Fig. 5J (lack of NSF-dependent activation of *nrt1* and *mei2* in delta hmt2) a consequence of the absence of the NSF receptor, or is it due to the lack of respiration of this background? Is delta hmt2 really auxotrophic for Cys? Why? In this background, H₂S should be enhanced, and Cys and Met biosynthesis improved. In fact, in one manuscript these cells grow fine in SG minimal media (Mol Microbiol 01 42:29), while another report indicates they are auxotrophic for Cys (Genes to Cells 2016, 21:530).

-> As an additional experiment, we confirmed the responsiveness of *nrt1* (*nrg1*) in the mutant strains with disrupted ETC-related genes, which are required for NSF-dependent adaptive growth. Despite the disruption of these genes, *nrt1*(*nrg1*) still exhibited responsiveness,

Revision Plan

suggesting that the effect is not solely due to the lack of the respiratory chain. We would be happy to show this data in the final revision of the paper, if the editor would consider it useful. Additionally, the cysteine auxotrophy of the strains was confirmed in Figure S3E. In a study by Mol Microbiol 01 42:29, it appears that *hmt2Δ* was tested only in WT on YG media using SG. In a review from FEMS Yeast Res. 2021 21(5) (PMID: 34279603), an alternative biosynthetic pathway for cysteine synthesis without involving Hmt2 is suggested. However, in Redox Biol. 2021 47:102169 (PMID: 34688157), a model proposing the importance of GSSH synthesis through Hmt2 in cysteine synthesis is presented. We believe that this model may explain why *hmt2Δ* becomes a cysteine auxotroph.

13) M&M: regarding RNA isolation and sequencing: add info about OD of cultures, genotype (*leu1-32?*), growth media; also, number of replicates and filtering (fold-change used, Q value...) -> We have added the respective information in the Material and Methods section (page 24)

14) M&M, ChIP seq: same as above. Also, MACS2 can be used for the unbiased identification of bona fide TF targets, by using a quantification tool reporting percentage of occupancy upstream the TSS (callpeak function).

We have added the respective information in the Material and Methods section (page 26). The quantification of TF binding levels in this analysis is performed using MACS2, as described in the Materials and Methods section.

MINOR COMMENTS

1) Who triggers NCR? Analysis of 137 genes in Figure 1b.

-> In *Saccharomyces cerevisiae*, NCR is regulated through homologues such as Gln3 and Ure2. However, equivalent homologues have not yet been identified in *Schizosaccharomyces pombe*. Therefore, the list of 137 genes identified in our screening is of great interest for future studies in understanding these regulatory mechanisms. Additionally, since this gene list may include factors related not only to NCR but also to the inhibition of respiration, we plan to explore these aspects in our future research. However, we have not discussed these topics in the current paper as they deviate from the main focus.

2) Synthesis of NSF: how is it regulated, where does it come from?

-> As for the synthesis, it is not yet well understood, and we are currently conducting analyses using non-essential libraries to identify the necessary factors.

3) NCR impairs import of BCAA. How are the aa importers such as Cat1 or Agp3 eliminated from the plasma membrane? Transporter internalization, degradation, transcriptional repression... And how does NSF block the NCR regarding aa uptake? Or aa usage?

-> As mentioned earlier, there are many unresolved aspects of the mechanism of NCR in fission yeast, so it is not possible to comment on how aa transporters are regulated. However, we hope to gain insights into this by analyzing the list of NCR-related genes mentioned earlier.

Revision Plan

4) How can enhanced respiration by NSF counteract all of the above? How can now leu1-32 cells grow?

-> This is a difficult question that we can't fully answer. As discussed in lines 398-400 of the discussion, in *S. cerevisiae*, the increased uptake of leucine as TCA cycle intermediates during glucose derepression suggests a correlation between respiration activity and BCAA uptake. In our RNA-seq results, the only gene predicted to be up-regulated as a putative aa transporter is *nrt1(nrg1)*, and overexpression of *nrt1(nrg1)* enhanced growth under NCR conditions in our preliminary data. However, since NSF-mediated adaptive growth is not abolished in *nrt1Δ(nrg1Δ)*, the involvement of other aa transporters is suggested. Therefore, the question will have to be addressed carefully in future studies.

5) Addition of NSF to any cell type would do the same, enhance respiratory rates? With or without previous NCR? Should this signaling molecule also drive different respiratory rates in a cell density-independent manner regarding glucose catabolite repression?

-> These are good questions, but we think they are somewhat irrelevant for the main conclusions of the paper. Also, it is unclear what is meant by "cell type".

REVIEWER 2

MAJOR COMMENTS

Major comments:

Several statements or results are not sufficiently clear, elaborated or nuanced. The paper would benefit from more explanation and discussion.

In the discussion section, the authors are not consistently referencing figures.

179-181: 'GO enrichment analysis of the 92 downregulated genes' but on line 167, it is '74 downregulated genes' that are mentioned. It is unclear where this difference in number of downregulated genes comes from. Similarly, for the upregulated genes. '156 genes' are mentioned on line 181 but only '98' on line 167.

-> Thank you for pointing this out. This was also pointed out by reviewer 1. As describe above, we performed GO analysis for all up- or down-regulated genes at 2h, 4h, and 6h, resulting in variability in the numbers. To avoid confusion, we used only the 2h RNA-seq data for GO analysis, analyzing 72 protein-coding genes among the up-regulated 98 genes and 38 protein-coding genes among the down-regulated 74 genes. We have reflected these results in Table S2 and Figure 2B.

189: The statement that the downregulation of flocculation could serve to avoid mating, though sounding logical, is undermined by the finding that mating-related processes are upregulated in the experiment. I find this statement rather speculative

-> That was a good point. As mating-related GO terms were not enriched in the RNA-seq results at 2h, we no longer mention mating and instead focus on the reduced expression of genes related to cell aggregation mediated by NSF in page 10.

Revision Plan

247-249: The statement is too broad, the effects are visible for maybe 3 TFs, the others don't seem to make a difference in occupancy. Also, why are these two highlighted genes of importance (*pex7+*, *yhb1+*), this is the first and last time they are mentioned?

-> It is correct that occupancy for most TFs that we tested is not different. Changes in gene expression are easy to explain by TF occupancy, which we observe for a few. For this reason, these TF were highlighted. We mention this more specifically in the revised version of the manuscript (page 14).

We used *pex7* and *yhb1* as examples of a significant change in the binding level of transcription factors that correlated with changes in gene (see page 13).

254-256: The statement that these TFs are indispensable may be too strong. Right before, the authors showed that most of these TFs don't change occupancy (and especially Fil1 and Reb1 do not show a correlation with up- and down-regulated genes, nor does Fil1 in FigS2B show a changed ChIP-seq signal).

→ These TFs were identified in our genetic screen. That is, they are indispensable.

365: 'independently of the carbon source'. As far as we can see, all experiments were performed using glucose as the carbon source, so this statement seems too strong as there is no clear proof for this. This could be an easy extra experiment to perform these tests on media with other carbon sources than glucose?

-> We agree. We have changed "independently of the carbon source" to "without any change in the carbon source" to avoid potential misunderstandings (page 19).

Fig1E: It is not clear if the experiment was performed with the Wild-Type or a deletion strain. In the case of the WT, colonies grew in the media not containing NSF but in Fig5E and Fig5I, the WT did grow in the media not containing NSF. It could be more relevant to plate out 1 colony like in the second screen. Thus, unless different strains were used for both experiments, the results seem inconsistent with each other, which is not mentioned in the manuscript.

We have also observed NSF-independent adaptive growth by cultivating for an extended period without the addition of NSF. However, to avoid confusion, we have refrained from discussing this NSF-independent adaptive growth in this section. Consequently, we have modified the figure used for clarity (Figure 1E). Additionally, we have included details about the strains used in the Figure legend (page 32, 34, and 35). See also our responses to reviewer 1.

Fig1E, Fig5B, Fig5E, and Fig5I: For these experiments, different nitrogen concentrations were used depending on the media, but this has not been addressed/mentioned in the manuscript.

->We added the following information to Material and method in page 22. :“For nitrogen catabolite repression (NCR) condition, YES and EMM were supplemented with extra NH₄Cl (for YES; final concentration 93.5 mM. for EMM; final concentration 187 mM for liquid culture, 374 mM for normal NCR condition on solid culture, and 748 mM for severe NCR condition on solid culture) and for non-NCR condition”

Revision Plan

Minor comments:

53-57: I would like more elaboration on why CCR and NCR are important for virulence of human pathogens or relevant for industrial applications, and link back to this in the discussion.

Otherwise, it is superfluous to include this in the introduction.

-> While lines 52-54 emphasize the significance of CCR and NCR in human pathogens, underscoring their crucial role in rapidly adapting to niche environments during infection, lines 58-63 highlight their importance in industrial settings. Specifically, in the brewing industry, CCR is indispensable for efficiently synthesizing ethanol through the fermentation pathway, utilizing carbon sources. Similarly, NCR is vital for optimizing nitrogen source utilization and biomass production. However, the fine-tuning of NCR becomes paramount, as an excessive presence can lead to the accumulation of substances like urea and proline, negatively impacting the quality of products such as wine. We have change to introduction accordingly in page 3-4.

108: Does having a h^- library have any impact on the outcome compared to the original h^+ library?

-> Due to the conventionally known slight differences in growth between h^- and h^+ strains, libraries typically align with one mating type. In additional experiments, h^+ strains were occasionally used as hosts; however, as of now, no significant phenotypic differences have been observed between h^- and h^+ . In page 6, the purpose behind aligning the library with h^- is explained.

170: Why would only one of the 117 NSF-linked genes change expression in the RNA-sequencing experiment? Any explanation as to why the expression remains unchanged for the 116 other NSF-linked genes?

-> Because the expression of all other genes is not influenced by NSF. That is, these genes are needed to respond, but they are not differentially expressed. Yet, there is a possibility that the induction of transcription for these genes may not be observable at the RNA-seq time points of 2, 4, and 6 hours, or alternatively, changes might occur at the protein level rather than at the transcriptional level. We mention this in revised manuscript (page 9).

212: Please elaborate the discussion of these results. I understand the point that at low cell densities, the cells do not produce NSF as much, and thus adding NSF induces $nrt1^+$. However, the added value of testing this in different media is unclear, especially when the results of strength of increase in $nrt1^+$ show the opposite trend for the two different media between low and high nitrogen content.

-> This revealed that transcriptional induction by NSF occurs not only under artificial conditions such as high ammonium concentration under NCR condition but also under more physiological conditions.

216: Why was the ADH1+ promoter chosen as a 'negative control'?

-> We used the *adh1* promoter, traditionally employed as a housekeeping gene in *S. pombe* and *S. cerevisiae*, for its well-established role as a constitutive promoter. To clarify its purpose, I

Revision Plan

added the following sentence in page 12: “the *adh1* promoter known as a constitutive promoter in *S. pombe* (ref, PMID: 6294096).”

385: ‘NSF would not strictly revoke NCR only, but also CCR’: the authors should try to provide experimental evidence, citation(s), or clearly state it to be a hypothesis. This comment links back to the major comment on line 365. -> While experimental results regarding CCR revocation have not been obtained, the observation that NSF induces respiration even under glucose culture conditions provides a fact suggesting that NSF can initiate respiration at least prior to CCR release. Therefore, the sentence that was pointed out has been revised to: ‘NSF could be considered a mediator not only for revoking NCR but also for inducing respiration, potentially preceding the release of CCR in carbon metabolism.’ (page 20)

405: typo: strains were validation, should be ‘validated’
-> Thank you for pointing out for typo. We corrected in page 22

644: typo: ‘+’ sign not in superscript
-> Thank you for pointing out the typo. We deleted “+”s as following Reviewer #3’s comment m7.

Fig1G: Could the differences in OCR be due to differences in growth rate or remaining glucose? It could for instance be that the culture in the control condition grew less fast, thus still having glucose and therefore still in fermentative metabolism. Showing or mentioning growth rates, nutrient concentrations could help to strengthen this finding.
-> We have been measuring the cell count before OCR measurements, and confirmed that there is no difference in growth, regardless of the presence or absence of NSF treatment for 8 hours. This information was added in page 8-9.

Fig2A: The top two ‘most’ upregulated genes (*nrt1* and *mei2*) were taken along for additional experiments. However, one gene with a significant upregulation labeled in red on the left seemingly shows stronger induction than the second gene (*mei2*). Why was this gene not taken along?
-> We added an explanation in page 11 with the following sentence stating that genes with low basal expression levels are prone to artifactual qRT-PCR measurements. Therefore, genes with expression levels exceeding 100 cpm were used as reporter genes.

Fig2C: the x-axis label is not immediately clear to the reader.
-> We modified the x-axis label to be simpler in Figure 2C. “down, no, up with glycerol versus glucose”

Fig4B: typo: ‘non treatmentt’
-> Thank you for pointing out this typo. We corrected it in Figure 4B.

Revision Plan

REVIEWER 3

MAJOR COMMENTS

1) In line 70, the authors state that "S. pombe cells rely on their own BCAA synthesis to sustain growth" when grown alongside Leucine when ammonium is supplied in the media. If prototrophs can inhibit NCR via NSFs in neighboring auxotrophic cells on the same plate, couldn't they also inhibit NCR within their own colony? How do we know that prototrophic cells grown in high quality nitrogen sources along with, say leucine, are not taking up leucine? The fact that leucine auxotrophs cannot grow in high quality nitrogen sources when leucine is present does not imply that wild type cells must use be synthesizing BCAAs rather than importing them. In a recent paper (Kamrad et al Nat. Microbiol. 2023, <https://www.nature.com/articles/s41564-022-01304-8>), it was shown that *S. cerevisiae* cells grown in lysine and in high concentrations of ammonium uptake lysine rather than synthesize it as lysine concentrations in the media are increased. I am aware via unpublished results that this is the case for Leucine as well. I would be surprised if the same isn't true in *S. pombe*. The authors should caveat or remove this assertion.

-> This is an interesting comment. Our reasoning would be the following: Given that prototrophs exhibit robust growth even under NCR conditions significantly affecting BCAA uptake control, it is reasonable to infer that they rely on their own BCAA synthesis, at least during the initial stages of adaptation. As demonstrated in Figure 3D, it is conceivable that, depending on the cell number, the synthesis of NSF by the cells increases leucine uptake even under NCR conditions. In fact, when spotting and increasing the cell number, adaptive growth is observed even in the absence of NSF, even in leucine auxotrophs. Considering the increased amino acid uptake under high ammonium conditions in *S. cerevisiae*, the revised sentence in page 4 reflects this understanding: Thus, *S. pombe* cells rely predominantly on their own BCAA synthesis for adaptation under NCR conditions. '

2) It is important for the authors to put their observation linking respiration to rescue from NCR in context with findings from a closely related study (Chiu et al 2022) which included some authors from this manuscript and which the authors cite. In that paper, it was shown that the siderophore ferrichrome can also rescue NCR in fission yeast. That paper stated "It is likely that ferrichrome increased mitochondrial activity, which enabled efficient utilization of glucose downstream of the glycolytic pathway" based on experiments in different concentrations of glucose. This evidence seems to support the link between respiration and rescue from NCR proposed by the authors of this manuscript. The authors should acknowledge this closely related and earlier work as it strengthens the case they are trying to make. They could even test if ferrichrome addition makes cells sensitive to antimycin A (as in fig 1E), but that extra experiment would be optional in my opinion.

-> The report (Chiu et al., 2022) has not demonstrated the connection between the rescue of growth under NCR and mitochondrial activity by ferrichrome. Therefore, we believe it would not be appropriate to extrapolate and strengthen the current results from that study. However, given that the enhancement of respiration under NCR conditions showed a discernible relationship with growth in the current study, we can illustrate the hypothetical mechanism from the previous research. The following sentences have been added to the discussion part page 19. " In our

Revision Plan

previous study, the addition of ferrichrome resulted in the recovery of growth under high ammonium conditions and indirectly suggested an increase in mitochondrial activity (Chiu *et al*, 2022). From the current study, a model could be proposed where ferrichrome enhances BCAA uptake under high ammonium conditions through the increase in mitochondrial activity."

3) In figure 1B for the second screen I do not understand what the photos represent. For the photos, two rows are meant to have no NH₄ and also no NSF and the label on that image makes no mention of Leucine supplementation. In the diagram there are two rows that have NH₄ and leucine and one row that has no NH₄ but does have leucine. I assume the diagram is correct and the labels on the images are incorrect.

-> Thank you for pointing out our mistake. We corrected these labels in Figure 1B.

4) It would be important for the authors to put their observation linking respiration to rescue from NCR in context with findings from Chiu *et al* 2022 which the authors cite. In that paper, it was shown that the siderophore Ferrichrome can also rescue NCR in fission yeast which the authors cite which found that a siderophore rescues NCR. Also the authors of that paper stated "It is likely that ferrichrome increased mitochondrial activity, which enabled efficient utilization of glucose downstream of the glycolytic pathway." based on experiments in different concentrations of glucose. This evidence seems to support the link between respiration and rescue from NCR proposed by the authors of this manuscript.

-> same as 2)

5) In line 133. The authors state that the 29 mutants that didn't grow under Leucine supplementation either without NH₄Cl or with NH₄Cl whether or not NSF was present were "related to EMM Growth, leucine uptake, or utilization of ammonium as the sole nitrogen source." The first two make sense, but I can't see why a strain with deletion of a gene related to utilization of ammonium as a sole nitrogen source wouldn't grow when supplemented with leucine. In fact for all the leucine auxotrophs in the screen, if one was to try to grow them with ammonium as the sole nitrogen source they would not grow, so it isn't clear that this screen can identify genes responsible for utilization of ammonium as a sole nitrogen source. The authors should clarify or remove this point.

-> Thank you for your feedback. As suggested, we have removed this point (see in page 7).

6) 203 strains are important for avoidance of NCR (because in the presence of Ammonium and Leucine, as well as a WT strain, they cannot grow). Of these 57 strains can't grow in the presence of a WT strain but they can grow in the presence of NSF. The authors conclude in line 138 that these strains are "likely to respond to a transmissible signal that is different from NSF". This is confusing because deletion of these genes still does allow cells to respond to NSF, however when these cells are growing in the presence of wild type cells (which in their model are releasing NSF), the cells don't grow. I am confused about the nature of the transmissible signal that the authors suggest. It would appear that when these genes are deleted and grown next to a wild type cell which sends the alternative signal and the NSF, the other transmissible signal would inhibit the ability of NSF to release NCR (as NSF can still rescue the gene). It is

Revision Plan

not clear how the other transmissible signal would work when the gene is present as it is clearly not necessary to rescue growth.

A simpler explanation might be that there was contamination in the second screen, or that there was a threshold effect - perhaps in the first screen the strains grew just below a threshold and in the second screen it grew just above that level.

The authors should clarify their interpretation for these strains, and acknowledge any alternative technical explanations.

-> As suggested, we believe the results may be attributed to a threshold effect. Therefore, the following sentences have been added to line page 7-8." In the 1st screen, it is speculated that these mutants fell below the threshold of NSF required for adaptive growth, while in the 2nd screen, they surpassed this threshold, leading to the manifestation of adaptive growth. While these factors may be partially involved in NSF response, detailed analysis of these factors will not be conducted in this study as the primary aim is the exploration of key factors in NSF response."

7) The authors' efforts to removed confounding effects that might stem from additional auxotrophic alleles made the screen more convincing. However, Fig 1E, 1F, 5B, and 5E were done with EMM+Leu+Ade+Ura, while the initial strain was just done in the presence of additional Leucine. It is unclear why this was done from the text and captions, but I assume it was because they used a strain that was ade- and ura- in addition to being leu-. Given that they had strains without these additional mutations, this seems like a strange choice. The authors should acknowledge that there are possible confounding effects of adding adenine and uracil to the media, and, if they did have additional metabolic deletions, acknowledge that that could possibly be confounding.

-> Since there is a possibility of using the Bioneer gene disruption library, we utilized strains with auxotrophy background leu1-32 ade6-M216 ura4-D14. There was an oversight in providing information about these strains. We have added this information to the Figure legend in page 32, and 35-36.

Considering the potential impact of auxotrophy for adenine and uracil on the results, we repeated this experiment in an *hmt2Δ* leucine auxotroph, put in Figure S3F, and mentioned in page 16-17. We also checked *hmt2Δ* prototroph did not show growth defect under NCR condition in Figure S3G and explain in page 17.

8) Fig 1E, it appears that cells can grow without NSF in the presence of ammonium and additional amino acids after 10 days (although NSF is required for growth at 5 days). This is not a problem for the screen as that was taken at 5-6 days, but it appears as though NSF does not rescue growth so much as speed it up. The authors should acknowledge this when describing the phenotype. It also argues for a quantitative time course growth experiment to compare growth over the course of 10 days with and without NSF, although this would not be necessary to the paper's main argument.

-> Page 6, we described NSF-independent adaptive growth and how growth was assessed as

Revision Plan

follows. To avoid confusion, we have refrained from discussing this NSF-independent adaptive growth in this section. Consequently, we have modified the figure used for clarity in Figure 1E.

9) In line 191 and 192, the authors suggest that the "downregulation of flocculation/adhesion related genes by NSF could serve to avoid undesirable mating during growth". If this is the case, I don't understand why mating genes and cellular fusion genes would be upregulated. What do the authors mean by undesirable mating? Wouldn't flocculation increase desirable mating as well? If all mating is undesirable, wouldn't upregulation of mating and cellular fusion genes be detrimental?

-> After reanalyzing the GO analysis specifically for up- or down-regulated genes at the earliest time point of 2 hours in the time course RNA-seq, the down-regulated genes still revealed factors related to flocculation, while the upregulated genes did not show enrichment in GO terms related to mating but exhibited enrichment in genes associated with trehalose synthesis. Therefore, the mention of 'undesirable mating' has been removed, and only the possibility of NSF inhibiting flocculation is now discussed. The corrected sentences have been added to page 10.

10) The authors mention that trehalose is an antioxidant, for which they reference Malecki 2019, however that paper shows no direct evidence of trehalose functioning as an antioxidant under respiratory conditions. It only shows that some trehalose synthesis genes are upregulated when cells are grown under glucose. The authors should identify primary literature to back this statement up, or soften the wording. Also trehalose is known to be a storage metabolite (which is mentioned in Malicki et al 2019, but not in this manuscript). In fact work in budding yeast has shown that trehalose can be a shared metabolite that can be produced by respiring cells and used as a fermentable carbon source in communities of budding yeast cells that consist of fermenting and non-fermenting cells (Varahan et al, eLife 2019 <https://doi.org/10.7554/eLife.46735>). It seems that this role should be considered as an alternative explanation for the induction of trehalose in respiratory cells.

->As per your suggestion, we have added the following sentence to page 10, referencing paper (Malicki et al., 2016) that discusses the induction of trehalose synthesis-related genes by glycerol in *S. pombe*. We also cite the paper (Moon et al., 2020) *in* to support the notion of trehalose functioning as an antioxidant: "Coherent with the induction of a respiration-like gene expression program, the trehalose synthesis pathway is also upregulated under respiratory conditions (Malecki *et al*, 2016). Furthermore, it has been reported that trehalose functions as an antioxidant in the probiotic yeast *Saccharomyces boulardii*. This suggests a potential role in counteracting ROS production associated with NSF-induced respiration."

11) Line 208: The stimulatory effect of NSF on NRT1 decreased with cell density, thus cell density is likely to be an important factor in terms of gene expression. The methods section, text and figure captions do not mention the density at which cells were inoculated/harvested for RNA-seq and other experiments. If that density was more than OD 0.1, then this would be inconsistent with the measurements from Fig 3. Also in fig 3D, The culture density is not mentioned in the figure or the caption, even though the text suggests that for that experiment

Revision Plan

cells were grown at low density (Lines 212-213). The authors should provide information on density for their experiments in order for them to be reproducible, as they show it is a key factor.

-> The information regarding cell density in RNA-seq experiments was detailed in the Materials and Methods section (see page 24), while the cell density information for qRT-PCR experiments was provided in the figure legends (see page 32-36).

12) In suggesting a name for NRT1 (NSF-responsive amino acid transporter 1), the authors assume that the gene has a role in amino acid transmembrane transport, but they have no experiments showing this phenotype. They mention that it is Inferred from homology with other amino acid transporters. I presume this name has already been approved by Pombase and is not provisional, but it seems that including phenotypes inferred from homology, rather than from experiments is unwise. Do the authors have any other direct evidence that this is a bona fide Amino Acid Transporter? Perhaps a name like "NSF-responsive gene" would be more appropriate.

We agree. We have changed the name to *Nrg1* (NSF-responsive gene).

Related to this, it appears that the expression level of *Nrt1* may be very low (see Fig S2B in which the scale of the RNA-seq track is very small [-1,1] and the amount of expression is very small even when NSF is added). Looking at Fig 2A, the total transcript abundance did not appear to be very low in terms of counts per million (over 100) is this a discrepancy in fig S2B? Perhaps the large fold change is the result of counts very close to zero in the control condition? Also in Fig 3 the *nrt1* expression levels did not appear to be especially low and they appeared repeatable. Is the RNA-seq data shown in fig S2B for *nrt1* a fluke or am I misinterpreting it? ->We have confirmed a gradual increase in the expression level of *nrt1* (*nrg1*) over time with NSF treatment. By switching from the 2h RNA-seq track to the 6h track, it became evident on the track that the transcription of *nrt1* is enhanced by NSF. The qRT-PCR experiments were performed with 6h-NSF treatment, as described.

13) To show that their Chip-seq worked, the authors showed specific examples of Chip-seq reads for target genes Line 240, "Previously determined target genes of these TFs were significantly enriched in our data set, demonstrating that the experiment has worked (Figure S2A)." Is the significance here, the threshold from fig S2B? If so that threshold should be clearly stated here in the text. If it is the fact that *asn1* shows up as "Fil1 bound" is strange as there are no genes that had significant changes in ChIP-seq signals for fig S2B. If there is another threshold the authors should describe it. While some of the examples they showed were convincing (e.g. *php3-flag* for the *php3* regulated gene *gln1* and the increased reads for *srw1* for the *reb1* target *srw1*), there were some targets that didn't seem to be especially enriched for their designated transcription factor. For example, the gene *trx1* which was identified as an Hsr1 binding target had some binding from Hsr1, but more from Php3 and equivalent amounts for many of the other transcription factors. A clear description of how genes are chosen to be significant in the text, alongside references/selection criteria the authors used to select the specific genes shown should be provided to improve reproducibility.

Revision Plan

-> The term 'significantly enriched' of our ChIP-seq data refers to peaks with a MACS2 score of 100 or higher identified using MACS2 between input and Ch-IP samples, as detailed in the Materials and Methods section (see page 26). To clarify, the following sentences have been added to page 26 for explanation: "Previously determined target genes of these TFs were significantly enriched compared to input sample in our data set, demonstrating that the experiment has worked (Figure S2A). "

The term 'significant changes' in Figure S2B pertains to the comparison between without NSF and with NSF conditions. Since the number of gray dots in Figure S2B represents peaks where Fil1 binding itself is significantly enriched compared to the input sample, we are confident that the ChIP-seq worked. The criteria for gene selection were based on information from ChIP-seq and ChIP-qPCR of TF and the transcriptional changes of target genes upon TF gene disruption. These details have been summarized in the Figure S2B legend."

14) In lines 244-246 the authors state that "These differences in TF occupancy were positively correlated with target gene expression changes. That is, individual genes that were upregulated by NSF tended to be more strongly bound by the TFs, whereas downregulated genes were less occupied by the respective TFs (Figure 4A)." This is far from a general trend. The trend is not there for reb1 and fil1. In fact fil1 looks to the eye like it shows a decrease in occupancy for genes with increased expression, and I worry that the authors did a one sided test for significance that would have missed this, although the variability of the genes that don't change in this case is very high, so there could be no significant effect. The authors elaborate on some of the detail in following statements, but they should soften or remove this statement.

Related to this, in line 254, the authors state: "These results imply that NSF exposure rewires the recipient cell's transcriptional program, for which the TFs Atf1, Adn2, Adn3, Fil1, Hsr1, Php3, Php5, and

Reb1 are indispensable (Table S3)." While I am convinced from the RNA-seq evidence and some of the chip-seq evidence that NSF exposure rewires cell's transcriptional program, I am not convinced that the 8 transcription factors they mention are indispensable for rewiring the transcriptional program. While they may be indispensable for the phenotype itself, Reb1, and Fil1 show no significant enrichment in occupancy of upregulated or downregulated targets (Fig 4A) and, along with Atf1, Reb1, and Fil1, have very few genes in which occupancy is changed significantly (Fig S2B), while no chip-seq experiments were shown for Php5 and Adn3.

The more specific summary of the data (Lines 250-253) from Fig S2B describing how hsr1 and adn2 have the strongest effects of the transcription factors required for NSF-mediated NCR bypass is a much stronger message for this section.

-> Thank you for this comment. Because it is similar to reviewer 2, see our response above.

15) In line 335, the authors state that "in contrast to other communication systems, NSF does not induce noticeable changes in *S. pombe*'s morphology", referring to changes in mating, filamentation, and bacterial biofilm formation. However they do show very clearly that NSF does

Revision Plan

cause a large decrease in expression in flocculation/adhesion genes. The fact that they do not see a change in morphology is likely due to the fact that the lab strain in the conditions used for this assay do not flocculate. We have recently identified conditions and strains which do exhibit flocculation in this preprint [<https://www.biorxiv.org/content/10.1101/2023.12.15.571870v2>]. It is likely that if they had a strain and conditions that did flocculate addition of NSF would break up flocculation and thus change the morphology based on their evidence. The authors should remove or caveat this point.

-> We have modified the explanation to suggest that NSF may represent a novel cell-to-cell communication system that not only influences the activity of respiration and transcription but also induces morphological changes associated with flocculation. page 10-11

16) Line 270 Fig 5B: The concentration of NH₄Cl listed in the text (374mM) does not match the concentration shown on the figure (748mM). I assume this is a typo but it should be corrected prior to publication.

-> Thank you for pointing out our mistake. We corrected it in page 15.

Also I have several minor comments to help improve the manuscript.

m1: Lines 66-70- state that "uptake of the branched-chain amino acids (BCAA) isoleucine (Ile), leucine (Leu), and valine (Val) is suppressed in the presence of high-quality nitrogen sources such as ammonium or glutamate, because the expression of transporters or permeases that are needed for the uptake of poorer nitrogen sources are down regulated (Zhang et al, 2018)." This reference is for *S. cerevisiae* and is a review. The authors should cite original results in *S. pombe* if possible, and if that is not available, alert the reader that this result is from a different species.

-> We added "in *S. cerevisiae*" to alert the reader in page 4.

m2: It is unclear from the methods section how the images taken for the screens were analyzed. Were they analyzed and scored by hand, or using custom image analysis software. Either way, when publishing the authors should publish the scores for each deletion mutant in their screen. If there was custom image analysis, the authors should mention in their methods the cutoffs which they used to score growth, and consider plotting the data as a supplement so readers can get a sense of how sensitive the screen was.

-> The information of judgement of growth was added in page 6

m3: The authors identify 137 mutants that did not require NSF signaling to bypass NCR and claimed these genes were required for NCR. It would be helpful and give more confidence in this screen to demonstrate the extent to which the genes identified in this study overlap with any previous genes required for NCR, and whether there was any GO-term enrichment in this set.

-> We could not find any genes and GO-term related to NCR

m5: It is unclear why Figure 1F has 'MP biomedical TM' listed in the figure. It doesn't seem to

Revision Plan

be listed in the caption or the methods. Is this different media than in other experiments? If so, the authors should add that information to the methods or the caption.

-> In page 22 of the Materials and Methods, we mentioned the use of FORMEDIUM's regular EMM containing 2% glucose, and we specified the use of glucose-free EMM media from MP Biomedical™ for the preparation of lower glucose EMM media.

m6: In Line 160, positively influenced is strange wording, do the authors mean "induced"?

-> Thank you for pointing out. We corrected it to "induced" in page 9.

m7: In the section on gene expression change upon exposure to NSF, the authors use a + after each gene name. My understanding is that that notation is meant to refer to strains with the wild type genotype of that gene, and not the gene itself. Shouldn't the gene be italicised in lower case to represent the gene? See: Lera-Ramirez et al

2023 <https://doi.org/10.1093/genetics/iyad143>.

-> We have corrected these.

m8: In Fig 2A, genes are displayed on a plot that depicts level vs log2FC, but a comparison between the fold change and p-value would be more useful, and I believe DESeq2 should provide an adjusted p-value for these genes. A related issue is that it appears as though there were no biological replicates, though there was data gathered at different time points. In these genome wide experiments, replicates can give confidence to data and help distinguish true change from intrinsic variability of expression in specific genes. Though the authors did qPCR to validate specific results, it would have improved the quality of their systems-level data to have replicates for these and other key experiments (ChIP-seq, affinity purification and even the screen).

-> We have employed MA plots to intuitively represent the basal expression levels of each gene for the selection of NSF-responsive reporter genes instead of volcano plot (y-axis is adjusted-p value). Each analysis primarily used only one strain (clone). However, RNA-seq cultivation was conducted in triplicate, consisting of three different flasks. Additionally, we checked the reproducibility, and the results were consistent with the conclusions of this paper. These results are presented in Figure S1D and E. For ChIP-seq, although we used 3 L culture with one flask for each condition due to lower cell density, we conducted technical replicates in triplicate for each condition. Furthermore, as a follow-up experiment, we generated gene knockout strains for important transcription factors, assessed their impact on transcription through qRT-PCR (Figure 4C), and complemented the results with ChIP-seq. In affinity purification experiments, we conducted technical triplicates for each experiment and performed reproducibility experiments. We have added this information about replicates for RNA-seq in page 24, ChIP-seq in page 26, and affinity purification in page 28.

m9: Supp Fig S1: To show that similar gene expression profiles exist for other time points, it would be more convincing to show Log fold change 2h vs 4h and 2h vs 6h and show correlation, or else to make a heat map with all genes to see that genes that go up in one condition go up in

Revision Plan

the other conditions. It is not clear if the red and blue colors are defined for the 2h dataset and then mapped onto the 4 and 6h dataset, or if they are independently assigned for each plot. -> We confirmed a positive correlation through the creation of a scatter plot; however, it was observed to be exceedingly weak. Therefore, in the main text, it is described that there is no significant alteration in the distribution of NSF-linked genes, rather than changes in the overall differential gene expression in page 9. Additionally, it was verified that a respiration-like gene expression program is triggered similarly at 4h and 6h, as observed at 2h, and this information has been incorporated into page 9 and put in Figure S1C.

m10: Mbx2 is a key transcription factor related to flocculation and adhesion genes, and its expression is correlated with expression of its targets. If this transcription factor's expression levels decreased in response to NSF, that might strengthen and help explain the decrease in expression the authors observe in flocculation/adhesion genes when cells encounter NSF. If it does not change, it might also be interesting for readers interested in these phenotypes. -> That is another interesting point. The transcription level of *mbx2* showed a slight decrease with a log₂FC of -0.44. However, the absence of a growth phenotype in the *mbx2*Δ strain in the growth screening suggests a low causative relationship between flocculation and NCR. Nevertheless, the regulation of *gsf2* and *pfl3* expression by Hsr1 and Adn2 is considered crucial insight into the control of flocculation-related genes. We have added the following sentences to address this in page 14." The gene expression regulation of *gsf2* and *pfl3* involves both positive control by Mbx2 and negative control by Rfl1 (Kwon *et al*, 2012). Our findings further reveal that Hsr1 and Adn2 bind to the promoters of *gsf2* and *pfl3* (Figure S2A), implicating them as factors involved in the control of gene expression. Given that Hsr1 is known to respond to hydrogen peroxide (Chen *et al.*, 2008), this presents a new opportunity to investigate the association between oxidative stress and cell adhesion." Hopefully this will be useful for readers interested in flocculation and adhesion phenotypes!

m11: In Fig 3D, The notation for the Ammonium concentrations for EMM and YES are inconsistent (+ vs parentheses), also the units (mM from the caption) are not on the figure, but the abbreviation "N" is which is confusing and inconsistent with the other plots in which NH₄CL is not abbreviated. Additionally, the caption lists additional nutrients in the media for the EMM conditions (Leu, Ade, Ura) which ought to also be listed.

-> thank you for pointing this out. We corrected labels in Figure 3E.

m12: In lines 233-235, the authors say "One possibility is that they remain bound to their target genes but become activated or deactivated by NSF directly, or posttranslational modification, such as phosphorylation in the case of Atf1". I don't think the authors intend this, but this sentence could be taken to mean that Atf1 has been shown to be phosphorylated by NSF in the reference they site. I think the authors should clarify, i.e. by saying "...such as phosphorylation which is known to regulate activity of Aft1 in response to oxidative and osmotic stress [Lawrence *et al* 2009]".

-> Thank you for pointing this out. We agree and have amended the text in page 13 as suggested here.

m15: In line 348, the wording "Would score" might be better replaced by "would be identified."
-> Thank you for pointing out. We corrected in line 395

4. Description of analyses that authors prefer not to carry out

REVIEWER 3 MINOR COMMENTS

m4: It would be interesting if the authors could speculate a bit in their discussion on why mitochondrial respiration counteracts NCR. Is there something about cells undergoing respiration that would make it easier for them to use BCAAs than to produce them, or conversely something about fermenting cells that makes it easier for them to produce BCAAs rather than importing them?

-> it is an interesting but also a difficult question to address as we mentioned in our response to Reviewer #1's minor comments 4. In page 19 of the initial submission, we mention that it has been demonstrated in *S. cerevisiae* that induction of respiration by glucose de-repression leads to an increase in leucine uptake. However, whether this is due to transcriptional regulation of transporters or regulation of transporter activity remains unclear. In our study, among the up-regulated genes identified by RNA-seq, the only gene encoding a putative amino acid transporter was *nrt1*. Given that overexpression of *nrt1* (*nrg1*) promotes growth under NCR conditions in our preliminary data, it is suggested that Hmt2-mediated regulation of *nrt1* (*nrg1*) expression is one mechanism. Additionally, as indicated in page 9, the time points for RNA-seq are relatively early in comparison to the growth rate, so it is anticipated that further regulation of amino acid transporters may occur at later time points. We believe that transcriptional profiling will be valuable for exploring this aspect. However, as the culture progresses, the effect of NSF diminishes due to the increase in cell density associated with cell proliferation. Therefore, for this experiment, it will be necessary to use strains incapable of NSF synthesis. Because we do not have such strains, we are not in a position to perform such experiments.

Dear Dr Bühler,

Thank you for transferring your manuscript with Review Commons referee reports and responses to The EMBO Journal.

Given the referees' positive recommendations, I would like to invite you to submit a revised version of the manuscript, addressing the comments of all three reviewers along the lines sketched in your response letter. I should add that it is EMBO Journal policy to allow only a single round of revision, and acceptance of your manuscript will therefore depend on the completeness of your responses in this revised version.

When preparing your letter of response to the referees' comments, please bear in mind that this will form part of the Review Process File, and will therefore be available online to the community. For more details on our Transparent Editorial Process, please visit our website.

Thank you for the opportunity to consider your work for publication. I look forward to your revision.

Kind regards,

Daniel Klimmeck

Daniel Klimmeck, PhD
Senior Editor
The EMBO Journal

We realize that it is difficult to revise to a specific deadline. In the interest of protecting the conceptual advance provided by the work, we recommend a revision within 3 months (10th Jun 2024). Please discuss the revision progress ahead of this time with the editor if you require more time to complete the revisions. Use the link below to submit your revision:

Link Not Available

Rev_Com_number: RC-2023-02313

New_manu_number: EMBOJ-2024-117143-T

Corr_author: Bühler

Title: Nitrogen signaling factor triggers a respiration-like gene expression program

Manuscript number: RC-2023-02313

Corresponding author(s): Yoko Yashiroda, Marc Bühler

1. General Statements [optional]

We extend our sincere gratitude to the reviewers and editors for their valuable feedback and guidance throughout the revisions process.

Reviewer #1 (Evidence, reproducibility and clarity (Required)):

SUMMARY

We have now reviewed the manuscript by the groups of Dr. Bühler and Dr Yashiroda entitled: “Nitrogen signaling factor triggers a respiration-like gene expression program”. We have enjoyed the topic, the experiments and the science behind it all. The authors study here one part of the fission yeast ‘quorum sensing’-like mechanism of counteracting NCR, mediated by the small molecule NSF: they identify pathways required to respond to NSF, and more specifically determine the mechanism by which NSF counteracts NCR: triggering respiration. This is a very interesting manuscript, with nicely executed experiments, and the topic is of great interest. Regarding the major comments, they are specific to the current data. The minor comments are questions raised in light of the present set of data, which should be appropriate for future research and future manuscripts.

MAJOR COMMENTS

1) Consistency with the numbers of mutants/genes should be improved. Line 119: 203 genes, 206 in 4th datasheet of Table S1; line 139: 117 mutants but 119 analyzed for GO analysis (1st datasheet in Table S2); lines 179 and 181: where are these numbers (92 down and 156 up) coming from? (compare with 74 and 98 in line 167) (maybe they come from the merge of 2, 4 and 6 h, but it is not indicated).

-> Thank you for pointing out these inconsistencies. We have made corrections to Appendix Table S1, changing it to 203 mutants. We have repeated the GO analysis for the 117 genes and updated the results in Figure 1C.

The initial GO analysis of the RNA-seq data was done for all time points combined. In response to this comment, we have focused the RNA-seq-based GO analysis on the 2h RNA-seq time point. Analysis was performed for 72 protein-coding genes among the 99 up-regulated genes (which includes non-coding genes) and 38 protein-coding genes among the 75 down-regulated genes (which includes non-coding genes). We incorporated the reanalysis of the data in Appendix Table S2 and Figure 2.

2) Lists of genes up- and down-regulated from the RNA seq data should be provided. GO terms are not useful. Add supplementary table, please.

-> We have attached up- and down-regulated gene lists to Appendix Table S2.

3) Comparing the transcriptomic response to that of Malecki et al 2020 in response to Antimycin A (EMBO Rep. 21:e50845) would be useful.

-> We are uncertain why this would be useful. Moreover, in the transcriptome analysis by Malecki et al. 2020, there is no direct comparison between antiA-treated and untreated samples from the same time points. Nevertheless, we compared the up- or down-regulated genes identified in Malecki et al. 2016 under antiA treatment with those under NSF treatment. This did not reveal any significant differences.

4) The optical densities and whether NCR has been induced has to be clearly specified in each experiment. For instance, RNA seq data. Line 165: for the transcriptome experiment, NSF is added or not to low density cells (not indicated in results, figure legend nor materials and methods). Should addition of NSF to wild-type strain en MM trigger the same transcriptomic changes?

-> Thank you for pointing this out. We have added this information ("at low cell density (OD 0.01)") to the figure legend of the RNA-seq experiments (page 32). Additionally, we have added qRT-PCR results for prototrophs in new Figure 3C and mentioned it on page 11.

5) Fig. 1G: Addition of NSF can enhance oxygen consumption at any cell density? And in prototrophs? And without NCR? Add in figure legend that this has been done at OD600 of 0.01.

-> We performed OCR measurement at higher cell-density and did not observe differences between NSF treated and non-treated cells. I.e., the effect of NSF is most prominent at low cell density. This is consistent with what we already discussed in the manuscript (see Figure 3D). We also assessed OCR in prototrophs. Like in auxotrophs, NSF treatment increased OCR (we do not show this data for simplicity).

6) Fig. 1 E and F: why 14 d after growth there is not growth at 2% glucose in panel F, but it is 10 d after in panel E? What is EMM Biomedical?

-> see response to reviewer 3, minor comment 5.

Note that sometimes we observe NSF-independent adaptive growth when cells are cultured for a very long time. To avoid confusion, we have chosen not to discuss this NSF-independent adaptive growth and no longer show the longer time points. We have modified Figure 1E accordingly.

7) Fig. 2BC: Venn diagrams should be more useful to demonstrate overlap withn the Malecki data.

-> We have added Venn diagrams to supplementary Figure EV1C and mention it on page 9.

8) Fig. 4A: not very useful

-> We find it useful because it displays general trends in TF occupancy.

9) Is Hsr1 required for some of the RNA seq changes upon NSF addition? Same with other TFs
-> Yes, this is the case for individual genes that we have checked by qRT-PCR (both up and down-regulation by NSF addition), but we can't generalize because we do not have RNA-seq data from TF knock-out strains. For example, the *nrg1* gene is no longer induced by NSF in the absence of Hsr1; or in absence of Hsr1, Adn2, or Php3, *gsf2* and *pfl3* expression is as low as after treating wt cells with NSF (i.e. Hsr1 and Adn2 are required for the expression of the *gsf2* and *pfl3* genes). We show decreased *gsf2* and *pfl3* expression levels in *hsr1Δ*, *adn2Δ*, or *php3Δ* mutants in Figure 4D.

10) Line 287: '...it is tempting to speculate that Ayr1 dampens adaptive responses by metabolizing NSF'. Calculating MEC for NSF in delta *ayr1* and in cells over-expressing Ayr1 would be required to confirm this speculation. According to Pombase, cells lacking Ayr1 have their respiratory functions compromised (no growth in galactose, glycerol...), why is so? The opposite should be expected, if NSF-mediated respiration is enhanced in this background.

-> We are purposely speculating here, because much more work will be needed to firmly show that Ayr1 metabolizes NSF (including in vitro biochemical experiments). Because this would go beyond the scope of this study, we would like to leave this speculative.

Nevertheless, we performed the suggested experiments. We checked MEC through a growth assay using the *ayr1Δ* mutant, but no decrease in MEC was observed with the *ayr1Δ* mutation. It is possible that the result could be attributed to the impact of respiratory deficiency, as indicated by the growth delay in glycerol or galactose conditions on Pombase. Overexpression of Ayr1 did indeed affect NSF-mediated growth and NSF-responsive gene expression, fueling our speculation. We show these results in Figure 5F and 5G and mention them on page 17.

11) Regarding the two pull-down experiments, one to identify Ayr1 and the second Hmt2, why different negative controls are used? Is addition of NSF to WCE prior to pull-down also used in the second experiment (with delta *ayr1* and AlkOle)?

-> We do not understand the critique. Both pull-down experiments were done with NSF as competitor. The confusion may have occurred when comparing associations of AlkNSF with Hmt2, Gst3, and Ayr1 (Figure 5I). Note that we did not conduct a competition here; instead, we compared the iBAQ values from pulldown experiments using alkNSF and alkOle.

12) The data regarding Hmt2 is very interesting. As for delta *ayr1*, delta *hmt2* cells cannot grow in glycerol nor galactose according to Pombase. Is the result shown in Fig. 5J (lack of NSF-dependent activation of *nrt1* and *mei2* in delta *hmt2*) a consequence of the absence of the NSF receptor, or is it due to the lack of respiration of this background? Is delta *hmt2* really auxotrophic for Cys? Why? In this background, H₂S should be enhanced, and Cys and Met biosynthesis improved. In fact, in one manuscript these cells grow fine in SG minimal media (Mol Microbiol 01 42:29), while another report indicates they are auxotrophic for Cys (Genes to Cells 2016, 21:530).

-> As an additional experiment, we confirmed the responsiveness of *nrt1* (*nrg1*) in the mutant strains with disrupted ETC-related genes, which are required for NSF-dependent adaptive growth. Despite the disruption of these genes, *nrt1(nrg1)* still exhibited responsiveness, suggesting that the effect is not solely due to the lack of the respiratory chain.

Additionally, the cysteine auxotrophy of the strains was confirmed in Figure EV3E. In the study mentioned (Mol Microbiol 01 42:29), *hmt2Δ* was tested only in WT on YG media using SG. In a review from FEMS Yeast Res. 2021 21(5) (PMID: 34279603), an alternative biosynthetic pathway for cysteine synthesis without involving Hmt2 is suggested. However, in Redox Biol. 2021 47:102169 (PMID: 34688157), a model proposing the importance of GSSH synthesis through Hmt2 in cysteine synthesis is presented. We believe that this model may explain why *hmt2Δ* becomes a cysteine auxotroph.

13) M&M: regarding RNA isolation and sequencing: add info about OD of cultures, genotype (*leu1-32?*), growth media; also, number of replicates and filtering (fold-change used, Q value...)

-> We have added the respective information in the Material and Methods section (page 24).

14) M&M, ChIP seq: same as above. Also, MACS2 can be used for the unbiased identification of bona fide TF targets, by using a quantification tool reporting percentage of occupancy upstream the TSS (callpeak function).

We have added the respective information to the Material and Methods section (page 26). The quantification of TF binding levels in this analysis was performed using MACS2, as described in the Materials and Methods section.

MINOR COMMENTS

1) Who triggers NCR? Analysis of 137 genes in Figure 1b.

-> In *Saccharomyces cerevisiae*, NCR is regulated through homologues such as Gln3 and Ure2. However, equivalent homologues have not yet been identified in *Schizosaccharomyces pombe*. Therefore, the list of 137 genes identified in our screen will be valuable for future studies investigating these regulatory mechanisms. Additionally, since this gene list may include factors related not only to NCR but also to the inhibition of respiration, we plan to explore these aspects in our future research. Because these topics deviate from the main focus of our work, we are not discussing this in the paper.

2) Synthesis of NSF: how is it regulated, where does it come from?

-> This is a great question, addressing of which is work in progress.

3) NCR impairs import of BCAA. How are the aa importers such as Cat1 or Agp3 eliminated from the plasma membrane? Transporter internalization, degradation, transcriptional repression... And how does NSF block the NCR regarding aa uptake? Or aa usage?

-> As mentioned above, there are many unresolved aspects of the mechanism of NCR in fission yeast, so it is not possible to comment on how aa transporters are regulated. However, we hope to gain insights into this by analyzing the list of NCR-related genes mentioned earlier.

4) How can enhanced respiration by NSF counteract all of the above? How can now leu1-32 cells grow?

-> This is a difficult question that we can't fully answer. As discussed in lines 398-400 of the discussion, in *S. cerevisiae*, the increased uptake of leucine as TCA cycle intermediates during glucose derepression suggests a correlation between respiration activity and BCAA uptake. In our RNA-seq results, the only gene predicted to be up-regulated as a putative aa transporter is *nrt1(nrg1)*, and overexpression of *nrt1 (nrg1)* enhanced growth under NCR conditions in our preliminary data. However, since NSF-mediated adaptive growth is not abolished in *nrt1Δ (nrg1Δ)*, the involvement of other aa transporters is suggested. Therefore, the question will have to be addressed carefully in future studies.

5) Addition of NSF to any cell type would do the same, enhance respiratory rates? With or without previous NCR? Should this signaling molecule also drive different respiratory rates in a cell density-independent manner regarding glucose catabolite repression?

-> These are good questions, but we think they are somewhat irrelevant for the main conclusions of the paper. Also, it is unclear what is meant by "cell type".

Reviewer #1 (Significance (Required)):

Within this manuscript, the authors study a cell-to-cell communication process, by which nitrogen catabolite repression can be counteracted by a small molecule called NSF. Specifically, the authors demonstrate here that NSF up-regulates respiratory metabolism as a mechanism to overcome the repression of amino acid internalization, which was blocked by excess nitrogen. This is a wonderful manuscript, with splendid data, on a very interesting topic.

Thank you!

Reviewer #2 (Evidence, reproducibility and clarity (Required)):

This paper uses the model system *Schizosaccharomyces pombe* to investigate how the oxylipin nitrogen signaling factor functions to send signals and adapt the metabolism upon a change in nutrients in the environment. Combining genome-wide screens, RNA-sequencing and chemical biology, the authors find that the nitrogen signaling factor triggers a change from fermentation to a respiratory metabolism, through a direct interaction with a mitochondrial oxidoreductase Hmt2.

Major comments:

Overall, the manuscript lacks readability and coherence. Quite a lot of genes/TFs and proteins are mentioned, it is difficult to find a coherent story and clear overview and connection between these subparts. The manuscript would benefit from a general proposed scheme/working mechanism in the discussion and streamlining the results and data into a single biological storyline.

-> We value this feedback and have taken steps to enhance the readability of our manuscript. Additionally, we have included a cartoon illustrating a working model, which we hope readers will find helpful (see Supplementary Figure EV4).

Several statements or results are not sufficiently clear, elaborated or nuanced. The paper would benefit from more explanation and discussion.

In the discussion section, the authors are not consistently referencing figures.

179-181: 'GO enrichment analysis of the 92 downregulated genes' but on line 167, it is '74 downregulated genes' that are mentioned. It is unclear where this difference in number of downregulated genes comes from. Similarly, for the upregulated genes. '156 genes' are mentioned on line 181 but only '98' on line 167.

-> Thank you for pointing this out. This was also pointed out by reviewer 1. As describe above, we performed GO analysis for all up- or down-regulated genes at 2h, 4h, and 6h, resulting in variability in the numbers. To avoid confusion, we used the 2h RNA-seq data for a GO re-analysis (analyzing 72 protein-coding genes among the up-regulated 99 genes and 38 protein-coding genes among the down-regulated 75 genes).

189: The statement that the downregulation of flocculation could serve to avoid mating, though sounding logical, is undermined by the finding that mating-related processes are upregulated in the experiment. I find this statement rather speculative

-> Mating-related GO terms were not enriched in the RNA-seq results at 2h. We do not mention mating anymore and instead focus on the reduced expression of genes related to cell aggregation mediated by NSF (page 10).

247-249: The statement is too broad, the effects are visible for maybe 3 TFs, the others don't seem to make a difference in occupancy. Also, why are these two highlighted genes of importance (pex7+, yhb1+), this is the first and last time they are mentioned?

Full Revision

-> It is correct that occupancy for most TFs that we have tested is not different. Changes in gene expression are easy to explain by TF occupancy, which we observe for a few. For this reason, these TFs were highlighted. We mention this more specifically in the revised version of the manuscript (page 13, 14) and we are showing examples in Fig 4C and 4D.

254-256: The statement that these TFs are indispensable may be too strong. Right before, the authors showed that most of these TFs don't change occupancy (and especially Fil1 and Reb1 do not show a correlation with up- and down-regulated genes, nor does Fil1 in FigS2B show a changed ChIP-seq signal).

→ These TFs were identified in our genetic screen. That is, they are indispensable.

365: 'independently of the carbon source'. As far as we can see, all experiments were performed using glucose as the carbon source, so this statement seems too strong as there is no clear proof for this. This could be an easy extra experiment to perform these tests on media with other carbon sources than glucose?

-> We agree. We have changed "independently of the carbon source" to "without any change in the carbon source" to avoid misunderstandings (page 19).

Fig1E: It is not clear if the experiment was performed with the Wild-Type or a deletion strain. In the case of the WT, colonies grew in the media not containing NSF but in Fig5E and Fig5I, the WT did grow in the media not containing NSF. It could be more relevant to plate out 1 colony like in the second screen. Thus, unless different strains were used for both experiments, the results seem inconsistent with each other, which is not mentioned in the manuscript.

We have also observed NSF-independent adaptive growth by cultivating cells for an extended period without the addition of NSF. However, to avoid confusion, we have refrained from discussing this NSF-independent adaptive growth in this section. Consequently, we have modified the figure used for clarity (Figure 1E). Additionally, we have included details about the strains used in the Figure legend (page 32, 34, and 35). See also our responses to reviewer 1.

Fig1E, Fig5B, Fig5E, and Fig5I: For these experiments, different nitrogen concentrations were used depending on the media, but this has not been addressed/mentioned in the manuscript.

-> We added the respective information to Material & Methods.

Minor comments:

53-57: I would like more elaboration on why CCR and NCR are important for virulence of human pathogens or relevant for industrial applications, and link back to this in the discussion.

Otherwise, it is superfluous to include this in the introduction.

-> While lines 52-54 emphasize the significance of CCR and NCR in human pathogens, underscoring their crucial role in rapidly adapting to niche environments during infection, lines 58-63 highlight their importance in industrial settings. Specifically, in the brewing industry, CCR is indispensable for efficiently synthesizing ethanol through the fermentation pathway, utilizing carbon sources. Similarly, NCR is vital for optimizing nitrogen source utilization and biomass production. However, the fine-tuning of NCR becomes paramount, as an excessive presence

can lead to the accumulation of substances like urea and proline, negatively impacting the quality of products such as wine. We have made changes to the introduction accordingly on pages 3-4.

108: Does having a h^- library have any impact on the outcome compared to the original h^+ library?

-> Because h^- and h^+ strains grow slightly different, *S. pombe* libraries are typically aligned on one mating type. In additional experiments, h^+ strains were occasionally used as hosts; however, as of now, no significant phenotypic differences have been observed between h^- and h^+ . On page 6, the purpose behind aligning the library with h^- is explained.

170: Why would only one of the 117 NSF-linked genes change expression in the RNA-sequencing experiment? Any explanation as to why the expression remains unchanged for the 116 other NSF-linked genes?

-> Because the expression of all other genes is not influenced by NSF. That is, these genes are needed to respond, but they are not differentially expressed. Yet, there is a possibility that the induction of transcription for these genes may not be observable at the RNA-seq time points of 2, 4, and 6 hours, or alternatively, changes might occur at the protein level rather than at the transcriptional level. We mention this in the revised manuscript (page 9).

212: Please elaborate the discussion of these results. I understand the point that at low cell densities, the cells do not produce NSF as much, and thus adding NSF induces $nrt1^+$. However, the added value of testing this in different media is unclear, especially when the results of strength of increase in $nrt1^+$ show the opposite trend for the two different media between low and high nitrogen content.

-> We have amended the text accordingly on page 12.

216: Why was the *ADH1* promoter chosen as a 'negative control'?

-> We used the *adh1* promoter, traditionally employed as a housekeeping gene in *S. pombe* and *S. cerevisiae*, for its well-established role as a constitutive promoter. To clarify its purpose, we added the following sentence on page 12: "the *adh1* promoter, which is known to be constitutively active in *S. pombe* (Russell & Hall, 1983)."

284-288: Fig5E: To test whether AYR1 is indeed metabolizing NSF (and thus supporting this statement), an overexpression strain of AYR1 could be made to see if it grows on the EMM + NH_4Cl without NSF added.

-> Thank you for suggesting this experiment. As mentioned in our response to reviewer 1, the new results are shown in Figure 5F and 5G and mentioned on page 17.

385: 'NSF would not strictly revoke NCR only, but also CCR': the authors should try to provide experimental evidence, citation(s), or clearly state it to be a hypothesis. This comment links back to the major comment on line 365.

Full Revision

-> While experimental results regarding CCR revocation have not been obtained, the observation that NSF induces respiration even under glucose culture conditions suggests that NSF can initiate respiration prior to CCR release. Therefore, the sentence that was pointed out has been revised to: 'NSF could be considered a mediator not only for revoking NCR but also for inducing respiration, potentially preceding the release of CCR in carbon metabolism.' (page 20)

405: typo: strains were validation, should be 'validated'

-> thanks

644: typo: '+' sign not in superscript

-> Thank you for pointing out the typo. We deleted "+" symbols as advised by Reviewer #3.

Fig1G: Could the differences in OCR be due to differences in growth rate or remaining glucose? It could for instance be that the culture in the control condition grew less fast, thus still having glucose and therefore still in fermentative metabolism. Showing or mentioning growth rates, nutrient concentrations could help to strengthen this finding.

-> We have been measuring the cell count before OCR measurements, and confirmed that there is no difference in growth, regardless of the presence or absence of NSF treatment for 8 hours (information added on page 8-9).

Fig2A: The top two 'most' upregulated genes (nrt1 and mei2) were taken along for additional experiments. However, one gene with a significant upregulation labeled in red on the left seemingly shows stronger induction than the second gene (mei2). Why was this gene not taken along?

-> Because genes with low basal expression levels are prone to artifactual qRT-PCR measurements. We therefore used genes with expression levels exceeding 100 cpm. We mention this on page 11 of the revised manuscript.

Fig2C: the x-axis label is not immediately clear to the reader.

-> Thank you. We have simplified the x-axis label.

Fig4B: typo: 'non treatmentt'

-> Thanks.

Reviewer #2 (Significance (Required)):

This manuscript advances our understanding of nitrogen signaling pathways and nitrogen catabolite repression in the model organism *S. pombe*. Specifically, it shows how a nitrogen signaling factor functions to send signals and adapt the metabolism upon a change in nutrients and reveals that this nitrogen signaling factor triggers a change from fermentation to a respiratory metabolism. These findings are relevant for the broad fields of applied microbiology, signal transduction and metabolic regulation. Relevant literature is appropriately cited, although

Full Revision

the links with Crabtree repression in *S. cerevisiae* are perhaps not fully supported.
This manuscript was reviewed by experts with expertise in *S. cerevisiae*, Crabtree effect, respiration-fermentation balance, adaptation to changing environments.

Reviewer #3 (Evidence, reproducibility and clarity (Required)):

Summary:

In this manuscript the authors are interested in understanding how fission yeast respond to a Nitrogen Signaling Factor (NSF) that has previously been shown to allow Leucine auxotrophs to grow in the presence of Leucine when Nitrogen Catabolite Repression (NCR) is triggered by the presence of a high quality Nitrogen source such as Ammonium Chloride (NH₄Cl).

The authors begin with a screen to identify genes that affect the ability of wild type cells grown near cells with leucine auxotrophy to enhance or abolish NCR phenotype. They screened the non-essential gene deletion library which they manipulate so that it only contains a leucine auxotrophy (unlike the original gene deletion library which contains additional auxotrophies). They identify 137 genes whose deletion allows growth of Leu auxotrophs in the presence of Leucine and Ammonia without the presence of WT cells. These genes are required for NCR. They further identify 203 genes which do not bypass NCR even in the presence of wild type cells, and are thus important for bypassing NCR in the presence of WT cells.

They then conduct a second screen to identify which of these genes are important for bypassing NCR in response to the Synthetic NSF, 10(R)-hydroxy-8(Z)-octadecenoic acid, by looking for genes which grow in the presence of leucine when ammonia is not present, but do not grow in the presence of leucine when ammonia is present, even when NSF is added. This second screen identifies 117 strains carrying deletions in a gene set enriched for genes related to cellular respiration and mitochondria. They then show that the NSF bypass of NCR is linked to respiration by showing that it is abolished in the presence of the respiration inhibitor Antimycin A, that growth in low levels of glucose can bypass NCR in the absence of NSF and that cells supplemented with NSF have a higher oxygen consumption rate.

To gain insight into how the cell responds to NSF, the authors then gather RNA expression data from cells grown in high ammonium concentrations following treatment with NSF relative to a negative control treated only with Methanol (the vehicle into which NSF is dissolved). They argue that the gene expression pattern resembles gene expression data from cells undergoing respiration in glycerol relative to cells undergoing fermentation in glucose. They show that the upregulated genes relate to trehalose synthesis, detoxification of Reactive Oxygen Species, and cellular fusion and the downregulated genes are related to cellular adhesion and flocculation.

They validate their RNA-seq measurements by showing that the two most highly induced and two most highly repressed genes respond to NSF addition in a dose dependent manner and do not respond oleic acid which is chemically similar to NSF. The most highly responsive gene they identify is an uncharacterized gene, SPBPB2B2.01, which they suggest naming "NSF-responsive amino acid transporter 1" (nrt1). They also show that the nrt1 response is dependent on the culture density, and that the response is present (though the magnitude varies) in YES and in EMM under varying nitrogen concentrations, and that yfp driven by the nrt1 promoter is induced by NSF.

The authors then investigate the 8 transcription factors that were present in their list of genes required for NSF-mediated adapted growth. They note that Hsr1 was the only one of these transcription factors, indeed the only gene, that was a hit in their screen for NSF-mediated adapted growth and whose expression was induced upon NSF treatment. To see if the activity of the other transcription factors changed in response to NSF treatment, the authors then gathered ChIP-seq data using 6 of these transcription factors as targets for IP. They saw that for Hsr1 and Php3, targets that had increased RNA-seq expression showed an increase in promoter occupancy while for Hsr1, Php3, Adn2, and Atf1, genes that had decreased RNA-seq expression showed a decrease in promoter activity.

Finally the authors attempt to identify the mode of action of NSF by generating a functionalized NSF with an alkyne tag (AlkNSF) which they then use as a probe to identify NSF binding partners. They first show that AlkNSF does allow bypass of NCR, although at 30-fold higher concentration. Also AlkNSF induces *nrt1* expression in a dose dependent manner, although the expression saturates at a lower level and requires a much higher concentration for induction. They then look for proteins that co-purify with AlkNSF compared to a control that was pre-incubated with NSF which was expected to compete off AlkNSF. The only significant protein they saw was Ayr1, which was not identified in their screen and which did not abrogate NSF bypass of NCR when deleted independently. They saw that Ayr1 deletion actually increases the response of *nrt1* and *mei2* targets to NSF, and speculate that Ayr1 metabolises NSF and reduces the cell's ability to respond to NSF to bypass NCR.

They then repeat the affinity purification / mass spec protocol in an Ayr1 delete cells to identify other interaction partners, this time incubating with a higher concentration of NSF, and also comparing to an experiment using Alkyne Oleic Acid as a control for non-specific binding. The top two specific hits from this assay are Hmt2 and Gst3. NSF was still able to rescue NCR in *gst3* deletes, indicating that it was not relevant for the phenotype. Cells lacking *hmt2* did not grow in EMM, but did grow in YES when not supplemented with ammonium and when supplemented with ammonium did not grow, and addition of NSF did not rescue growth. They also see that *nrt1* and *mei2* gene induction in response to NSF is abolished when *hmt2* is deleted. They then argue that *hmt2*, a sulfide:quinone oxidoreductase localized in the inner membrane of mitochondria is a direct target of NSF that triggers a switch to respiratory metabolism and allows bypass of NCR.

Below are comments that I think ought to be addressed prior to publication (Major comments)

1) In line 70, the authors state that "S. pombe cells rely on their own BCAA synthesis to sustain growth" when grown alongside Leucine when ammonium is supplied in the media. If prototrophs can inhibit NCR via NSFs in neighboring auxotrophic cells on the same plate, couldn't they also inhibit NCR within their own colony? How do we know that prototrophic cells grown in high quality nitrogen sources along with, say leucine, are not taking up leucine? The fact that leucine

auxotrophs cannot grow in high quality nitrogen sources when leucine is present does not imply that wild type cells must use be synthesizing BCAAs rather than importing them. In a recent paper (Kamrad et al Nat. Microbiol. 2023, <https://www.nature.com/articles/s41564-022-01304-8>), it was shown that *S. cerevisiae* cells grown in lysine and in high concentrations of ammonium uptake lysine rather than synthesize it as lysine concentrations in the media are increased. I am aware via unpublished results that this is the case for Leucine as well. I would be surprised if the same isn't true in *S. pombe*. The authors should caveat or remove this assertion.

-> This is an interesting comment. Our reasoning would be the following: Given that prototrophs exhibit robust growth even under NCR conditions significantly affecting BCAA uptake control, it is reasonable to infer that they rely on their own BCAA synthesis, at least during the initial stages of adaptation. As demonstrated in Figure 3D, it is conceivable that, depending on the cell number, the synthesis of NSF by the cells increases leucine uptake even under NCR conditions. In fact, when spotting and increasing the cell number, adaptive growth is observed even in the absence of NSF, even in leucine auxotrophs. Considering the increased amino acid uptake under high ammonium conditions in *S. cerevisiae*, the revised sentence on page 4 reflects this understanding: Thus, *S. pombe* cells rely predominantly on their own BCAA synthesis for adaptation under NCR conditions.

2) It is important for the authors to put their observation linking respiration to rescue from NCR in context with findings from a closely related study (Chiu et al 2022) which included some authors from this manuscript and which the authors cite. In that paper, it was shown that the siderophore ferrichrome can also rescue NCR in fission yeast. That paper stated "It is likely that ferrichrome increased mitochondrial activity, which enabled efficient utilization of glucose downstream of the glycolytic pathway" based on experiments in different concentrations of glucose. This evidence seems to support the link between respiration and rescue from NCR proposed by the authors of this manuscript. The authors should acknowledge this closely related and earlier work as it strengthens the case they are trying to make. They could even test if ferrichrome addition makes cells sensitive to antimycin A (as in fig 1E), but that extra experiment would be optional in my opinion.

-> The report (Chiu et al., 2022) has not demonstrated the connection between the rescue of growth under NCR and mitochondrial activity by ferrichrome. Therefore, we believe it would not be appropriate to extrapolate and strengthen the current results by referring to that study. However, given that the enhancement of respiration under NCR conditions showed a discernible relationship with growth in the current study, we can illustrate the hypothetical mechanism from the previous research. The following sentences have been added to the discussion part on page 19. " In one of our previous studies, we found that the addition of ferrichrome results in the recovery of growth under high ammonium conditions and we speculated that this could be linked to increased mitochondrial activity (Chiu et al, 2022). From the current study, a model could be proposed in which ferrichrome enhances BCAA uptake under high ammonium conditions through increased mitochondrial activity."

3) In figure 1B for the second screen I do not understand what the photos represent. For the

photos, two rows are meant to have no NH₄ and also no NSF and the label on that image makes no mention of Leucine supplementation. In the diagram there are two rows that have NH₄ and leucine and one row that has no NH₄ but does have leucine. I assume the diagram is correct and the labels on the images are incorrect.

-> Thank you for pointing out this mistake. We corrected the labels in Figure 1B.

4) It would be important for the authors to put their observation linking respiration to rescue from NCR in context with findings from Chiu et al 2022 which the authors cite. In that paper, it was shown that the siderophore Ferrichrome can also rescue NCR in fission yeast which the authors cite which found that a siderophore rescues NCR. Also the authors of that paper stated "It is likely that ferrichrome increased mitochondrial activity, which enabled efficient utilization of glucose downstream of the glycolytic pathway." based on experiments in different concentrations of glucose. This evidence seems to support the link between respiration and rescue from NCR proposed by the authors of this manuscript.

-> see our response to comment 2

5) In line 133. The authors state that the 29 mutants that didn't grow under Leucine supplementation either without NH₄Cl or with NH₄Cl whether or not NSF was present were "related to EMM Growth, leucine uptake, or utilization of ammonium as the sole nitrogen source." The first two make sense, but I can't see why a strain with deletion of a gene related to utilization of ammonium as a sole nitrogen source wouldn't grow when supplemented with leucine. In fact for all the leucine auxotrophs in the screen, if one was to try to grow them with ammonium as the sole nitrogen source they would not grow, so it isn't clear that this screen can identify genes responsible for utilization of ammonium as a sole nitrogen source. The authors should clarify or remove this point.

-> Thank you for your advice. As suggested, we have removed this point (page 7).

6) 203 strains are important for avoidance of NCR (because in the presence of Ammonium and Leucine, as well as a WT strain, they cannot grow). Of these 57 strains can't grow in the presence of a WT strain but they can grow in the presence of NSF. The authors conclude in line 138 that these strains are "likely to respond to a transmissible signal that is different from NSF". This is confusing because deletion of these genes still does allow cells to respond to NSF, however when these cells are growing in the presence of wild type cells (which in their model are releasing NSF), the cells don't grow. I am confused about the nature of the transmissible signal that the authors suggest. It would appear that when these genes are deleted and grown next to a wild type cell which sends the alternative signal and the NSF, the other transmissible signal would inhibit the ability of NSF to release NCR (as NSF can still rescue the gene). It is not clear how the other transmissible signal would work when the gene is present as it is clearly not necessary to rescue growth.

A simpler explanation might be that there was contamination in the second screen, or that there was a threshold effect - perhaps in the first screen the strains grew just below a threshold and in the second screen it grew just above that level.

The authors should clarify their interpretation for these strains, and acknowledge any alternative technical explanations.

-> As suggested, we believe the results may be attributed to a threshold effect. Therefore, the following sentences have been added to pages 7-8." We speculate that these mutants fell below the threshold of NSF required for adaptive growth in the 1st screen, whereas in the 2nd screen they surpassed this threshold, leading to the manifestation of adaptive growth. Although these factors may be partially involved in the NSF response, we did not include them in any further analyses in this study to keep the focus on the strongest hits."

7) The authors' efforts to removed confounding effects that might stem from additional auxotrophic alleles made the screen more convincing. However, Fig 1E, 1F, 5B, and 5E were done with EMM+Leu+Ade+Ura, while the initial strain was just done in the presence of additional Leucine. It is unclear why this was done from the text and captions, but I assume it was because they used a strain that was ade- and ura- in addition to being leu-. Given that they had strains without these additional mutations, this seems like a strange choice. The authors should acknowledge that there are possible confounding effects of adding adenine and uracil to the media, and, if they did have additional metabolic deletions, acknowledge that that could possibly be confounding.

-> Since there is a possibility of using the Bioneer gene disruption library, we utilized strains with auxotrophy background leu1-32 ade6-M216 ura4-D14. There was an oversight in providing information about these strains. We have added this information to the Figure legend on pages 32, and 35-36.

Considering the potential impact of auxotrophy for adenine and uracil on our results, we repeated this experiment in an *hmt2Δ* leucine auxotroph (Figure EV3H, mentioned on page 16-17). We also checked *hmt2Δ* prototroph did not show growth defect under NCR condition in Figure EV3I and explain on page 17.

8) Fig 1E, it appears that cells can grow without NSF in the presence of ammonium and additional amino acids after 10 days (although NSF is required for growth at 5 days). This is not a problem for the screen as that was taken at 5-6 days, but it appears as though NSF does not rescue growth so much as speed it up. The authors should acknowledge this when describing the phenotype. It also argues for a quantitative time course growth experiment to compare growth over the course of 10 days with and without NSF, although this would not be necessary to the paper's main argument.

-> We address this on page 6 of the revised manuscript:" When grown for an extended period (> 1 week), instances of cell-to-cell communication-independent growth can be observed in *S. pombe*. Therefore, we assessed growth after 5 or 6 days of cultivation in our screen." To avoid confusion, we have refrained from discussing this NSF-independent adaptive growth in this section. Consequently, we have modified the figure used for clarity in Figure1E.

9) In line 191 and 192, the authors suggest that the "downregulation of flocculation/adhesion related genes by NSF could serve to avoid undesirable mating during growth". If this is the case,

Full Revision

I don't understand why mating genes and cellular fusion genes would be upregulated. What do the authors mean by undesirable mating? Wouldn't flocculation increase desirable mating as well? If all mating is undesirable, wouldn't upregulation of mating and cellular fusion genes be detrimental?

-> After reanalyzing the GO analysis specifically for up- or down-regulated genes at the earliest time point of 2 hours in the time course RNA-seq, the down-regulated genes still revealed factors related to flocculation, while the upregulated genes did not show enrichment in GO terms related to mating but exhibited enrichment in genes associated with trehalose synthesis. Therefore, the mention of 'undesirable mating' has been removed, and only the possibility of NSF inhibiting flocculation is now discussed (page 10).

10) The authors mention that trehalose is an antioxidant, for which they reference Malecki 2019, however that paper shows no direct evidence of trehalose functioning as an antioxidant under respiratory conditions. It only shows that some trehalose synthesis genes are upregulated when cells are grown under glucose. The authors should identify primary literature to back this statement up, or soften the wording. Also trehalose is known to be a storage metabolite (which is mentioned in Malicki et al 2019, but not in this manuscript). In fact work in budding yeast has show that trehalose can be a shared metabolite that can be produced by respiring cells and used as a fermentable carbon source in communities of budding yeast cells that consist of fermenting and non-fermenting cells (Varahan et al, eLife 2019 <https://doi.org/10.7554/eLife.46735>). It seems that this role should be considered as an alternative explanation for the induction of trehalose in respiratory cells.

-> Thank you. We added the following sentence to page 10, referencing the paper (Malicki et al., 2016) that discusses the induction of trehalose synthesis-related genes by glycerol in *S. pombe*. We also cite the paper (Moon et al., 2020) in the same part to support the notion of trehalose functioning as an antioxidant: "Coherent with the induction of a respiration-like gene expression program, the trehalose synthesis pathway is also upregulated under respiratory conditions (Malecki et al, 2016). Notably, it has been reported that trehalose functions as an antioxidant in the probiotic yeast *Saccharomyces boulardii* (Moon et al, 2020). This could indicate a potential role in counteracting ROS production associated with NSF-induced respiration."

11) Line 208: The stimulatory effect of NSF on NRT1 decreased with cell density, thus cell density is likely to be an important factor in terms of gene expression. The methods section, text and figure captions do not mention the density at which cells were inoculated/harvested for RNA-seq and other experiments. If that density was more than OD 0.1, then this would be inconsistent with the measurements from Fig 3. Also in fig 3D, The culture density is not mentioned in the figure or the caption, even though the text suggests that for that experiment cells were grown at low density (Lines 212-213). The authors should provide information on density for their experiments in order for them to be reproducible, as they show it is a key factor.

-> The information regarding cell density in RNA-seq experiments was detailed in the Materials and Methods section (see page 24), while the cell density information for qRT-PCR experiments was provided in the figure legends (see page 32-36).

12) In suggesting a name for NRT1 (NSF-responsive amino acid transporter 1), the authors assume that the gene has a role in amino acid transmembrane transport, but they have no experiments showing this phenotype. They mention that it is Inferred from homology with other amino acid transporters. I presume this name has already been approved by Pombase and is not provisional, but it seems that including phenotypes inferred from homology, rather than from experiments is unwise. Do the authors have any other direct evidence that this is a bona fide Amino Acid Transporter? Perhaps a name like "NSF-responsive gene" would be more appropriate.

We agree. We have decided to change the name to *Nrg1* (NSF-responsive gene).

Related to this, it appears that the expression level of *Nrt1* may be very low (see Fig S2B in which the scale of the RNA-seq track is very small [-1,1] and the amount of expression is very small even when NSF is added). Looking at Fig 2A, the total transcript abundance did not appear to be very low in terms of counts per million (over 100) is this a discrepancy in fig S2B? Perhaps the large fold change is the result of counts very close to zero in the control condition? Also in Fig 3 the *nrt1* expression levels did not appear to be especially low and they appeared repeatable. Is the RNA-seq data shown in fig S2B for *nrt1* a fluke or am I misinterpreting it?

-> We have confirmed a gradual increase in the expression level of *nrt1* (*nrg1*) over time with NSF treatment. By switching from the 2h to the 6h RNA-seq time point, it became evident that the transcription of *nrg1* is enhanced by NSF. The qRT-PCR experiments were performed with 6h-NSF treatment, as described.

13) To show that their Chip-seq worked, the authors showed specific examples of Chip-seq reads for target genes Line 240, "Previously determined target genes of these TFs were significantly enriched in our data set, demonstrating that the experiment has worked (Figure S2A)." Is the significance here, the threshold from fig S2B? If so that threshold should be clearly stated here in the text. If it is the fact that *asn1* shows up as "Fil1 bound" is strange as there are no genes that had significant changes in ChIP-seq signals for fig S2B. If there is another threshold the authors should describe it. While some of the examples they showed were convincing (e.g. *php3*-flag for the *php3* regulated gene *gln1* and the increased reads for *srw1* for the *reb1* target *srw1*), there were some targets that didn't seem to be especially enriched for their designated transcription factor. For example, the gene *trx1* which was identified as an Hsr1 binding target had some binding from Hsr1, but more from Php3 and equivalent amounts for many of the other transcription factors. A clear description of how genes are chosen to be significant in the text, alongside references/selection criteria the authors used to select the specific genes shown should be provided to improve reproducibility.

-> The term 'significantly enriched' of our ChIP-seq data refers to peaks with a MACS2 score of 100 or higher identified using MACS2 between input and ChIP samples, as detailed in the Materials and Methods section (see page 26). To clarify, the following sentences have been

added to page 14 for explanation:" Previously determined target genes of these TFs were significantly enriched over input samples in our data set, demonstrating that the experiment has worked (Figure EV2). "

The term 'significant changes' in Figure S2B moved to Fig 4A pertains to the comparison between without NSF and with NSF conditions. Since the number of gray dots in Figure S2B moved to Fig 4A represents peaks where Fil1 binding itself is significantly enriched compared to the input sample, we are confident that the ChIP-seq worked. The criteria for gene selection were based on information from ChIP-seq and ChIP-qPCR of TF and the transcriptional changes of target genes upon TF gene disruption. These details have been summarized in the Figure 4A legend."

14) In lines 244-246 the authors state that "These differences in TF occupancy were positively correlated with target gene expression changes. That is, individual genes that were upregulated by NSF tended to be more strongly bound by the TFs, whereas downregulated genes were less occupied by the respective TFs (Figure 4A)." This is far from a general trend. The trend is not there for reb1 and fil1. In fact fil1 looks to the eye like it shows a decrease in occupancy for genes with increased expression, and I worry that the authors did a one sided test for significance that would have missed this, although the variability of the genes that don't change in this case is very high, so there could be no significant effect. The authors elaborate on some of the detail in following statements, but they should soften or remove this statement.

Related to this, in line 254, the authors state: "These results imply that NSF exposure rewires the recipient cell's transcriptional program, for which the TFs Atf1, Adn2, Adn3, Fil1, Hsr1, Php3, Php5, and Reb1 are indispensable (Table S3)." While I am convinced from the RNA-seq evidence and some of the chip-seq evidence that NSF exposure rewires cell's transcriptional program, I am not convinced that the 8 transcription factors they mention are indispensable for rewiring the transcriptional program. While they may be indispensable for the phenotype itself, Reb1, and Fil1 show no no significant enrichment in occupancy of upregulated or downregulated targets (Fig 4A) and, along with Atf1, Reb1, and Fil1, have very few genes in which occupancy is changed significantly (Fig S2B), while no chip-seq experiments were shown for Php5 and Adn3.

The more specific summary of the data (Lines 250-253) from Fig S2B describing how hsr1 and adn2 have the strongest effects of the transcription factors required for NSF-mediated NCR bypass is a much stronger message for this section.

-> Thank you for this comment. Because it is similar to reviewer 2, see our response above.

15) In line 335, the authors state that "in contrast to other communication systems, NSF does not induce noticeable changes in *S. pombe*'s morphology", referring to changes in mating, filamentation, and bacterial biofilm formation. However they do show very clearly that NSF does cause a large decrease in expression in flocculation/adhesion genes. The fact that they do not see a change in morphology is likely due to the fact that the lab strain in the conditions used for

this assay do not flocculate. We have recently identified conditions and strains which do exhibit flocculation in this preprint [<https://www.biorxiv.org/content/10.1101/2023.12.15.571870v2>]. It is likely that if they had a strain and conditions that did flocculate addition of NSF would break up flocculation and thus change the morphology based on their evidence. The authors should remove or caveat this point.

-> We have modified the explanation to suggest that NSF may represent a novel cell-to-cell communication system that not only influences the activity of respiration and transcription but also induces morphological changes associated with flocculation. The following sentences have been added to pages 10-11 in the Results section instead of the Discussion: "Because cell-to-cell communication is often linked to morphological changes such as biofilm formation in bacteria (Miller & Bassler, 2001; Hammer & Bassler, 2003; Mukherjee & Bassler, 2019), mating, or filamentation in fungi (Chen *et al*, 2004; Hornby *et al*, 2001; Merlini *et al*, 2013; Ramage *et al*, 2002), it is possible that NSF-mediated cell-to-cell communication could be linked to morphological changes. Although an actual impact of NSF on flocculation has not been confirmed, it might be worth investigating this further."

16) Line 270 Fig 5B: The concentration of NH₄Cl listed in the text (374mM) does not match the concentration shown on the figure (748mM). I assume this is a typo but it should be corrected prior to publication.

-> Thank you for pointing out this mistake. We corrected it (page 15).

Also I have several minor comments to help improve the manuscript.

m1: Lines 66-70- state that "uptake of the branched-chain amino acids (BCAA) isoleucine (Ile), leucine (Leu), and valine (Val) is suppressed in the presence of high-quality nitrogen sources such as ammonium or glutamate, because the expression of transporters or permeases that are needed for the uptake of poorer nitrogen sources are down regulated (Zhang *et al*, 2018)." This reference is for *S. cerevisiae* and is a review. The authors should cite original results in *S. pombe* if possible, and if that is not available, alert the reader that this result is from a different species.

-> Thanks. We specify on page 4 that this is the case for *S. cerevisiae*.

m2: It is unclear from the methods section how the images taken for the screens were analyzed. Were they analyzed and scored by hand, or using custom image analysis software. Either way, when publishing the authors should publish the scores for each deletion mutant in their screen. If there was custom image analysis, the authors should mention in their methods the cutoffs which they used to score growth, and consider plotting the data as a supplement so readers can get a sense of how sensitive the screen was.

-> The information of judgement of growth was added on page 6

m3: The authors identify 137 mutants that did not require NSF signaling to bypass NCR and claimed these genes were required for NCR. It would be helpful and give more confidence in this screen to demonstrate the extent to which the genes identified in this study overlap with any

Full Revision

previous genes required for NCR, and whether there was any GO-term enrichment in this set.
-> We could not find any genes and GO-term related to NCR

m4: It would be interesting if the authors could speculate a bit in their discussion on why mitochondrial respiration counteracts NCR. Is there something about cells undergoing respiration that would make it easier for them to use BCAAs than to produce them, or conversely something about fermenting cells that makes it easier for them to produce BCAAs rather than importing them?

-> it is an interesting but also a difficult question to address as we mentioned in our response to Reviewer #1's minor comment 4. On page 19 of the initial submission, we mention that it has been demonstrated in *S. cerevisiae* that induction of respiration by glucose de-repression leads to an increase in leucine uptake. However, whether this is due to transcriptional regulation of transporters or regulation of transporter activity remains unclear. In our study, among the up-regulated genes identified by RNA-seq, the only gene encoding a putative amino acid transporter was *nrt1*. Given that overexpression of *nrt1* (*nrg1*) promotes growth under NCR conditions in our preliminary data, it is suggested that Hmt2-mediated regulation of *nrt1* (*nrg1*) expression is one mechanism. Additionally, as indicated on page 9, the time points for RNA-seq are relatively early in comparison to the growth rate, so it is anticipated that further regulation of amino acid transporters may occur at later time points. We believe that transcriptional profiling will be valuable for exploring this aspect in future studies. However, as the culture progresses, the effect of NSF diminishes due to the increase in cell density associated with cell proliferation. Therefore, for this experiment, it will be necessary to use strains incapable of NSF synthesis. Because we do not have such strains, we are not able to perform such experiments.

m5: It is unclear why Figure 1F has 'MP biomedical's TM' listed in the figure. It doesn't seem to be listed in the caption or the methods. Is this different media than in other experiments? If so, the authors should add that information to the methods or the caption.

-> see page 22, Materials and Methods: we mention the use of FORMEDIUM's regular EMM containing 2% glucose, and we specify the use of glucose-free EMM media from MP Biomedical™ for the preparation of lower glucose EMM media.

m6: In Line 160, positively influenced is strange wording, do the authors mean "induced"?

-> Thank you for pointing this out. We corrected it to "induced" (page 9).

m7: In the section on gene expression change upon exposure to NSF, the authors use a + after each gene name. My understanding is that that notation is meant to refer to strains with the wild type genotype of that gene, and not the gene itself. Shouldn't the gene be italicised in lower case to represent the gene? See: Lera-Ramirez et al 2023 <https://doi.org/10.1093/genetics/iyad143>.

-> Thank you for alerting us that the nomenclature guidelines have changed. We have corrected this throughout the manuscript.

m8: In Fig 2A, genes are displayed on a plot that depicts level vs log2FC, but a comparison

between the fold change and p-value would be more useful, and I believe DESeq2 should provide an adjusted p-value for these genes. A related issue is that it appears as though there were no biological replicates, though there was data gathered at different time points. In these genome wide experiments, replicates can give confidence to data and help distinguish true change from intrinsic variability of expression in specific genes. Though the authors did qPCR to validate specific results, it would have improved the quality of their systems-level data to have replicates for these and other key experiments (Chip-seq, affinity purification and even the screen).

-> We have employed MA plots to intuitively represent the basal expression levels of each gene for the selection of NSF-responsive reporter genes instead of volcano plot (y-axis is adjusted-p value). Each analysis primarily used only one strain (clone). However, RNA-seq cultivation was conducted in triplicate, consisting of three different flasks. Additionally, we checked the reproducibility, and the results were consistent with the conclusions of this paper. These results are presented in Figure EV1E and F. For ChIP-seq, although we used 3 L culture with one flask for each condition due to lower cell density, we conducted technical replicates in triplicate for each condition. Furthermore, as a follow-up experiment, we generated gene knockout strains for important transcription factors, assessed their impact on transcription through qRT-PCR (Figure 4D), and complemented the results with ChIP-seq. In affinity purification experiments, we conducted technical triplicates for each experiment and performed reproducibility experiments. We have added this information about replicates for RNA-seq on page 24, ChIP-seq in page 26, and affinity purification in page 28.

m9: Supp Fig S1: To show that similar gene expression profiles exist for other time points, it would be more convincing to show Log fold change 2h vs 4h and 2h vs 6h and show correlation, or else to make a heat map with all genes to see that genes that go up in one condition go up in the other conditions. It is not clear if the red and blue colors are defined for the 2h dataset and then mapped onto the 4 and 6h dataset, or if they are independently assigned for each plot.

-> We confirmed a positive correlation between 2h versus 4h and 2h versus 6h, and added it to Figure EV1A. We also mention that there is no significant alteration in the distribution of NSF-linked genes on page 9 (Figure EV1B). Additionally, we verified that a respiration-like gene expression program is triggered similarly at 4h and 6h, as observed at 2h, and this information has been incorporated into page 9 and is shown in Figure EV1D.

m10: Mbx2 is a key transcription factor related to flocculation and adhesion genes, and its expression is correlated with expression of its targets. If this transcription factor's expression levels decreased in response to NSF, that might strengthen and help explain the decrease in expression the authors observe in flocculation/adhesion genes when cells encounter NSF. If it does not change, it might also be interesting for readers interested in these phenotypes.

-> That is an interesting point. The transcription level of *mbx2* showed a slight decrease with a log2FC of -0.44. Yet, the absence of a growth phenotype in the *mbx2* Δ strain in the growth screening suggests a low causative relationship between flocculation and NCR.

m11: In Fig 3D, The notation for the Ammonium concentrations for EMM and YES are

Full Revision

inconsistent (+ vs parentheses), also the units (mM from the caption) are not on the figure, but the abbreviation "N" is which is confusing and inconsistent with the other plots in which NH₄CL is not abbreviated. Additionally, the caption lists additional nutrients in the media for the EMM conditions (Leu, Ade, Ura) which ought to also be listed.

-> thank you for pointing this out. We corrected labels in Figure 3E.

m12: In lines 233-235, the authors say "One possibility is that they remain bound to their target genes but become activated or deactivated by NSF directly, or posttranslational modification, such as phosphorylation in the case of Atf1". I don't think the authors intend this, but this sentence could be taken to mean that Atf1 has been shown to be phosphorylated by NSF in the reference they cite. I think the authors should clarify, i.e. by saying "...such as phosphorylation which is known to regulate activity of Atf1 in response to oxidative and osmotic stress [Lawrence et al 2009]".

-> Thank you for pointing this out. We agree and have amended the text on page 13 as suggested.

m13: In Fig 4B and Fig S2A, there are grey and colored tracks for the chip-seq (- and + NSF), but they are very difficult to see. If grey is in front it is hard to tell how close the colored peak when the colored peak is lower. For example, grey is in front for pex7 while color is in front for yhb1. Could the authors add some transparency so that the data for both conditions could be seen at once?

Also there is little information on the control. My assumption for the input(ChIP) sample was that it was cross-linked and sonicated but not immunoprecipitated, but it is not clear what conditions it was in. I would assume it was done without NSF treatment in WT cells, but those details should be added in the caption or methods. In particular, in the input there is a large spike for Gsf2. Do the authors have any explanation for this and does it have anything to do with that gene's NSF responsiveness?

-> The cultivation conditions and information about the input samples used for ChIP-seq are described on page 26. Additionally, the input track shown in Figure EV2 represents the input sample from Adn2-FLAG under NSF treatment for reference. We have confirmed that there are no significant changes in the tracks between different inputs at the specific gene loci of interest. Moreover, each peak was called using the respective conditions and the input from the corresponding strain.

We modified - and +NSF tracks with black line plots and colored bar plots, respectively, in Figure 4C and Figure EV2.

m14: The authors might consider putting something like Fig S2B (or even a corresponding volcano plot) as a main figure for Fig 4 in addition to the other two panels, as the individual examples from fig 4B are nice to see, but do not give a broad overview of the data.

-> We put this Figure to main Figure as Fig 4A.

m15: In line 348, the wording "Would score" might be better replaced by "would be identified."

Full Revision

-> We have amended the text as suggested.

Reviewer #3 (Significance (Required)):

Assessment:

In general I find the authors arguments compelling and their experiments convincing. The initial and follow on screens were well designed and the authors linked respiration and the action of NSF in a convincing way. The analysis of RNA-seq data was also convincing, especially regarding the decreased expression of flocculation and adhesion genes, and the follow up of specific targets gives confidence in the data (though see Major point 12 below regarding the naming and expression levels of *nsf1*). The identification of *hmt2* as a functional target of NSF was compelling and rigorous, and the authors offer an interesting hypothesis to connect this to respiration that could form the basis of future studies.

At times I thought that some of the interpretation of the results was hard to follow, poorly worded, or off the mark (see comments below). The presentation of the CHiP seq data also felt incomplete, though the influence of *Hsr1* and *Adn2* on expression of NSF1 targets was convincing. The genome wide assays (RNA-seq, CHiP seq, screen and pull-down/mass spec) could have done with replicates which would have improved statistics and reliability of the results presented for those experiments, although for key messages, the authors followed up with convincing targeted experiments.

The study represents an advance on recent work in NCR in fission yeast in linking this with the broad metabolic switch between fermentation and respiration, and in that sense makes this of interest to a broader swathe of the microbiology community, outside those interested in metabolic regulation in microbes. In addition to being of interest to applied researchers interested in producing metabolites with yeast and other microbes, the link to cell signaling and, via flocculation and adhesion genes, to microbial multicellular-like phenotypes would make this work of interest to those interested in microbial communities.

Dear Marc,

Thank you for submitting your revised manuscript (EMBOJ-2024-117143-T-R; RC-2023-02313) to The EMBO Journal. Your amended study was sent back to the three original Review Commons referees for their scientific re-evaluation, and we have received detailed comments from all of them, which I enclose below. As you will see, the experts state that the work has been substantially improved by the revisions and they are now in favour of publication, pending minor revision.

Thus, we are pleased to invite you for a final revision of your manuscript considering the remaining minor points made by the experts. We should upon resubmission be able to swiftly proceed with acceptance of this work.

Please contact me at any time if you have additional questions related.

We also now need you to take care of a number of issues related to formatting and data annotation as detailed below, which should be addressed at re-submission.

As you might have seen on our web page, every paper at the EMBO Journal now includes a 'Synopsis', displayed on the html and freely accessible to all readers. The synopsis includes a 'model' figure as well as 2-5 one-short-sentence bullet points that summarize the article. I would appreciate if you could provide this figure and the bullet points.

Thank you for giving us the chance to consider your manuscript for The EMBO Journal. I look forward to your final revision.

Again, please contact me at any time if you need any help or have further questions.

Best regards,

Daniel

>> Please add up to five keywords to your study.

>> Author Contributions: Please remove the author contributions information from the manuscript text. Note that CRediT has replaced the traditional author contributions section as of now because it offers a systematic machine-readable author contributions format that allows for more effective research assessment. and use the free text boxes beneath each contributing author's name to add specific details on the author's contribution.

More information is available in our guide to authors.
<https://www.embopress.org/page/journal/14602075/authorguide>

>> Add a 'Disclosure and Competing Interests Statement' to your study.

>> Section order should be corrected as follows: title page with complete author information, abstract, keywords, introduction, results, discussion, materials & methods, data availability section, acknowledgements, disclosure and competing interests statement, references, main figure legends, tables, expanded figure legends

>> Funding information: the following also needs to be entered in our online manuscript system as it is acknowledged in the manuscript file: 'Novartis Research Foundation'. In turn, 'Friedrich Miescher Institute for Biomedical Research' needs to be added in the manuscript file.

>>Dataset EV legends: Appendix Tables S1-S6 should be renamed to Dataset EV1-EV6 with the appropriate callouts, and legends included as a separate tab in each Excel file

>> Source data: Source data files need to be reorganized to one file/folder per figure and ZIPing for each main figure. For EV and/or appendix figures, please ZIP together all source data.

>> Data availability section: please remove the referee token for the GEO and PRIDE datasets and make sure privacy is released.

>> Please enter a separate 'statistical analysis' section into your manuscript detailing the algorithms and tests applied.

>> Recheck the publication status of the bioRxiv reference 'Gaspar et al, 2018' and update journal information in case.

>> Consider additional changes and comments from our production team as indicated below:

- Data citations: no comments

- Figure Legends (main + EV): "1. Please note that the exact p values are not provided in the legends of figures 2b; EV 1a, e.

2. Please indicate the statistical test used for data analysis in the legends of figures 1c; 2d; 5d, h; EV 1a-b.

"

"1. Please note that the box plots need to be defined in terms of minima, maxima, centre, bounds of box and whiskers, and percentile in the legends of figures 1g; 2c; 4b; EV 1d, f.

2. Please note that information related to n is missing in the legends of figures 1g; 2c; 4b; EV 1d, f; EV 3e.

3. Please note that the error bars are not defined in the legend of figure EV 3e."

Please note that scale bar and its definition are missing for figure EV 3b.

Please use the link below to submit your revision:

Link Not Available

Referee #1:

GENERAL SUMMARY

Within this manuscript, the authors study a cell-to-cell communication process, by which nitrogen catabolite repression can be counteracted by a small molecule called NSF. Specifically, the authors demonstrate here that NSF up-regulates respiratory metabolism as a mechanism to overcome the repression by high nitrogen of amino acid internalization. Several of our comments are answered, but two important points are unaddressed: the role of transcription factors in the nuclear response is very poorly developed (points 8-9), and the role of such transcriptional response in bypassing the nitrogen catabolite repression is unclear, based on the response to our queries (point 12). For the sake of simplicity, we include here the number used in our previous correspondence, and add our new comments at the end (R):

MAJOR COMMENTS

3) Comparing the transcriptomic response to that of Malecki et al 2020 in response to Antimycin A (EMBO Rep. 21:e50845) would be useful.

-> We are uncertain why this would be useful. Moreover, in the transcriptome analysis by Malecki et al. 2020, there is no direct comparison between antiA-treated and untreated samples from the same time points. Nevertheless, we compared the up- or down-regulated genes identified in Malecki et al. 2016 under antiA treatment with those under NSF treatment. This did not reveal any significant differences.

Reviewer (R): Antimycin A treatment activates a retrograde response involved in mitochondrial metabolism, only part of which is repressed in the presence of arginine. The data shown in Malecki 2020 and Malecki 2016 indicate that ETC blockage leads to downregulation of ETC subunits and upregulation of stress genes. Let me list a few genes that DO overlap with your genes in Table S2, 'up-regulated genes':

obr1, brf1, gst2, SPBC23G7.10c, ght5, SPBC12C2.04.

4) The optical densities and whether NCR has been induced has to be clearly specified in each experiment. For instance, RNA seq data. Line 165: for the transcriptome experiment, NSF is added or not to low density cells (not indicated in results, figure legend nor materials and methods). Should addition of NSF to wild-type strain en MM trigger the same transcriptomic changes? -> Thank you for pointing this out. We have added this information ("at low cell density (OD 0.01)") to the figure legend of the RNA-seq experiments (page 32). Additionally, we have added qRT-PCR results for prototrophs in new Figure 3C and mentioned it on page 11.

(R): We do not see the explanation about cell densities on Fig. 1G. We would have appreciated if the author helped us with the review process by adding the changes in the letter.

7) Fig. 2BC: Venn diagrams should be more useful to demonstrate overlap with the Malecki data.

-> We have added Venn diagrams to supplementary Figure EV1C and mention it on page 9.

(R): Send main Fig. 2B, with poor correlation of 0.24, to EV1C. Move current EV1C to Figure 2 in substitution.

8) Fig. 4A: not very useful

-> We find it useful because it displays general trends in TF occupancy.

(R): The characterization/role of the 6 TFs on the response to NSF is poor. We understand that the 6 (8, if we include Php5 and Adn3) TFs have been selected in the screening, but ChIP seq data has failed to demonstrated occupancy as a factor modulating the response; probably activity is important. Current Fig. 4A (formerly at EV) can stay there, to highlight that Hsr1 and Adn2 may have an impact on some chromosomal loci. Current Fig. 4B has to be sent to EV. The ChIP seq data is not useful. If current Fig 4C is the best you have, send Reb1 and Fil1 data to EV. Then, develop a bit more the role of Δ TFs on gene expression changes (see below, point 9).

9) Is Hsr1 required for some of the RNA seq changes upon NSF addition? Same with other TFs

-> Yes, this is the case for individual genes that we have checked by qRT-PCR (both up and down-regulation by NSF addition), but we can't generalize because we do not have RNA-seq data from TF knock-out strains. For example, the *nrg1* gene is no longer induced by NSF in the absence of Hsr1; or in absence of Hsr1, Adn2, or Php3, *gsf2* and *pfl3* expression is as low as after treating wt cells with NSF (i.e. Hsr1 and Adn2 are required for the expression of the *gsf2* and *pfl3* genes). We show decreased *gsf2* and *pfl3* expression levels in *hsr1* Δ , *adn2* Δ , or *php3* Δ mutants in Figure 4D.

(R): Why only genes downregulated are shown? If the authors do not want to prepare RNAseq for this manuscript, why don't they qPCR the most up- and down-regulated genes (Table S2) in WT and Δ TFs, at least Δ hsr1, Δ adn2 and Δ php3? Of course, *nrg1* has to be shown. Current Fig. 4D shows that Hsr1, Adn2 and Php3 are activators, probably, of these two down-regulated genes; but the author says that the *nrg1* gene is no longer induced by NSF in the absence of Hsr1. Then, at least Hsr1 can activate *nrg1* after NSF, but activate *gsf2* and *pfl3* under basal conditions?

10) Line 287: '...it is tempting to speculate that Ayr1 dampens adaptive responses by metabolizing NSF'. Calculating MEC for NSF in delta *ayr1* and in cells over-expressing Ayr1 would be required to confirm this speculation. According to Pombase, cells lacking Ayr1 have their respiratory functions compromised (no growth in galactose, glycerol...), why is so? The opposite should be expected, if NSF-mediated respiration is enhanced in this background.

-> We are purposely speculating here, because much more work will be needed to firmly show that Ayr1 metabolizes NSF (including in vitro biochemical experiments). Because this would go beyond the scope of this study, we would like to leave this speculative.

Nevertheless, we performed the suggested experiments. We checked MEC through a growth assay using the *ayr1* Δ mutant, but no decrease in MEC was observed with the *ayr1* Δ mutation. It is possible that the result could be attributed to the impact of respiratory deficiency, as indicated by the growth delay in glycerol or galactose conditions on Pombase. Overexpression of Ayr1 did indeed affect NSF-mediated growth and NSF-responsive gene expression, fueling our speculation. We show these results in Figure 5F and 5G and mention them on page 17.

(R): You mention them on page 15, not 17. I would have appreciated copy-pasting the changes in the letter, for the sake of simplicity.

12) The data regarding Hmt2 is very interesting. As for delta *ayr1*, delta *hmt2* cells cannot grow in glycerol nor galactose according to Pombase. Is the result shown in Fig. 5J (lack of NSF-dependent activation of *nrt1* and *mei2* in delta *hmt2*) a consequence of the absence of the NSF receptor, or is it due to the lack of respiration of this background? Is delta *hmt2* really auxotrophic for Cys? Why? In this background, H2S should be enhanced, and Cys and Met biosynthesis improved. In fact, in one manuscript these cells grow fine in SG minimal media (Mol Microbiol 01 42:29), while another report indicates they are auxotrophic for Cys (Genes to Cells 2016, 21:530).

-> As an additional experiment, we confirmed the responsiveness of *nrt1* (*nrg1*) in the mutant strains with disrupted ETC-related genes, which are required for NSF-dependent adaptive growth. Despite the disruption of these genes, *nrt1*(*nrg1*) still exhibited responsiveness, suggesting that the effect is not solely due to the lack of the respiratory chain.

(R): Where is the additional experiment ? Not added ? Which mutants were used ? Is the gene response blocked with antimycin ?

This info has to be included. And is relevant for the final model and the Discussion. If this occurs for most of the NSF genes, the upgrade in respiratory function would trigger a change in the gene expression program, but the transcriptional response would not be required for unblocking the NCR. The model in Figure EV4 and the Discussion do not support the fact that mutants with deficient ETC are deficient in NSF-dependent adaptive growth, but *nrg1* is still responding.

MINOR COMMENTS

1) Who triggers NCR? Analysis of 137 genes in Figure 1b.

2) Synthesis of NSF: how is it regulated, where does it come from?

3) NCR impairs import of BCAA. How are the aa importers such as Cat1 or Agp3 eliminated from the plasma membrane? Transporter internalization, degradation, transcriptional repression... And how does NSF block the NCR regarding aa uptake? Or aa usage?

(R): Add to Discussion a sentence or paragraph about these future directions.

Referee #2:

I have previously provided a general summary and opinion about the significance of the study and outlined my concern in my initial review for Review Commons.

The following comments pertain to the authors response to my comments and those of the other reviewers from that initial reviewers. In general the revisions the authors have done have improved and clarified the study, and I still believe that it would be of broad interest to the microbiology community, those interested in metabolism and metabolic regulation, and also those interested in cell-cell communication.

For the most part the authors have addressed my major concerns, but the following concerns still remain for me. Numbers refer to the points from my original review:

1) The authors have partially addressed my concern by stating that reliance on BCAA synthesis during "adaptation" under NCR conditions, but it is unclear what adaptation refers to in this context, and it is unclear what NCR conditions mean. I assume that NCR conditions means Ammonium Chloride in the presence of Leucine. I think the authors should be more clear.

To rephrase my concern, the way that it is worded implies that NCR is happening in WT cells grown under NCR conditions, however WT cells secrete NSF to allow bypass of NCR for neighbouring Leu2 auxotrophs. Why shouldn't they do the same within a culture of WT cells? In other words, I think it is reasonable to believe, in light of the evidence the authors present, that NCR is a phenotype that only exists in Leu2 auxotrophs, because NCR bypass is in effect in WT cells. Therefore WT *S. pombe* cells ought to be able to import BCAAs, rather than having to synthesize it themselves. I am not as familiar with this topic as the authors, so I defer to their judgement, but I would suggest they soften the statement even more to the following:

Thus, *S. pombe* cells may rely predominantly on their own BCAA synthesis for at least the initial stages of growth under NCR conditions (high NH_4Cl_2 in the presence of BCAAs).

6) Although the authors did incorporate some of my feedback about the 57 strains that can't grow in the presence of a WT strain, but can grow in the presence of NSF, they still say for these 57 "Yet, because they are needed for adaptive growth (1st screen), they are likely required to respond to a transmissible signal that is different from NSF)"

I don't understand how being needed for adaptive growth is evidence of being required to respond to a transmissible signal different from NSF, and even if this is possible, it doesn't appear to me to be likely. Clearly these strains can respond to NSF, and clearly the WT strains secrete NSF. I still feel it is likely that it is just a matter of a threshold which the authors acknowledge.

I think that this is too strongly worded.

I would find something like this acceptable:

"Yet, because they are needed for adaptive growth (1st screen), they may be required to respond to a transmissible signal that is different from NSF), or else they could alter the sensitivity of the strain to NSF. We speculate..."

Also when they discuss the idea of the deletions altering the threshold of NSF I think it is a bit confusing. I don't think the mutants are surpassing a threshold, but rather the threshold of NSF required for avoidance of NCR is different in the mutants. They say:

"We speculate that these mutants fell below the threshold of NSF required for adaptive growth in the 1st screen, whereas in the 2nd screen they surpassed this threshold, leading to the manifestation of adaptive growth."

I believe that they may mean something more like:

"We speculate that these mutants are less sensitive to NSF for avoidance of NCR, and the amount of NSF provided by neighboring WT cells in the 1st screen was below the threshold, while the amount provided in the 2nd screen surpassed this threshold, leading to the manifestation of adaptive growth."

12) Figure EV2 (previously S2A), the *nrg1* RNA-seq track seems to have different data than before - scale has gone from [-1, 1] to [-5,5], and many more reads and there are not as many gray reads on the antisense strand. How is this data different? The Figure caption does not mention the RNA-seq track and does not state what conditions it was collected in. I assume it was collected at 2 hours (the same as for the CHIP-seq). In general the timing for the CHIP-seq experiment should be mentioned in the caption or main text as it seems somewhat buried in the methods.

13) The authors clarified most of my concerns, for this point but I feel one concern is still outstanding. I think it would be useful to a reader to know how the authors chose the genes selected in EV2 as "previously determined target genes of these TFS". A reference for the paper(s) in which those target genes were identified would be helpful, or else a reference to the resource used (e.g. Pombase).

14) Although the author's clarified some of my concerns, my concern for Fig 4B still stands.

The authors state

"These differences in TF occupancy were positively correlated with target gene expression changes. That is, individual genes that were upregulated by NSF tended to be more strongly bound by the TFs, whereas downregulated genes were less occupied by the respective TFs (Fig. 4B)."

This is far from a general trend. The trend is only fully apparent for Hsr1, is half present for Adn2, Php3, and Atf1 and is not present for Reb1 and Fil1. I still think that the authors should be more precise about this statement.

"For Hsr1, and to some extent for Adn2, Php3 and Atf1, these differences in TF occupancy were positively correlated with target gene expression changes. That is, individual genes that were upregulated by NSF tended to be more strongly bound by the TFs, whereas downregulated genes were less occupied by the respective TFs (Fig. 4B)."

Some nuance is added in lines 285-294, but this is a bit removed from the general statement, and therefore the statement still feels too broad.

Additional Minor Points that may improve the manuscript:

m2.1) Fig 3F: Reviewer 2 asked about the *adh1* promoter as a control for the gene expression test. The authors responded that it is often used as a constitutive promoter in *S. pombe* and *S. cerevisiae*. There are two issues with this:

1) My understanding is that it is a fairly high expression promoter, and so it is not the best control to compare gene induction for a gene that has lower baseline expression like *nrg1*. I know that there are other constitutive promoters that have been validated at different levels in *S. pombe* and *S. cerevisiae*, (Fig 3 from <https://pubs.acs.org/doi/10.1021/acssynbio.3c00529> and Fig 3A from <https://pubs.acs.org/doi/full/10.1021/sb500366v>).

2) There is some evidence that it is not constitutive - from SGD: "Although originally thought to be expressed constitutively, ADH1 transcription is repressed when cells are grown on a non-fermentable carbon source such as ethanol or glycerol (Ref: <https://pubmed.ncbi.nlm.nih.gov/6337132/>)." One issue with this is that my understanding is that it is a high expression constitutive promoter

m2.2) Fig EV1E, a more natural way to show reproducibility of the RNA-seq results would be to plot the results from each replicate against each other with the expectation that they would lie on the $y=x$ line, rather than against a third dataset.

m2.3) Reviewer 2 had a problem with the wording "Independently of the carbon source" in the main text. The authors changed that wording in the text, but they did not make the change in the abstract.

m2.4) Line 115- Typo "than be"

m2.5) Line 166 - Typo "by the addition" should probably be something like "alongside the addition"

m2.6) Fig 4D is meant to show that in the mutant strains, mRNA levels are decreased.

1. The figure itself and the legend does not state that blue indicates NSF addition,.

2. The statistical tests are between samples with and without NSF addition, but the more relevant test would be between mRNA levels between WT and Deletion mutants.

Referee #3:

Below is our re-review of the manuscript, replying to the thread of our original comments and the author's response to these, point-per-point. Overall, we believe that the manuscript matured significantly and most of the remaining suggestions are minor.

Review RC-2023-02313 Nitrogen signaling factor
Reviewer #2 (Evidence, reproducibility and clarity (Required)):

This paper uses the model system *Schizosaccharomyces pombe* to investigate how the oxylipin nitrogen signaling factor functions to send signals and adapt the metabolism upon a change in nutrients in the environment. Combining genome-wide screens, RNA-sequencing and chemical biology, the authors find that the nitrogen signaling factor triggers a change from fermentation to a respiratory metabolism, through a direct interaction with a mitochondrial oxidoreductase Hmt2.

Major comments:

Overall, the manuscript lacks readability and coherence. Quite a lot of genes/TFs and proteins are mentioned, it is difficult to find a coherent story and clear overview and connection between these subparts. The manuscript would benefit from a general proposed scheme/working mechanism in the discussion and streamlining the results and data into a single biological storyline.

-> We value this feedback and have taken steps to enhance the readability of our manuscript. Additionally, we have included a cartoon illustrating a working model, which we hope readers will find helpful (see Supplementary Figure EV4).

-> The manuscript does indeed read more clearly now, with a more considered flow and explanation of the results. Thank you! We still have a few comments that should be addressed, and some small textual changes that could increase readability further. Several statements or results are not sufficiently clear, elaborated or nuanced. The paper would benefit from more explanation and discussion.

In the discussion section, the authors are not consistently referencing figures.

179-181: 'GO enrichment analysis of the 92 downregulated genes' but on line 167, it is '74 downregulated genes' that are mentioned. It is unclear where this difference in number of downregulated genes comes from. Similarly, for the upregulated genes. '156 genes' are mentioned on line 181 but only '98' on line 167.

-> Thank you for pointing this out. This was also pointed out by reviewer 1. As describe above, we performed GO analysis for all up- or down-regulated genes at 2h, 4h, and 6h, resulting in variability in the numbers. To avoid confusion, we used the 2h RNA-seq data for a GO re-analysis (analyzing 72 protein-coding genes among the up-regulated 99 genes and 38 protein-coding genes among the down-regulated 75 genes).

-> This is more clear now.

189: The statement that the downregulation of flocculation could serve to avoid mating, though sounding logical, is undermined by the finding that mating-related processes are upregulated in the experiment. I find this statement rather speculative

-> Mating-related GO terms were not enriched in the RNA-seq results at 2h. We do not mention mating anymore and instead focus on the reduced expression of genes related to cell aggregation mediated by NSF (page 10).

-> 213; 215-219: I still do not completely agree with this explanation. The expression data shows that there could be less flocculation when NSF is present to avoid cell-cell adhesion. But then, it is stated that cell-to-cell communication is linked to biofilm formation/mating/.... This is exactly what you would expect/want from a cell that needs the NSF signal to grow, no, to be closer/more adhered to each other? So, I still find it counterintuitive, and the text glances over this issue. Unless you actually do not want cell-cell adhesion for transferring signals to each other and grow. This is not well elaborated on in the text. Do you actually visually see the cells flocculating when adding NSF?

247-249: The statement is too broad, the effects are visible for maybe 3 TFs, the others don't seem to make a difference in occupancy. Also, why are these two highlighted genes of importance (*pex7+*, *yhb1+*), this is the first and last time they are mentioned?

-> It is correct that occupancy for most TFs that we have tested is not different. Changes in gene expression are easy to explain by TF occupancy, which we observe for a few. For this reason, these TFs were highlighted. We mention this more specifically in the revised version of the manuscript (page 13, 14) and we are showing examples in Fig 4C and 4D.

-> While I appreciate this expansion and revision, on line 279 it is curious that you take along *mei2*, *gsf2*, *pfl3* but then not the *nrp1* gene. Why not include this one as well? This brings more balance overall since figure 3 also looked at all 4.

254-256: The statement that these TFs are indispensable may be too strong. Right before, the authors showed that most of these TFs don't change occupancy (and especially *Fil1* and *Reb1* do not show a correlation with up- and down-regulated genes, nor does *Fil1* in *FigS2B* show a changed ChIP-seq signal).

→ These TFs were identified in our genetic screen. That is, they are indispensable.

-> I noticed that whole section has been re-written, excluding the phrasing and usage of 'indispensable' to a more neutral way, which I appreciate.

365: 'independently of the carbon source'. As far as we can see, all experiments were performed using glucose as the carbon source, so this statement seems too strong as there is no clear proof for this. This could be an easy extra experiment to perform these tests on media with other carbon sources than glucose?

-> We agree. We have changed "independently of the carbon source" to "without any change in the carbon source" to avoid misunderstandings (page 19).

-> OK. The page number is 18 instead of 19. Could you please also change this statement in the abstract line 33 since there is remained unchanged.

Fig1E: It is not clear if the experiment was performed with the Wild-Type or a deletion strain. In the case of the WT, colonies grew in the media not containing NSF but in Fig5E and Fig5I, the WT did grow in the media not containing NSF. It could be more relevant to plate out 1 colony like in the second screen. Thus, unless different strains were used for both experiments, the results seem inconsistent with each other, which is not mentioned in the manuscript.

We have also observed NSF-independent adaptive growth by cultivating cells for an extended period without the addition of NSF. However, to avoid confusion, we have refrained from discussing this NSF-independent adaptive growth in this section. Consequently, we have modified the figure used for clarity (Figure 1E). Additionally, we have included details about the strains used in the Figure legend (page 32, 34, and 35). See also our responses to reviewer 1.

-> OK

Fig1E, Fig5B, Fig5E, and Fig5I: For these experiments, different nitrogen concentrations were used depending on the media, but this has not been addressed/mentioned in the manuscript.

-> We added the respective information to Material & Methods.

-> I still have unaddressed questions on this. Fig1: Why are there differences in the concentration of ammonium used in the different screens? This should perhaps be elaborated on in the text. In M&Ms, it now is indeed briefly mentioned without extra explanation that a distinction is made between normal and severe NCR conditions. It would be great if the authors could add their reasoning and distinction between those two conditions in the main text.

Minor comments:

53-57: I would like more elaboration on why CCR and NCR are important for virulence of human pathogens or relevant for industrial applications, and link back to this in the discussion. Otherwise, it is superfluous to include this in the introduction.

-> While lines 52-54 emphasize the significance of CCR and NCR in human pathogens, underscoring their crucial role in rapidly adapting to niche environments during infection, lines 58-63 highlight their importance in industrial settings. Specifically, in the brewing industry, CCR is indispensable for efficiently synthesizing ethanol through the fermentation pathway, utilizing carbon sources. Similarly, NCR is vital for optimizing nitrogen source utilization and biomass production. However, the fine-tuning of NCR becomes paramount, as an excessive presence can lead to the accumulation of substances like urea and proline, negatively impacting the quality of products such as wine. We have made changes to the introduction accordingly on pages 3-4.

-> OK

108: Does having a h- library have any impact on the outcome compared to the original h+ library?

-> Because h- and h+ strains grow slightly different, *S. pombe* libraries are typically aligned on one mating type. In additional experiments, h+ strains were occasionally used as hosts; however, as of now, no significant phenotypic differences have been observed between h- and h+. On page 6, the purpose behind aligning the library with h- is explained.

-> Thank you

170: Why would only one of the 117 NSF-linked genes change expression in the RNA-sequencing experiment? Any explanation as to why the expression remains unchanged for the 116 other NSF-linked genes?

Rev_Com_number: RC-2023-02313

New_manu_number: EMBOJ-2024-117143R

Corr_author: Bühler

Title: Nitrogen signaling factor triggers a respiration-like gene expression program

Manuscript EMBOJ-2024-117143-T-R
Final Revision

Point-by-point response

Our responses to the final remarks are highlighted in blue.

Referee #1:

GENERAL SUMMARY

Within this manuscript, the authors study a cell-to-cell communication process, by which nitrogen catabolite repression can be counteracted by a small molecule called NSF. Specifically, the authors demonstrate here that NSF up-regulates respiratory metabolism as a mechanism to overcome the repression by high nitrogen of amino acid internalization. Several of our comments are answered, but two important points are unaddressed: the role of transcription factors in the nuclear response is very poorly developed (points 8-9), and the role of such transcriptional response in bypassing the nitrogen catabolite repression is unclear, based on the response to our queries (point 12). For the sake of simplicity, we include here the number used in our previous correspondence, and add our new comments at the end (R):

MAJOR COMMENTS

3) Comparing the transcriptomic response to that of Malecki et al 2020 in response to Antimycin A (EMBO Rep. 21:e50845) would be useful.
-> We are uncertain why this would be useful. Moreover, in the transcriptome analysis by Malecki et al. 2020, there is no direct comparison between antiA-treated and untreated samples from the same time points. Nevertheless, we compared the up- or down-regulated genes identified in Malecki et al. 2016 under antiA treatment with those under NSF treatment. This did not reveal any significant differences.

Reviewer (R): Antimycin A treatment activates a retrograde response involved in mitochondrial metabolism, only part of which is repressed in the presence of arginine. The data shown in Malecki 2020 and Malecki 2016 indicate that ETC blockage leads to downregulation of ETC subunits and upregulation of stress genes. Let me list a few genes that DO overlap with your genes in Table S2, 'up-regulated genes':

obr1, brf1, gst2, SPBC23G7.10c, ght5, SPBC12C2.04.

We remain uncertain about drawing strong conclusions from comparing experiments that were conducted differently and, therefore, refrain from conducting further analysis.

4) The optical densities and whether NCR has been induced has to be clearly specified in each experiment. For instance, RNA seq data. Line 165: for the transcriptome experiment, NSF is added or not to low density cells (not indicated in results, figure legend nor materials and methods). Should addition of NSF to wild-type strain en MM trigger the same transcriptomic changes?

-> Thank you for pointing this out. We have added this information ("at low cell density (OD 0.01)") to the figure legend of the RNA-seq experiments (page 32). Additionally, we have added qRT-PCR results for prototrophs in new Figure 3C and mentioned it on page 11.

(R): We do not see the explanation about cell densities on Fig. 1G. We would have appreciated if the author helped us with the review process by adding the changes in the letter.

The information on cell density can be found in Materials and Methods. It is now also mentioned in the figure legend.

7) Fig. 2BC: Venn diagrams should be more useful to demonstrate overlap with the Malecki data.

-> We have added Venn diagrams to supplementary Figure EV1C and mention it on page 9.

(R): Send main Fig. 2B, with poor correlation of 0.24, to EV1C. Move current EV1C to Figure 2 in substitution.

We have swapped Figure 2b with EV1C as suggested.

8) Fig. 4A: not very useful

-> We find it useful because it displays general trends in TF occupancy.

(R): The characterization/role of the 6 TFs on the response to NSF is poor. We understand that the 6 (8, if we include Php5 and Adn3) TFs have been selected in the screening, but ChIP seq data has failed to demonstrated occupancy as a factor modulating the response; probably activity is important. Current Fig. 4A (formerly at EV) can stay there, to highlight that Hsr1 and Adn2 may have an impact on some chromosomal loci. Current Fig. 4B has to be sent to EV. The ChIP seq data is not useful. If current Fig 4C is the best you have, send Reb1 and Fil1 data to EV. Then, develop a bit more the role of Δ TFs on gene expression changes (see below, point 9).

The ChIP-seq data has not "failed" to show anything. It indicates that TF occupancy remains largely unchanged, meaning that differences in TF occupancy positively

correlate with target gene expression changes for only a few TFs. This is a relevant finding (see reviewer 2, comment 14).

9) Is Hsr1 required for some of the RNA seq changes upon NSF addition?
Same with other TFs

-> Yes, this is the case for individual genes that we have checked by qRT-PCR (both up and down-regulation by NSF addition), but we can't generalize because we do not have RNA-seq data from TF knock-out strains. For example, the *nrg1* gene is no longer induced by NSF in the absence of Hsr1; or in absence of Hsr1, Adn2, or Php3, *gsf2* and *pfl3* expression is as low as after treating wt cells with NSF (i.e. Hsr1 and Adn2 are required for the expression of the *gsf2* and *pfl3* genes). We show decreased *gsf2* and *pfl3* expression levels in *hsr1* Δ , *adn2* Δ , or *php3* Δ mutants in Figure 4D.

(R): Why only genes downregulated are shown? If the authors do not want to prepare RNAseq for this manuscript, why don't they qPCR the most up- and down-regulated genes (Table S2) in WT and Δ TFs, at least Δ hsr1, Δ adn2 and Δ php3? Of course, *nrg1* has to be shown. Current Fig. 4D shows that Hsr1, Adn2 and Php3 are activators, probably, of these two down-regulated genes; but the author says that the *nrg1* gene is no longer induced by NSF in the absence of Hsr1. Then, at least Hsr1 can activate *nrg1* after NSF, but activate *gsf2* and *pfl3* under basal conditions?

We included *nrg1* expression in *hsr1* Δ , *adn2* Δ , and *php3* Δ strains in Fig. 4D.

10) Line 287: '...it is tempting to speculate that Ayr1 dampens adaptive responses by metabolizing NSF'. Calculating MEC for NSF in delta *ayr1* and in cells over-expressing Ayr1 would be required to confirm this speculation. According to Pombase, cells lacking Ayr1 have their respiratory functions compromised (no growth in galactose, glycerol...), why is so? The opposite should be expected, if NSF-mediated respiration is enhanced in this background.

-> We are purposely speculating here, because much more work will be needed to firmly show that Ayr1 metabolizes NSF (including in vitro biochemical experiments). Because this would go beyond the scope of this study, we would like to leave this speculative.

Nevertheless, we performed the suggested experiments. We checked MEC through a growth assay using the *ayr1* Δ mutant, but no decrease in MEC was observed with the *ayr1* Δ mutation. It is possible that the result could be attributed to the impact of respiratory deficiency, as indicated by the growth delay in glycerol or galactose conditions on Pombase. Overexpression of Ayr1 did indeed affect NSF-mediated growth and NSF-responsive gene expression, fueling our speculation. We show these results in Figure 5F

and 5G and mention them on page 17.

(R): You mention them on page 15, not 17. I would have appreciated copy-pasting the changes in the letter, for the sake of simplicity.

Okay.

12) The data regarding Hmt2 is very interesting. As for delta ayr1, delta hmt2 cells cannot grow in glycerol nor galactose according to Pombase. Is the result shown in Fig. 5J (lack of NSF-dependent activation of nrt1 and mei2 in delta hmt2) a consequence of the absence of the NSF receptor, or is it due to the lack of respiration of this background? Is delta hmt2 really auxotrophic for Cys? Why? In this background, H₂S should be enhanced, and Cys and Met biosynthesis improved. In fact, in one manuscript these cells grow fine in SG minimal media (Mol Microbiol 01 42:29), while another report indicates they are auxotrophic for Cys (Genes to Cells 2016, 21:530).

-> As an additional experiment, we confirmed the responsiveness of nrt1 (nrg1) in the mutant strains with disrupted ETC-related genes, which are required for NSF-dependent adaptive growth. Despite the disruption of these genes, nrt1(nrg1) still exhibited responsiveness, suggesting that the effect is not solely due to the lack of the respiratory chain.

(R): Where is the additional experiment ? Not added ? Which mutants were used ? Is the gene response blocked with antimycin ? This info has to be included. And is relevant for the final model and the Discussion. If this occurs for most of the NSF genes, the upgrade in respiratory function would trigger a change in the gene expression program, but the transcriptional response would not be required for unblocking the NCR. The model in Figure EV4 and the Discussion do not support the fact that mutants with deficient ETC are deficient in NSF-dependent adaptive growth, but nrg1 is still responding.

We confirmed the NSF responsiveness of *nrg1* using ETC mutants involved in NSF-mediated adaptive growth from our growth screening. We consider this data preliminary (e.g. we have no genome-wide expression data) and thus do not want to include it in the current paper. Obviously additional work will be needed to dissect the role of Hmt2, which goes beyond to scope of this study.

MINOR COMMENTS

- 1) Who triggers NCR? Analysis of 137 genes in Figure 1b.
- 2) Synthesis of NSF: how is it regulated, where does it come from?
- 3) NCR impairs import of BCAA. How are the aa importers such as Cat1 or Agp3 eliminated from the plasma membrane? Transporter internalization,

degradation, transcriptional repression... And how does NSF block the NCR regarding aa uptake? Or aa usage?

(R): Add to Discussion a sentence or paragraph about these future directions.

We could only speculate wildly, which we refrain from doing.

Referee #2:

I have previously provided a general summary and opinion about the significance of the study and outlined my concern in my initial review for Review Commons.

The following comments pertain to the authors response to my comments and those of the other reviewers from that initial reviewers. In general the revisions the authors have done have improved and clarified the study, and I still believe that it would be of broad interest to the microbiology community, those interested in metabolism and metabolic regulation, and also those interested in cell-cell communication.

For the most part the authors have addressed my major concerns, but the following concerns still remain for me. Numbers refer to the points from my original review:

1) The authors have partially addressed my concern by stating that reliance on BCAA synthesis during "adaptation" under NCR conditions, but it is unclear what adaptation refers to in this context, and it is unclear what NCR conditions mean. I assume that NCR conditions means Ammonium Chloride in the presence of Leucine. I think the authors should be more clear.

To rephrase my concern, the way that it is worded implies that NCR is happening in WT cells grown under NCR conditions, however WT cells secrete NSF to allow bypass of NCR for neighbouring Leu2 auxotrophs. Why shouldn't they do the same within a culture of WT cells? In other words, **I think it is reasonable to believe, in light of the evidence the authors present, that NCR is a phenotype that only exists in Leu2 auxotrophs**, because NCR bypass is in effect in WT cells. Therefore WT *S. pombe* cells ought to be able to import BCAAs, rather than having to synthesize it themselves. I am not as familiar with this topic as the authors, so I defer to their judgement, but I would suggest they soften the statement even more to the following:

Thus, *S. pombe* cells may rely predominantly on their own BCAA synthesis for at least the initial stages of growth under NCR conditions (high NH₄CL₂ in the presence of BCAAs).

As following your suggestion, we changed to “Thus, *S. pombe* cells may rely predominantly on their own BCAA synthesis for at least the initial stages of adaptation growth under NCR conditions”.

6) Although the authors did incorporate some of my feedback about the 57 strains that can't grow in the presence of a WT strain, but can grow in the presence of NSF, they still say for these 57 "Yet, because they are needed for adaptive growth (1st screen), they are likely required to respond to a transmissible signal that is different from NSF)"

I don't understand how being needed for adaptive growth is evidence of being required to respond to a transmissible signal different from NSF, and even if this is possible, it doesn't appear to me to be likely. Clearly these strains can respond to NSF, and clearly the WT strains secrete NSF. I still feel it is likely that it is just a matter of a threshold which the authors acknowledge.

I think that this is too strongly worded.

I would find something like this acceptable:

"Yet, because they are needed for adaptive growth (1st screen), they may be required to respond to a transmissible signal that is different from NSF), or else they could alter the sensitivity of the strain to NSF. We speculate..."

Also when they discuss the idea of the deletions altering the threshold of NSF I think it is a bit confusing. I don't think the mutants are surpassing a threshold, but rather the threshold of NSF required for avoidance of NCR is different in the mutants. They say:

"We speculate that these mutants fell below the threshold of NSF required for adaptive growth in the 1st screen, whereas in the 2nd screen they surpassed this threshold, leading to the manifestation of adaptive growth."

I believe that they may mean something more like:

"We speculate that these mutants are less sensitive to NSF for avoidance of NCR, and the amount of NSF provided by neighboring WT cells in the 1st screen was below the threshold, while the amount provided in the 2nd screen surpassed this threshold, leading to the manifestation of adaptive growth."

We changed these sentences to “Yet, because they are needed for adaptive growth (1st screen), they are likely required to respond to a transmissible signal that is different from NSF (Sun et al, 2016; Chiu et al, 2022) or else they could alter the sensitivity of the strain to NSF. We speculate that these mutants are less sensitive to NSF for avoidance of NCR, and the amount of NSF provided by neighboring WT cells in the 1st screen was below the threshold, while the amount provided in the 2nd screen surpassed this threshold, leading to the manifestation of adaptive growth”.

12) Figure EV2 (previously S2A), the *nrg1* RNA-seq track seems to have different data than before - scale has gone from [-1, 1] to [-5,5], and many more reads and there are not as many gray reads on the antisense strand.

How is this data different? The Figure caption does not mention the RNA-seq track and does not state what conditions it was collected in. I assume it was collected at 2 hours (the same as for the CHIP-seq). In general the timing for the CHIP-seq experiment should be mentioned in the caption or main text as it seems somewhat buried in the methods.

We added "for 2 hours under NCR condition" in main text, "in cells treated with MeOH or NSF for 2 hours at low cell density (OD 0.01)" in figure legend of Fig 4A, "The RNA-seq tracks were derived from RNA-seq data of cells under NCR conditions with or without NSF for 2 hours (see Fig. 2A)" in figure legend of Fig 4C for RNA-seq track, and "The RNA-seq tracks were derived from RNA-seq data of cells under NCR conditions with or without NSF for 2 hours (for genes other than *nrg1*) and 6 hours (for *nrg1*) (see Fig. 2A and Fig. EV1B)" in figure legend of Fig. EV2 for RNA-seq track.

13) The authors clarified most of my concerns, for this point but I feel one concern is still outstanding. I think it would be useful to a reader to know how the authors chose the genes selected in EV2 as "previously determined target genes of these TFS". **A reference for the paper(s) in which those target genes were identified would be helpful, or else a reference to the resource used (e.g. Pombase).**

There references concerned are mentioned in the Fig EV2 legend.

14) Although the author's clarified some of my concerns, my concern for Fig 4B still stands.

The authors state

"These differences in TF occupancy were positively correlated with target gene expression changes. That is, individual genes that were upregulated by NSF tended to be more strongly bound by the TFs, whereas downregulated genes were less occupied by the respective TFs (Fig. 4B)." This is far from a general trend. The trend is only fully apparent for Hsr1, is half present for Adn2, Php3, and Atf1 and is not present for Reb1 and Fil1. I still think that the authors should be more precise about this statement.

"For Hsr1, and to some extent for Adn2, Php3 and Atf1, these differences in TF occupancy were positively correlated with target gene expression changes. That is, individual genes that were upregulated by NSF tended to be more strongly bound by the TFs, whereas downregulated genes were less occupied by the respective TFs (Fig. 4B)."

Some nuance is added in lines 285-294, but this is a bit removed from the general statement, and therefore the statement still feels too broad.

Thank you. We have amended the text accordingly (added "For Hsr1, and to some extent for Adn2, Php3 and Atf1").

Additional Minor Points that may improve the manuscript:

m2.1) Fig 3F: Reviewer 2 asked about **the adh1 promoter** as a control for the gene expression test. The authors responded that it is often used as a constitutive promoter in *S. pombe* and *S. cerevisiae*. There are two issues with this:

1) My understanding is that it is a fairly high expression promoter, and so it is not the best control to compare gene induction for a gene that has lower baseline expression like *nrg1*. I know that there are other constitutive promoters that have been validated at different levels in *S. pombe* and *S. cerevisiae*, (Fig 3 from <https://pubs.acs.org/doi/10.1021/acssynbio.3c00529> and Fig 3A from <https://pubs.acs.org/doi/full/10.1021/sb500366v>).

2) There is some evidence that it is not constitutive - from SGD: "Although originally thought to be expressed constitutively, ADH1 transcription is repressed when cells are grown on a non-fermentable carbon source such as ethanol or glycerol (Ref: <https://pubmed.ncbi.nlm.nih.gov/6337132/>)." One issue with this is that my understanding is that it is a high expression constitutive promoter

We agree that this is an artificial configuration. Nevertheless, this experiment supports the importance of the *nrg1* promoter in receiving the NSF mediated signal.

m2.2) **Fig EV1E, a more natural way to show reproducibility of the RNA-seq results would be to plot the results from each replicate against each other with the expectation that they would lie on the y=x line, rather than against a third dataset.**

We included a heatmap of pairwise pearson correlation coefficients for all samples in Fig. EV1.

m2.3) Reviewer 2 had a problem with the wording "Independently of the carbon source" in the main text. The authors changed that wording in the text, **but they did not make the change in the abstract.**

We made this amendment to the abstract.

m2.4) Line 115- Typo "than be" **corrected**

m2.5) Line 166 - Typo "by the addition" should probably be something like "alongside the addition"

corrected

m2.6) Fig 4D is meant to show that in the mutant strains, mRNA levels are decreased.

1. The figure itself and the legend does not state that blue indicates NSF addition,.

We included this information in the figure itself.

2. The statistical tests are between samples with and without NSF addition, but the more relevant test would be between mRNA levels between WT and Deletion mutants.

We included the statistical tests between WT and mutants. As a result, these plots are now quite busy. We leave it up to the editor to decide whether or not it is useful to show all p-values.

Referee #3:

Below is our re-review of the manuscript, replying to the thread of our original comments and the author's response to these, point-per-point. Overall, we believe that the manuscript matured significantly and most of the remaining suggestions are minor.

Review RC-2023-02313 Nitrogen signaling factor

Reviewer #2 (Evidence, reproducibility and clarity (Required)):

This paper uses the model system *Schizosaccharomyces pombe* to investigate how the oxylipin nitrogen signaling factor functions to send signals and adapt the metabolism upon a change in nutrients in the environment. Combining genome-wide screens, RNA-sequencing and chemical biology, the authors find that the nitrogen signaling factor triggers a change from fermentation to a respiratory metabolism, through a direct interaction with a mitochondrial oxidoreductase Hmt2.

Major comments:

Overall, the manuscript lacks readability and coherence. Quite a lot of genes/TFs and proteins are mentioned, it is difficult to find a coherent story and clear overview and connection between these subparts. The manuscript would benefit from a general proposed scheme/working mechanism in the discussion and streamlining the results and data into a single biological storyline.

-> We value this feedback and have taken steps to enhance the readability of our manuscript. Additionally, we have included a cartoon illustrating a working model, which we hope readers will find helpful (see Supplementary Figure EV4).

-> The manuscript does indeed read more clearly now, with a more considered flow and explanation of the results. Thank you! We still

have a few comments that should be addressed, and some small textual changes that could increase readability further.

Several statements or results are not sufficiently clear, elaborated or nuanced. The paper would benefit from more explanation and discussion.

In the discussion section, the authors are not consistently referencing figures.

179-181: 'GO enrichment analysis of the 92 downregulated genes' but on line 167, it is '74 downregulated genes' that are mentioned. **It is unclear where this difference in number of downregulated genes comes from.** Similarly, for the upregulated genes. '156 genes' are mentioned on line 181 but only '98' on line 167.

-> Thank you for pointing this out. This was also pointed out by reviewer 1. As describe above, we performed GO analysis for all up- or down-regulated genes at 2h, 4h, and 6h, resulting in variability in the numbers. To avoid confusion, we used the 2h RNA-seq data for a GO re-analysis (analyzing 72 protein-coding genes among the up-regulated 99 genes and 38 protein-coding genes among the down-regulated 75 genes).

-> This is more clear now.

189: The statement that the downregulation of flocculation could serve to avoid mating, though sounding logical, is undermined by the finding that mating-related processes are upregulated in the experiment. I find this statement rather speculative

-> Mating-related GO terms were not enriched in the RNA-seq results at 2h. We do not mention mating anymore and instead focus on the reduced expression of genes related to cell aggregation mediated by NSF (page 10).

-> 213; 215-219: I still do not completely agree with this explanation. The expression data shows that there could be less flocculation when NSF is present to avoid cell-cell adhesion. But then, it is stated that cell-to-cell communication is linked to biofilm formation/mating/.... This is exactly what you would expect/want from a cell that needs the NSF signal to grow, no, to be closer/more adhered to each other? So, I still find it counterintuitive, and the text glances over this issue. Unless you actually do not want cell-cell adhesion for transferring signals to each other and grow. This is not well elaborated on in the text. **Do you actually visually see the cells flocculating when adding NSF?**

No, we do not observe any obvious flocculating phenotype.

247-249: The statement is too broad, the effects are visible for maybe 3 TFs, the others don't seem to make a difference in occupancy. Also, why

are these two highlighted genes of importance (pex7+, yhb1+), this is the first and last time they are mentioned?

-> It is correct that occupancy for most TFs that we have tested is not different. Changes in gene expression are easy to explain by TF occupancy, which we observe for a few. For this reason, these TFs were highlighted. We mention this more specifically in the revised version of the manuscript (page 13, 14) and we are showing examples in Fig 4C and 4D.
-> While I appreciate this expansion and revision, on line 279 it is curious that you take along mei2, gsf2, pfl3 but then not the nrg1 gene. Why not include this one as well? This brings more balance overall since figure 3 also looked at all 4.

We moved *nrg1* from Fig. EV2 to Fig. 4C.

254-256: The statement that these TFs are indispensable may be too strong. Right before, the authors showed that most of these TFs don't change occupancy (and especially Fil1 and Reb1 do not show a correlation with up- and down-regulated genes, nor does Fil1 in FigS2B show a changed ChIP-seq signal).

→ These TFs were identified in our genetic screen. That is, they are indispensable.

-> I noticed that whole section has been re-written, excluding the phrasing and usage of 'indispensable' to a more neutral way, which I appreciate.

365: 'independently of the carbon source'. As far as we can see, all experiments were performed using glucose as the carbon source, so this statement seems too strong as there is no clear proof for this. This could be an easy extra experiment to perform these tests on media with other carbon sources than glucose?

-> We agree. We have changed "independently of the carbon source" to "without any change in the carbon source" to avoid misunderstandings (page 19).

-> OK. The page number is 18 instead of 19. Could you please also change this statement in the abstract line 33 since there is remained unchanged.

Done.

Fig1E: It is not clear if the experiment was performed with the Wild-Type or a deletion strain. In the case of the WT, colonies grew in the media not containing NSF but in Fig5E and Fig5I, the WT did grow in the media not containing NSF. It could be more relevant to plate out 1 colony like in the second screen. Thus, unless different strains were used for both experiments, the results seem inconsistent with each other, which is not mentioned in the manuscript.

We have also observed NSF-independent adaptive growth by cultivating cells for an extended period without the addition of NSF. However, to avoid confusion, we have refrained from discussing this NSF-independent adaptive growth in this section. Consequently, we have modified the figure used for clarity (Figure 1E). Additionally, we have included details about the strains used in the Figure legend (page 32, 34, and 35). See also our responses to reviewer 1.

-> OK

Fig1E, Fig5B, Fig5E, and Fig5I: For these experiments, different nitrogen concentrations were used depending on the media, but this has not been addressed/mentioned in the manuscript.

-> We added the respective information to Material & Methods.

-> I still have unaddressed questions on this. Fig1: Why are there differences in the concentration of ammonium used in the different screens? This should perhaps be elaborated on in the text. In M&Ms, it now is indeed briefly mentioned without extra explanation that a distinction is made between normal and severe NCR conditions. **It would be great if the authors could add their reasoning and distinction between those two conditions in the main text.**

We added this information in the main text.

Minor comments:

53-57: I would like more elaboration on why CCR and NCR are important for virulence of human pathogens or relevant for industrial applications, and link back to this in the discussion. Otherwise, it is superfluous to include this in the introduction.

-> While lines 52-54 emphasize the significance of CCR and NCR in human pathogens, underscoring their crucial role in rapidly adapting to niche environments during infection, lines 58-63 highlight their importance in industrial settings. Specifically, in the brewing industry, CCR is indispensable for efficiently synthesizing ethanol through the fermentation pathway, utilizing carbon sources. Similarly, NCR is vital for optimizing nitrogen source utilization and biomass production. However, the fine-tuning of NCR becomes paramount, as an excessive presence can lead to the accumulation of substances like urea and proline, negatively impacting the quality of products such as wine. We have made changes to the introduction accordingly on pages 3-4.

-> OK

108: Does having a h- library have any impact on the outcome compared to the original h+ library?

-> Because h- and h+ strains grow slightly different, *S. pombe* libraries are typically aligned on one mating type. In additional experiments, h+ strains

were occasionally used as hosts; however, as of now, no significant phenotypic differences have been observed between h- and h+. On page 6, the purpose behind aligning the library with h- is explained.

-> Thank you

170: Why would only one of the 117 NSF-linked genes change expression in the RNA-sequencing experiment? Any explanation as to why the expression remains unchanged for the 116 other NSF-linked genes?

We answered this question in the previous point-by-point response.

Dear Dr Bühler,

Thank you for submitting your amended manuscript for consideration by the EMBO Journal. It has now been reassessed by three referees whose comments are enclosed. As you will see, all three referees are broadly in favour of publication, pending satisfactory minor revision.

Please carefully consider the remaining points by referee #2 and amend the manuscript by revisiting discussion of the findings and data -statistics presentation.

Upon resubmission, we are happy to swiftly proceed with formal acceptance and production of the study.

Please let me know should there be any questions related.

We look forward to your final resubmission.

Best regards,

Daniel Klimmeck

Daniel Klimmeck, PhD
Senior Editor
The EMBO Journal

Referee #1:

No additional comments.

Referee #2:

I have previously reviewed this manuscript, once for Review Commons and then a follow up with EMBO Journal. I still believe that it would be of broad interest to the microbiology community, those interested in metabolism and metabolic regulation, and also those interested in cell-cell communication.

I believe the authors efforts in response to reviewer comments have improved the paper, and I have no more major issues with the paper, however I do still have the following non-essential suggestions to improve the study.

The numbers refer to my previous comments:

12) I am glad the authors for added details on when each experiment was collected. It is unclear within the manuscript why they chose to use 6 hours for nrg1 for RNA-seq, while just showing Chip-Seq traces for 2 hours. I assume it is because the nrg1 expression at 2 hours was very low. This is a bit misleading - a reader looking at fig 4C will assume that the chip-seq and RNA-seq were taken at similar time points. I believe the original version of the manuscript contained RNA-seq data from the 2 hour time point, which had extremely low expression levels. In my opinion it would be better to either show the 2 hour data, or else make clear on the figure that the RNA-seq data is from a very different time point.

M2.2) In figure EV1 the authors include a heatmap of pairwise pearson correlation coefficients for two repeated RNA-seq experiments (Exp1 and Exp2) conducted with and without NSF and compared to data from another study in which RNA-seq data was collected with either glycerol or glucose as a carbon source. The Pearson R value between these replicates and the external dataset was quite low (0.31 and 0.42). Perhaps more worrying, the correlation between replicates was also fairly low

(0.57). In fig EV1A, the correlation coefficient was 0.93 between two biologically different conditions - LFC of 6h vs 2h compared to LFC of 2h vs 0h, although perhaps this is an unfair comparison because I assume fig EV1A reports LFC after information from all biological replicates is integrated within the framework of DEseq. This makes me worry that:

1) The conclusion that "NSF triggers a change from a fermentation- to a respiration-like gene expression program" is a bit weak based. The data from figures 1E, 1F and 1G make it clear that NSF triggers a change to a more respiration-like state, but the gene expression program of the glycerol state and the NSF induced state seem broadly different.

2) There is some unknown source of biological variability between the authors' RNA-seq replicates.

M2.6) The authors added additional statistical tests in Figure 4D, but they worries that the resulting figures are too busy and leave it up to the editor to decide which statistical tests to keep. I would suggest that the author remove the statistical tests that don't make the point that that figure is trying to make.

The authors use the figure to make two points:

"transcription of *gsf2* and *pfl3*, mRNA levels were strongly reduced in the respective TF knock out strains (Fig. 4D)."

And

"*nrg1* expression was reduced in *hsr1Δ* and *php3Δ* cells, and it was hardly induced by NSF, underscoring the regulatory role of these TFs"

Thus it seems the most necessary p-values to include would be the blue ones for the plots of *gsf2* and *pfl3* transcript levels, and the red ones for the plot of *nrg1* transcript levels. If the authors want to keep more p-values for comparison, they could make the plots bigger, or remove some plots to EV figures.

Finally, looking through the authors' responses to the other reviewers, I have additional concerns:

Reviewer 1 pointed out that the authors ought to compare their data with data from Malecki et al 2020 with or without Antimycin A treatment

The authors respond that they "remain uncertain about drawing strong conclusions from comparing experiments that were conducted differently and, therefore, refrain from conducting further analysis." This response seems a bit dismissive of the reviewer's concerns, and also slightly inconsistent as the authors do seem to draw strong conclusions by comparing their data to another dataset from Malecki et al 2016 (Fig 2C, Fig EV1C and EV1D).

The additional analysis requested by the reviewer doesn't seem overly onerous - I would have imagined that they could have done a similar analysis as was done in figures EV1C and EV1D and at least presented the results to the reviewers if they do not merit inclusion in the manuscript.

Referee #3:

The authors have addressed all our concerns. Yet, I do think that this is not the case for some concerns raised by the two other reviewers, and in my opinion, the authors should resolve these since the comments seem pertinent.

Manuscript EMBOJ-2024-117143-T-R
Final Revision

Point-by-point response

We thank all three reviewers for the time spent on our manuscript and their constructive criticism, which has improved our paper. Thank you all very much! Our responses to the final remarks are highlighted in blue.

Referee #1:

No additional comments.

Referee #2:

I have previously reviewed this manuscript, once for Review Commons and then a follow up with EMBO Journal. I still believe that it would be of broad interest to the microbiology community, those interested in metabolism and metabolic regulation, and also those interested in cell-cell communication. I believe the authors efforts in response to reviewer comments have improved the paper, and I have no more major issues with the paper, however I do still have the following non-essential suggestions to improve the study.

The numbers refer to my previous comments:

12) I am glad the authors for added details on when each experiment was collected. It is unclear within the manuscript why they chose to use 6 hours for *nrg1* for RNA-seq, while just showing Chip-Seq traces for 2 hours. I assume it is because the *nrg1* expression at 2 hours was very low. This is a bit misleading - a reader looking at fig 4C will assume that the chip-seq and RNA-seq were taken at similar time points. I believe the original version of the manuscript contained RNA-seq data from the 2 hour time point, which had extremely low expression levels. In my opinion it would be better to either show the 2 hour data, or else make clear on the figure that the RNA-seq data is from a very different time point.

This is correct, induction of *nrg1* expression was more prominent after 6 hours. In addition to mentioning this in the legend only, we have now indicated the time points also in the figure itself (Fig. 4C).

M2.2) In figure EV1 the authors include a heatmap of pairwise pearson correlation coefficients for two repeated RNA-seq experiments (Exp1 and Exp2) conducted with and without NSF and compared to data from another study in which RNA-seq data was collected with either glycerol or glucose

as a carbon source. The Pearson R value between these replicates and the external dataset was quite low (0.31 and 0.42). Perhaps more worrying, the correlation between replicates was also fairly low (0.57). In fig EV1A, the correlation coefficient was 0.93 between two biologically different conditions - LFC of 6h vs 2h compared to LFC of 2h vs 0h, although perhaps this is an unfair comparison because I assume fig EV1A reports LFC after information from all biological replicates is integrated within the framework of DEseq. This makes me worry that:

1) The conclusion that "NSF triggers a change from a fermentation- to a respiration-like gene expression program" is a bit weak based. The data from figures 1E, 1F and 1G make it clear that NSF triggers a change to a more respiration-like state, but the gene expression program of the glycerol state and the NSF induced state seem broadly different.

Yes, compared to technical replicates these R values could be considered low. However, taking into account that we used minimum media (EMM) with high ammonium and glucose as the carbon source, whereas Malecki 2016 used rich media (YE) with no ammonium and glycerol as the carbon source, as well as the different culture conditions (including OD and cultivation time), these data sets are still decently correlated.

2) There is some unknown source of biological variability between the authors' RNA-seq replicates.

The source for this is likely twofold. First, these are true biological replicates (NSF treatments were not performed simultaneously, but at different days). Secondly, the fold change in RNA expression can be quite variable at the early 2h time point. I.e. although qualitatively the same, quantitatively there can be quite some variability. This is reflected by the R value of 0.57.

M2.6) The authors added additional statistical tests in Figure 4D, but they worries that the resulting figures are too busy and leave it up to the editor to decide which statistical tests to keep. I would suggest that the author remove the statistical tests that don't make the point that that figure is trying to make.

The authors use the figure to make two points:

"transcription of *gsf2* and *pfl3*, mRNA levels were strongly reduced in the respective TF knock out strains (Fig. 4D)."

And

"*nrg1* expression was reduced in *hsr1* Δ and *php3* Δ cells, and it was hardly induced by NSF, underscoring the regulatory role of these TFs"

Thus it seems the most necessary p-values to include would be the blue ones for the plots of *gsf2* and *pfl3* transcript levels, and the red ones for the plot of *nrg1* transcript levels. If the authors want to keep more p-values for comparison, they could make the plots bigger, or remove some plots to EV figures.

Thank you for your advice. We rearranged the figure to make the plots bigger and keep all the tests.

Finally, looking through the authors' responses to the other reviewers, I have additional concerns:

Reviewer 1 pointed out that the authors ought to compare their data with data from Malecki et al 2020 with or without Antimycin A treatment

The authors respond that they "remain uncertain about drawing strong conclusions from comparing experiments that were conducted differently and, therefore, refrain from conducting further analysis." This response seems a bit dismissive of the reviewer's concerns, and also slightly inconsistent as the authors do seem to draw strong conclusions by comparing their data to another dataset from Malecki et al 2016 (Fig 2C, Fig EV1C and EV1D).

The additional analysis requested by the reviewer doesn't seem overly onerous - I would have imagined that they could have done a similar analysis as was done in figures EV1C and EV1D and at least presented the results to the reviewers if they do not merit inclusion in the manuscript.

We apologize if we came across as dismissive, it was certainly not our intention. As mentioned in our first responses to this comment, in the transcriptome analysis by Malecki et al there was no direct comparison between antiA-treated and untreated samples from the same time points, which would have been a nice control. As we mentioned in our initial response, a comparison of the available datasets did not reveal any significant differences.

Referee #3:

The authors have addressed all our concerns. Yet, I do think that this is not the case for some concerns raised by the two other reviewers, and in my opinion, the authors should resolve these since the comments seem pertinent.

We are hopeful that we have addressed the remaining points to everyone's satisfaction.

Dear Dr Bühler,

Thank you for submitting the revised version of your manuscript. I have now evaluated your amended manuscript and concluded that the remaining minor concerns have been sufficiently addressed.

I am thus pleased to inform you that your manuscript has been accepted for publication in the EMBO Journal.

On a different note, I would like to alert you that EMBO Press offers a format for a video-synopsis of work published with us, which essentially is a short, author-generated film explaining the core findings in hand drawings, and, as we believe, can be very useful to increase visibility of the work. Please see the following link for representative examples and their integration into the article web page:

<https://www.embopress.org/doi/full/10.15252/emj.2019103932>

Finally, we have noted that the submitted version of your article is also posted on the preprint platform bioRxiv. We would appreciate if you could alert bioRxiv on the acceptance of this manuscript at The EMBO Journal in order to allow for an update of the entry status. Thank you in advance!

Best regards,

Daniel Klimmeck

Daniel Klimmeck, PhD
Senior Editor
The EMBO Journal
EMBO
Postfach 1022-40
Meyerhofstrasse 1

D-69117 Heidelberg
contact@embojournal.org
Submit at: <http://emboj.msubmit.net>

t